# Organic radical ferroelectric crystals with martensitic phase transition

Nan Zhang[1,4], Wencong Sun [2,4], Yao Zhang[1], Huan-Huan Jiang[1], Ren-Gen Xiong [1], Shuai Dong [2] ✉ & Han-Yue Zhang [3] ✉

Organic martensitic compounds are an emerging type of smart material with intriguing physical properties including thermosalient effect, ferroelasticity, and shape memory effect. However, due to the high structural symmetry and limited design theories for these materials, the combination of ferroelectricity and martensitic transformation has rarely been found in organic systems. Here, based on the chemical design strategies for molecular ferroelectrics, we show a series of asymmetric 1,4,5,8-naphthalenediimide derivatives with the homochiral amine and 2,2,6,6-tetramethylpiperidine-$N$-oxyl components, which adopt the low-symmetric polar structure and so allow ferroelectricity. Upon H/F substitution, the fluorinated compounds exhibit reversible ferroelectric and martensitic transitions at 399 K accompanied by a large thermal hysteresis of 132 K. This large thermal hysteresis with two competing (meta)-stable phases is further confirmed by density functional theory calculations. The rare combination of martensitic phase transition and ferroelectricity realizes the bistability with two different ferroelectric phases at room temperature. Our finding provides insight into the exploration of martensitic ferroelectric compounds with potential applications in switchable memory devices, soft robotics, and smart actuators.

Ferroelectric is a type of important functional material whose ferroelectric state can be modulated under various external stimuli including electric, light, magnetic, and thermal fields[1–5]. This kind of material basically undergoes solid-solid structural phase transition and thus has at least two stable low-energy states that can be modulated. Such switchable feature makes them promising candidates for sensors, switches, and memory devices[6]. In pursuit of biocompatibility and light weight, organic multi-functional ferroelectric materials are highly desirable for smart medicine and wearable electronics[7,8]. Organic radicals are attractive building blocks for achieving multifunctionality since they usually exhibit hysteretic responses to external stimuli[9,10]. 2,2,6,6-Tetramethylpiperidine-$N$-oxyl (TEMPO) is well-known as a stable single-component radical compound[11]. Unfortunately, this ferroelectric-ferroelastic compound has a limited working-

temperature range of a relatively low phase transition temperature ($T_c$) of 287 K and a low melting point of 311 K (Supplementary Fig. 1). Recently, another organic radical ferroic crystal [(NH₃-TEMPO)([18] crown-6)](ReO₄) has been found but it is devoid of structural phase transition (Supplementary Fig. 1)[12]. Thus, it is challenging to construct organic single-component radical ferroelectric compounds with high-$T_c$ phase transition behavior.

One type of typical solid-solid phase transition is martensitic transition, which involves a cooperative and collective displacement of molecules with lattice strain[13,14]. This unique transition can be associated with ferroelectricity in some inorganic oxides and organic crystals, such as Y-doped HfO₂[15], guanidinium nitrate[16] and diisobutylammonium bromide[17]. During the martensitic phase transition, many interesting phenomena would occur including the shape memory

[1]Jiangsu Key Laboratory for Science and Applications of Molecular Ferroelectrics, Southeast University, Nanjing 211189, P. R. China. [2]Key Laboratory of Quantum Materials and Devices of Ministry of Education, School of Physics, Southeast University, Nanjing 211189, P. R. China. [3]Jiangsu Key Laboratory for Biomaterials and Devices, School of Biological Science and Medical Engineering, Southeast University, Nanjing 210009, P. R. China. [4]These authors contributed equally: Nan Zhang, Wencong Sun. ✉e-mail: sdong@seu.edu.cn; zhanghanyue@seu.edu.cn

effect[18,19], superelasticity[20], negative thermal expansion[21], and thermosalient effect[22–25]. Among them, thermosalient effect could show larger actuation by an order of magnitude in comparison with the conventional converse piezoelectric effect which is well-known in ferroelectric materials[26,27]. Therefore, martensitic phase transition can endow ferroelectric materials with not only modulable physical properties but also intriguing functions. The combination of ferroelectricity and martensitic transition will realize the "all-in-one" effect (where multiple functions coexist in one material) and bring insight into the application of ferroelectric materials. However, pure organic single-component ferroelectric compound accompanying martensitic transition has rarely been found to date.

Structurally, organic martensitic compounds usually form lamellar or herringbone packing structures by weak intra- and inter-layer interactions, such as π–π stacking, hydrogen bonds, and van der Waals interactions[13,15]. Thus, we select 1,4,5,8-naphthalenediimide (NDI) derivatives—a type of electron-deficient planar aromatic compounds—as the platform for the design of martensitic ferroelectrics[28,29]. Firstly, to induce ferroelectricity, we introduce a homochiral $\alpha$-methyl benzylamine to one side of NDI according to the design strategy of introducing homochirality[30,31]. Further, NH$_2$-TEMPO is applied to the other side to construct organic radical ferroelectrics with intriguing multi-functional properties. Based on these strategies, we synthesize a series of ferroelectric compounds (N-[(S)- and (R)-1-phenylethyl]-N'-[1-oxyl-2,2,6,6-tetramethyl-4-piperidinyl]-1,4,5,8-naphthalenediimide (S- and R-H) and their fluorinated counterparts (N-[(S)- and (R)-1-(4-fluorophenyl)ethyl]-N'-[1-oxyl-2,2,6,6-tetramethyl-4-piperidinyl]-1,4,5,8-naphthalenediimide, S- and R-F) (Supplementary Fig. 1). In comparison with the parent compounds S-/R-H, the fluorinated compounds undergo martensitic phase transitions at 399 K with large thermal hysteresis ($\Delta T$) of 132 K. The density functional theory (DFT) results show that this thermal hysteresis can be attributed to the large barrier and entropy change during this martensitic phase transition. These features realize wide temperature windows of bistable ferroelectric states beyond all of the molecular ferroelectrics. Contrary to the traditional high-temperature ferroelectrics with only one ferroelectric phase at room temperature, S- and R-F show bistability with two different ferroelectric phases and thereby both of the low- and high-temperature ferroelectric phases can

be investigated and utilized at room temperature (Fig. 1). To our knowledge, organic radical ferroelectric crystals having martensitic phase transition accompanied by the "jumping crystal" effect has not been reported before. This work will provide guidance for the chemical design of emerging organic martensitic ferroelectric compounds with unique effects.

## Results

A series of NDI derivatives, S-F, R-F, Rac-F, S-H, R-H, and Rac-H were synthesized with the utilization of NH$_2$-TEMPO and $\alpha$-methyl benzylamine with corresponding chirality. Their phase purities and high thermal stabilities (up to 557 K) were confirmed by powder X-ray diffraction (PXRD) measurements (Supplementary Fig. 2), infrared (IR) spectroscopy (Supplementary Fig. 3), and thermogravimetric analyses (TGA) (Supplementary Fig. 4). The enantiomorphic feature of the synthesized compounds was confirmed by their vibrational circular dichroism (VCD) and circular dichroism (CD) spectra (Supplementary Figs. 3 and 5). Electron paramagnetic resonance (EPR) measurements were also employed to prove the existence of TEMPO radicals. The prominent EPR signals of S- and R-F powders were observed with g values of 2.00717 and 2.00751, respectively (Supplementary Fig. 6).

### Phase transition behaviors

Differential scanning calorimetry (DSC) was used to investigate the thermodynamic phase transition of NDI derivatives (Fig. 2a and Supplementary Fig. 7). As shown in Supplementary Fig. 7, we did not observe the structural phase transitions of the parent compounds S-/R-H and racemic compounds Rac-H/F. H/F substitution can maintain the crystallography point group and provides possibilities to induce phase transition through the regulation of molecular rotational energy barriers and intermolecular forces[30,32,33]. As expected, the fluorinated compounds showed thermal anomalies at 399 K upon heating and 267 K upon cooling in both powder and single-crystalline forms (Fig. 2a and Supplementary Fig. 7b). They exhibited reversible first-order phase transitions with large thermal hysteresis ($\Delta T$) of 132 K, which was also confirmed by Raman spectra (Supplementary Fig. 8). On the basis of Boltzmann's equation, N, the ratio of the number of possible configurations in the HTP and LTP, was deduced to be 4.3 and 5.1 for S- and

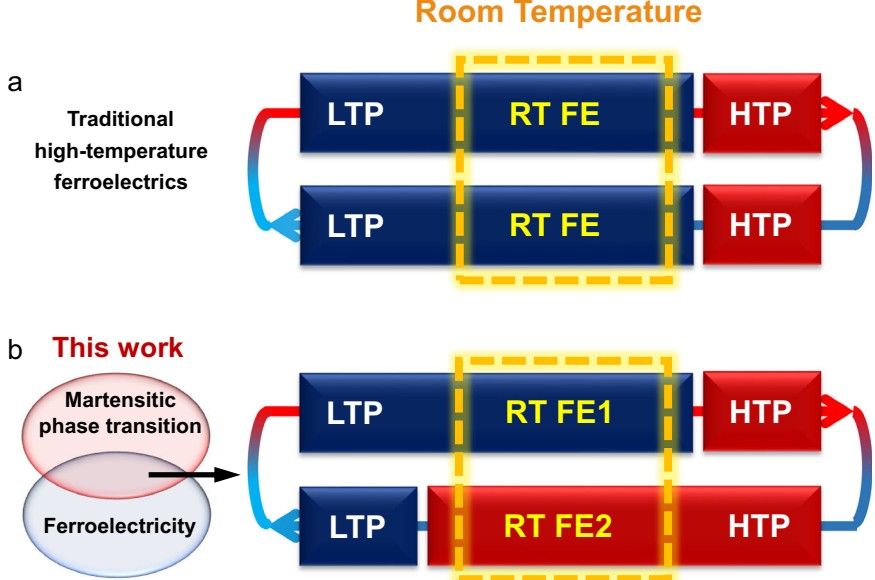

**Fig. 1 | Comparison between traditional high-temperature ferroelectrics and S-/R-F. a** Traditional high-temperature ferroelectrics show only one ferroelectric phase at room temperature. **b** The combination of martensitic phase transition and ferroelectricity in S-/R-F brings forth bistability with two different ferroelectric phases at room temperature. LTP and HTP represent low-temperature phase and high-temperature phase, respectively. RT FE, RT FE1, and RT FE2 represent different ferroelectric phases at room temperature.

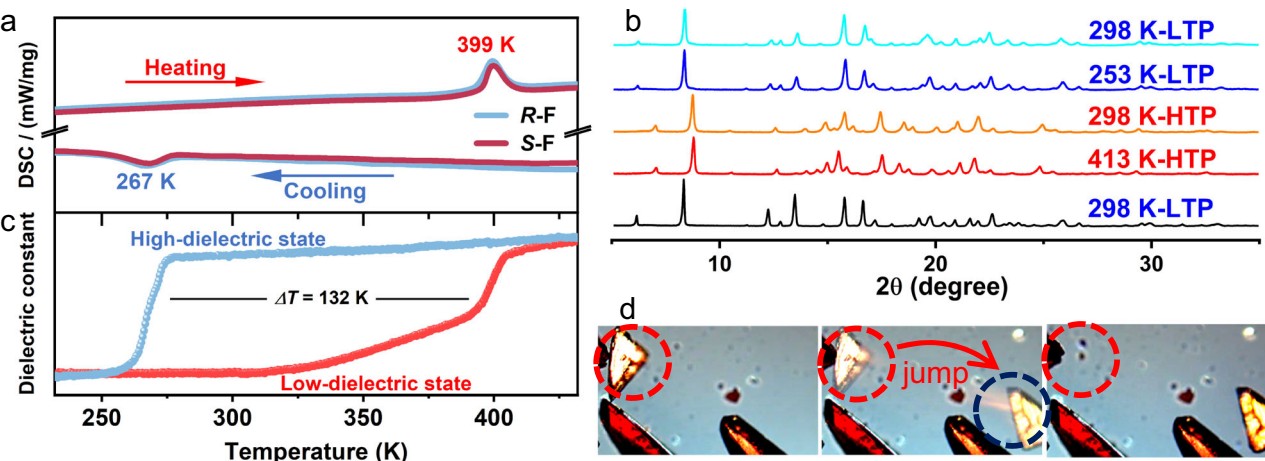

**Fig. 2 | Phase transition of the fluorinated compounds. a** DSC curves of *S*- and *R*-F powders. **b** Temperature-dependent PXRD results of *R*-F. **c** Temperature-dependent dielectric constant curves of *S*-F. **d** Optical images of the "jumping crystal" effect in *R*-F crystals upon heating.

*R*-F respectively (Supplementary Fig. 7b)[34]. This indicates that *S*- and *R*-F undergo order-disorder phase transitions.

To further investigate the structural phase transition with large $\Delta T$, the temperature-variable PXRD measurements were carried out at the temperature range of 253–413 K. With the temperature heating up/cooling down to $T_c$, the diffraction peaks at 6–8° shifted and peaks at 12–20° merged or split, confirming the occurrence of phase transition (Fig. 2b and Supplementary Fig. 9). We found that the HTP of the fluorinated compounds remained stable when the temperature dropped to room temperature, resulting in the bistability over large temperature range. Such a rare bistable feature was investigated by the dielectric measurement. As shown in Fig. 2c and Supplementary Fig. 10, the step-like anomalies enclosed a rectangular loop with a wide temperature window of 132 K, indicating improper ferroelectricity[6]. The fluorinated compounds are in the low-dielectric states below 399 K in the heating run. As temperature rose to 399 K, the dielectric constants of *S*- and *R*-F changed from low-dielectric states to high-dielectric ones. When cooling down to room temperature, the high-dielectric states maintained because of the large thermal hysteresis resulted from the martensitic transition. Therefore, both two distinct dielectric states could exist and be utilized at room temperature, which is seldom found in other ferroelectrics[30].

To our knowledge, this large thermal hysteresis of *S*-/*R*-F is rare in molecular ferroelectrics whose $\Delta T$ do not exceed 70 K[35] (Supplementary Tables 1 and 2). However, it is comparable to some inorganic oxides with martensitic transitions generally accompanied by large shear strain such as bulk HfO$_2$ (-120 K) and ZrO$_2$ (-150 K)[15]. This implies that *S*- and *R*-F probably undergo martensitic phase transitions where large shear strain would occur by lattice deformation[36]. During the martensitic phase transition, shape memory[18,19], self-actuation[37], or explosion[38] could occur with the release of elastic energy. A common phenomenon observed in the thermally induced phase transition of organic compounds is thermosalient effect[13]. Such a phenomenon was first observed in 1983[39], but did not attract much attention till 2010 when the emerging "jumping crystal" effect on the anticholinergic agent was investigated by ref. 38. As expected, some unusual behaviors were noted when we optically followed the behavior of crystal *R*-F upon heating above $T_c$ on the microscope hot stage. When the crystalline phase transferred from LTP to HTP, some crystals jumped, even hopping off the view (Supplementary Movie 1). Figure 2d exhibited the forward (marked in the red cycles) and backward (marked in the blue cycle) positions of a "jumping crystal". It is noteworthy that the jumping distance is about 275% of its crystal size, which demonstrates the large actuation far beyond that of the converse piezoelectric effect[40].

## Crystal structures of NDI derivatives

The crystal structures of the synthesized compounds were determined by single-crystal X-ray diffraction (XRD) measurement at 298 K. *Rac*-H and *Rac*-F both crystallized in the non-chiral polar space group *P*c (Supplementary Fig. 11, Tables 3 and 4). Based on the introduction of homochirality, *S*-/*R*-H and *S*-/*R*-F belong to the chiral-polar space group *P*2$_1$ as expected (Fig. 3a, Supplementary Fig. 12, and Tables 3 and 4). Their polar structures were confirmed by second harmonic generation (SHG) measurements (Supplementary Fig. 13). Due to the close van der Waals radius and similar steric parameters of H and F atoms, the F-substituted compounds and the parent ones show similar crystal structures and crystallographic point groups[32]. Their asymmetric units consist of one corresponding NDI derivatives molecule, where the chiral C atom has the "*R/S*" conformation, indicating their enantiomeric feature. The parallel molecules form a one-dimensional column along the *a*-axis, where the NDI units are connected through the weak interlayer π-π interactions (Supplementary Fig. 14). From the packing view, the NDI derivative molecules are symmetrically aligned along the 2$_1$ screw axes ([0 1 0] direction), resulting in the formation of spontaneous polarization as well as ferroelectricity (Fig. 3a). Differently, the introduction of F atoms changes the crystal packing environment, that is, the intermolecular interactions[41] (Supplementary Fig. 15). The molecular Hirshfeld surface and the related 2D-fingerprint plot calculations indicated that the fluorination brought forth C−H···F interactions and enhanced other intermolecular interactions such as π-π stacking between the NDI planes (Supplementary Fig. 16). Accordingly, the newly formed and enhanced intermolecular interactions may play an important role in inducing phase transition in the fluorinated compounds.

To further investigate the possible mechanism of structural transition, the variable-temperature single-crystal XRD was employed to study the crystal structures of *S*-/*R*-F in the HTP. The enantiomeric fluorinated crystals also adopt the chiral-polar space group *P*2$_1$ at 418 K (Fig. 3a and Supplementary Table 3). This indicates that *S*- and *R*-F undergo isostructural phase transitions. As the temperature increased, the TEMPO radical units displayed a partially disordered state, exhibiting a back-and-forth vibrating motion, and thus the C−H···F interactions disappeared in the HTP (Fig. 3b). This transition from LTP (298 K) to HTP (418 K) is accompanied by dramatic mechanical movements of *S*-/*R*-F, which could be associated with the expansion of the unit cell along *a* by 10.4/10% and *b* by 7.2/7.4% and contraction along *c* by 12.6/12.5%, while the overall volume barely changes (Fig. 3c–d and Supplementary Table 5). The lattice deformation could be attributed to the interlayer shearing movement. This movement is

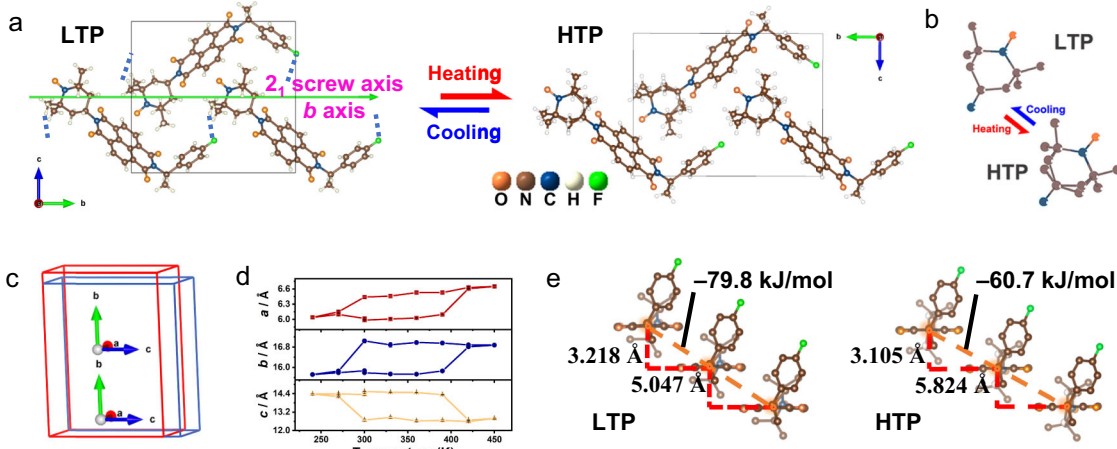

**Fig. 3 | Crystal structures of *S*-F. a** Packing views of *S*-F at LTP and HTP along the *a*-axis. Green arrow represents $2_1$ screw axes. Blue dotted lines represent C–H...F interactions. **b** Partial schematic diagrams of TEMPO units at LTP and HTP. **c** Comparison of the cell lattices in the LTP (blue) and HTP (red). **d** Variation of unit cell parameters in the heating-cooling mode. Abrupt changes correspond to the phase transition. **e** Schematic diagrams of the distance and interaction energy between adjacent molecules at LTP and HTP. H atoms are partly omitted for clarity.

specifically realized as that the distance between π-stacked NDI units decreased by 0.113 Å in the vertical direction and increased by 0.777 Å in the lateral direction from LTP to HTP (Fig. 3e). The interaction energy of the π-π interaction between NDI units was calculated by the UNI force field in the Mercury program[42]. The calculated energies are –79.8 kJ mol$^{-1}$ in the LTP and –60.7 kJ mol$^{-1}$ in the HTP, respectively (Fig. 3e). This energy change indicates that the shear strain is generated by lattice deformation[37,43]. Consequently, the order-disorder change of TEMPO units and the lattice deformation cooperatively promote the martensitic phase transition under thermodynamic drive.

## DFT calculations

To further understand the aforementioned experimental observations, preliminary physical calculations have been performed based on DFT and thermodynamics. Our DFT calculation is performed using Vienna ab initio Simulation Package[44] and the details can be found in Methods section. The structures of both LTP and HTP are optimized rationally for accurate calculation. For HTP, the eclectic positions of partially occupied C and H ions are chosen to mimic their occupations, as shown in Supplementary Fig. 17. Our DFT results suggest that both HTP and LTP of *S*-/*R*-F are stable during the structural relaxation. The optimized lattice constants are close to their experimental one, where the lattice constants *a*, *b*, *c* of LTP and HTP are significantly different by −5.58%, −7.43%, and 13.19%, respectively, as compared in Supplementary Table 6. What's more, the internal energy of HTP is indeed higher (51.8 meV/f.u.) than the LTP one, as expected. The above analysis shows that there is indeed a structural phase transition in *S*-/*R*-F. However, the optimized lattice constants of LTP and hypothetic HTP (derived from the F-case) of *S*-/*R*-H are very close: −0.05%, −1.75%, 1.24% for *a*, *b*, *c*, respectively. Considering the numerical precision and soft van der Waals (vdW) packing between molecules, such tiny differences are in the tolerant range to conclude that no phase transition occurred for *S*-/*R*-H, which is consistent with the experimental observation. DFT calculation also indicates a local magnetic moment ~1 μ$_B$ /f.u. arises from the unpaired electron of the covalent N–O bond (N: ~0.5 μ$_B$; O ~ 0.5 μ$_B$), as shown in Fig. 4a. However, due to the long distance and vdW gap, the magnetic coupling between the nearest neighbor moments is very weak, which will lead to magnetic ordering only in low temperature (much below the room temperature).

In principle, the above DFT results are obtained at zero temperature. To understand the martensitic phase transition at finite temperatures and atmospheric pressure, the Gibbs free energies of HTP and LTP should be compared. Analytically, the Gibbs free energy can be expressed as $G = E - TS + PV$, where *E* denotes the internal energy (which can be obtained from DFT), *T* is the temperature, *S* is the entropy, *P* and *V* are pressure and volume respectively. Due to the partial occupancy of C and H ions in the HTP, there is more than one possible configuration of HTP (characterized by an effective *N* as defined before), which leads to a larger entropy. Ideally, the value of *N* should range from 2 (the limit of fully locked C–H partial occupations) to $2^5$ (the limit of fully free C–H partial occupations). Then, the relative free energy at finite temperatures (399 K and 267 K) can be derived as a function of *N*, as shown in Fig. 4b. It is clear that when $4.4 < N < 9.3$, the free energies of HTP and LTP satisfy the phase transition condition, namely the $F(HTP) < F(LTP)$ at 399 K while $F(HTP) > F(LTP)$ at 267 K. As mentioned before, the experimental value of *N* is ~4.3–5.1, which almost falls in this region. Then taking $N = 5$, for example, the phase transition process is simulated using Climbing Image Nudged Elastic Band (CI-NEB) method. An animation of the HTP-LTP transition trace in our CI-NEB simulation is shown in Supplementary Movie 2. As shown in Fig. 4c, there are high energy barriers (~20–30 meV/f.u.) during the phase transition between LTP and HTP. Taking the molecule as an entirety (i.e., by neglecting its internal degree of freedom), its thermal energy can be expressed as $E = 3/2 k_B T$ ($k_B$: Boltzmann constant). Then, the magnitude estimation of active temperatures corresponding to the energy barriers leads to 150 - 230 K, which are just in the same order of magnitude as thermal hysteresis. In short, our DFT calculation and thermal dynamics analysis agree well with the experimental observation of martensitic phase transition.

## Characterization of ferroelectricity

The higher electronegativity of F atoms may result in a larger dipole moment and further enhance ferroelectricity[32]. To estimate the ferroelectric polarization of the crystal, we calculated the vector sum of the corresponding molecular dipole moment in the unit cell (Supplementary Fig. 18). As expected, the estimated saturated polarization ($P_s$) value of *S*-F is about 0.83 μC cm$^{-2}$ (along the *b*-axis), larger than that of the parent one (*S*-H, 0.23 μC cm$^{-2}$). To determine the ferroelectricity of *S*- and *R*-F, we carried out the polarization–voltage (*P–V*) hysteresis loop measurements using the double-wave method on their thin films (Supplementary Fig. 19)[45]. As shown in Fig. 5a, two opposite peaks caused by the charge displacement appeared in the current density–voltage (*J–V*) curves, which indicated two opposite stable polarization states. By integrating the current density, we obtained the

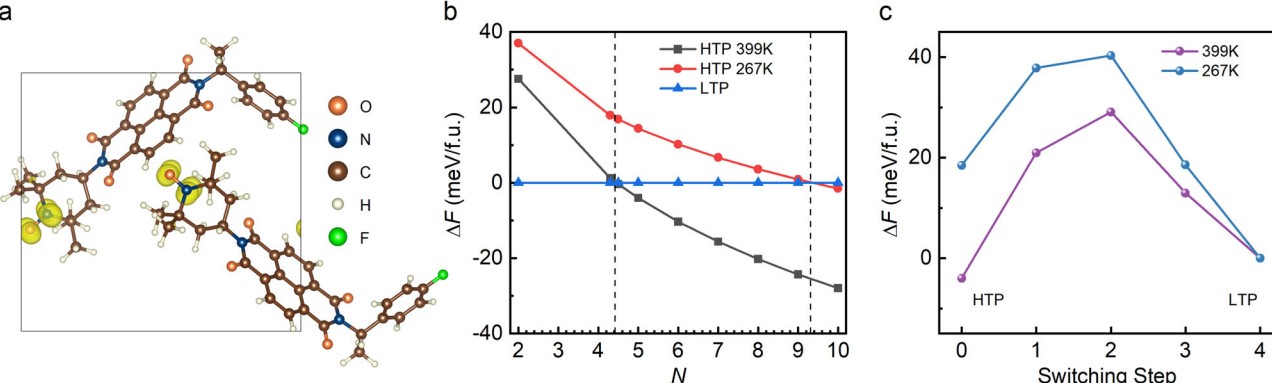

**Fig. 4 | Simulation of phase transition between HTP and LTP. a** Spin density plot of magnetic state. The local magnetic moments are from the O and N ions, while the neighboring N−O pairs are far from each other. **b** Relative free energy of HTP as a function of effective disorder (characterized by the state number $N$ as mentioned before) under different temperatures. The color middle region: HTP owns a higher (lower) free energy than LTP at 267 K (399 K). The experimental $N$ is just in this range. The free energy of LTP is taken as the base. **c** The simulated transition barriers between HTP to LTP with $N = 5$, at 399 K and 267 K, respectively. The barrier heights, defined from the top to the high free energy side are about 29.06 meV (399 K) and 21.84 meV (267 K), respectively.

typical $P−V$ hysteresis loop, indicating the existence of ferroelectricity in $S$- and $R$-F at LTP (Fig. 5a and Supplementary Fig. 20). We have also conducted $P−V$ hysteresis loop measurements at room temperature on the thin film of $S$-F which was annealed at 413 K for 10 min to obtain the HTP. The typical $P−V$ hysteresis loop confirms the ferroelectricity of HTP (Supplementary Fig. 21). According to the $P−V$ hysteresis loops, the $V_c$ of HTP is lower than that of LTP. This could be attributed to the weaker interactions between the molecules in the HTP, which enables easier switching of ferroelectric polarization. The lower $V_c$ makes HTP a useful state whose ferroelectricity is easier to be investigated and used. Thus, the LTP and HTP of fluorinated compounds have different ferroelectric behaviors and both of them can be investigated and utilized at room temperature.

Piezoresponse force microscopy (PFM) was further carried out to study the ferroelectric behaviors of $S$- and $R$-F from the microscopic perspective. PFM has long been accepted as a powerful tool for the visualization of static domains and the manipulation of dynamic switching at the micrometer scale[46,47]. As shown in Fig. 5b−c and Supplementary Fig. 22, we observed apparent phase contrasts and domain walls on the thin films of $S$-/$R$-F at LTP, and the domain patterns did not overlap with their topography. Then, as the essential properties of ferroelectric materials, domain switching ability can be observed by the switching spectroscopy PFM (SS-PFM)[48]. We recorded the typical rectangle-like phase and butterfly-like amplitude PFM loops, indicating the pointwise polarization by measuring the phase and amplitude signal on the DC tip bias on a single point (Supplementary Figs. 22−23). Typically, the polarization switching process involves four steps: (1) domain nucleation, (2) forward extension, (3) lateral expansion, and (4) domain merging[49]. The switchable ferroelectric behavior was visualized on the thin films of $S$- and $R$-F, where their ferroelectric domain can be reversed and switched back with the application of opposite voltage (Fig. 5d−f and Supplementary Fig. 22). As shown in Fig. 5, the domain switching process of compound $S$-F only involves the domain wall movement instead of the domain nucleation because of the relatively high coercive voltage ($V_c$). These provide solid evidence for the ferroelectricity of $S$- and $R$-F at LTP. To further investigate their ferroelectric behavior in the HTP, we carried out the PFM measurement at room temperature on the annealed thin film of $S$-F. We observed the ferroelectric domains and these domains can be electrically switched, confirming the ferroelectricity of compound $S$-F in the HTP (Supplementary Fig. 23). Besides, we also confirmed the ferroelectric polarization switching of $S$-H on its thin films (Supplementary Fig. 24).

## Magnetic and optical properties
As a type of ferroelectric material that has organic radicals, we also investigated the magnetic properties of $S$- and $R$-F. The variable-temperature magnetic measurements were performed for the fluorinated compounds under a DC field of 1.0 kOe from 2 to 300 K (Supplementary Fig. 25). The room-temperature $\chi_M T$ values are 0.3698 and 0.3798 cm$^3$ K mol$^{-1}$ for $S$- and $R$-F, respectively, which is in good agreement with the $S = 1/2$ spin[12]. Such a $\chi_M T$ value indicates the existence of an unpaired electron. With the decrease in temperature, $\chi_M T$ gradually decreased. This indicated the paramagnetic property of $S$- and $R$-F, which was also confirmed by the Curie−Weiss law (Supplementary Fig. 25)[50]. For the racemic compound, the $\chi_M T−T$ plot displayed a horizontal line with a value of around 0.37, suggesting a similar spin-half paramagnetism (Supplementary Fig. 25).

Solid-state Ultraviolet−visible (UV/Vis) absorption spectra were recorded on the powder samples of the fluorinated compounds to investigate the optical behaviors. The clear absorption edge of $S$-F at around 780 nm was noticed (Supplementary Fig. 26a). According to the *Tauc* plot, the optical band gap of $S$-F was calculated as 1.59 eV (Supplementary Fig. 26a, inset). This was also verified by DFT calculations (Supplementary Fig. 27). Additionally, the UV/Vis absorption changed under the irradiation of 365 nm light (Supplementary Fig. 28), indicating the generation of a free radical state [NDI•]$^{-}$[51]. The parent compounds showed similar optical behaviors in comparison with the fluorinated ones (Supplementary Figs. 26 and 28).

## Discussion
The combination of the martensitic phase transition and ferroelectricity endows $S$- and $R$-F with many advantages over the other ferroelectric and thermosalient materials. This martensitic phase transition brings forth a large $\Delta T$ crossing the high-temperature and low-temperature regions and thus realizes the wide bistability beyond all of the molecular ferroelectrics (Supplementary Tables 1 and 2). For most high-temperature ferroelectric compounds, their high-temperature ferroelectric or paraelectric states can only be obtained at high temperature, which somewhat impede their study and application (Fig. 1)[30]. However, for martensitic ferroelectrics $S$- and $R$-F, their physical properties especially ferroelectricity of HTP can be retained at room temperature. Therefore, both the high- and low-temperature ferroelectric states for $S$- and $R$-F can be investigated and utilized at room temperature, which is seldom found in other high-temperature ferroelectric materials[30] (Fig. 1). The feature of bistable ferroelectric states at room temperature makes $S$ and $R$-F a type of functional

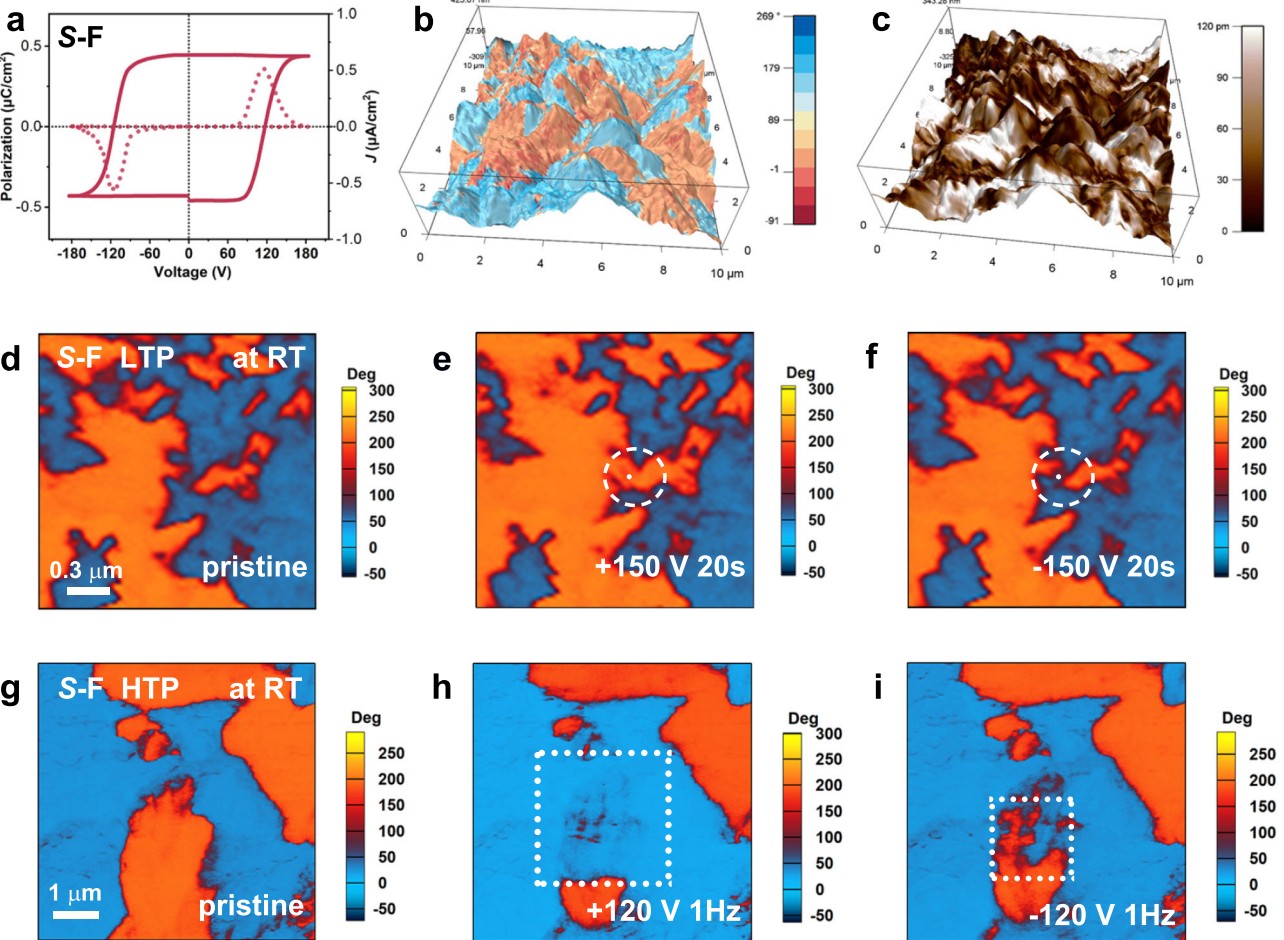

**Fig. 5 | Ferroelectricity of S-F. a** The *J–V* (dotted) and *P–V* (solid) curves of *S*-F showing a typical ferroelectric hysteresis loop. Vertical PFM phase (**b**) and amplitude (**c**) images of *S*-F overlaid on the 3D topographic image. **d–f** The electrical switching of the ferroelectric domain at LTP by applying ±150 V voltage on the white dots for 20 s: (**d**) pristine, (**e**) after applying a positive bias voltage, and (**f**) after applying a negative bias voltage. The regions of domain wall movement are marked with white dashed circles. **g–i** The electrical switching of the ferroelectric domain at HTP by applying ±120 V voltage in the white boxes: (**g**) pristine, (**h**) after applying a positive bias voltage, and (**i**) after applying a negative bias voltage.

material with great application potential in next-generation smart memory devices with multiple physical channels. Besides, ferroelectric compounds can induce mechanical strain under the electric field[40]. Meanwhile, this martensitic phase transition causes thermosalient effect, which could show larger actuation by an order of magnitude in comparison with the conventional converse piezoelectric effect. Thus, the actuation of *S*- and *R*-F can realize the modulation between the millimeter-scale (thermodynamic control) and the picometer-scale movement (electric control)[52]. This rare feature would satisfy application scenarios requiring multiple-scale actuation, thereby making *S*- and *R*-F good candidates for lightweight actuators, sensors, and soft robots[14].

In summary, we have presented reversible martensitic transitions in a pair of organic single-component radical ferroelectric crystals *S*- and *R*-F. During this transition, they undergo high-temperature isostructural phase transformations with large Δ*T* of 132 K. Such a large Δ*T* could be ascribed to the martensitic transition, where large activation energy (temperature change) is required[43]. These experimental results are confirmed by DFT calculations. The ferroelectricity of *S*- and *R*-F in both LTP and HTP is confirmed by PFM and *P–V* hysteresis loop measurements. The combination of martensitic phase transition and ferroelectricity realizes the bistability in multiple channels over large temperature range and bistable ferroelectric states at room temperature. This rare

feature will inspire many interesting applications for smart memory devices and mechanical and/or electrical actuators[16,53].

## Methods

### Preparation of NDI derivatives and thin films

All reagents were purchased from commercial sources and used without further purification. 1,4,5,8-Naphthalenetetracarboxylic dianhydride (5 mmol, 1 equiv.), 4-amino-tempo (5 mmol, 1 equiv.) and (*S*)-1-phenylethanamine (5 mmol, 1 equiv.) were taken into 30 mL of *N*, *N*-dimethylformamide and 2 mL $CH_3COOH$ was added. The reaction mixture was stirred at 100 °C for 2 h. The distilled water was added to the reaction solution until the precipitate is no longer produced. The purple precipitation was filtered and dried at 70 °C. The crude product was purified by column chromatography (silica gel, 5:1 (v/v) petroleum ether/ethyl acetate) to obtain the target compound *S*-F. The purple crystals of *S*-F were recrystallized in the mixed solution of dichloromethane and ethyl acetate. The other compounds were synthesized in the same way by replacing them with the corresponding 1-phenylethanamine. The precursor solutions of *S*-F, *R*-F, and *S*-H were prepared by dissolving 10 mg of crystals in 200 µL of dichloromethane. 20 µL of precursor solution was spread on a clean ITO-coated (indium tin oxide) glass (1 × 1 cm²). After evaporating the solution at room temperature, the thin films were obtained.

## Elemental analyses and high-resolution mass spectroscopy (HRMS)

Elemental analyses were carried out on an Elemantar Vario EL cube with CHNO mode. The mean content values of C, H, N and O were calculated from twice the measurements of each compound. HRMS was obtained using a Q-TOF instrument equipped with an ESI source.

$S$-F ($C_{31}H_{29}FN_3O_5$•, 542.21). Elemental analyses: calcd C, 68.62; H, 5.39; N, 7.74; O, 14.74. found C, 68.53; H, 4.95; N, 7.71; O, 14.77. HRMS (ESI, $m/z$) calcd: 543.2125 $[M + H]^+$; found: 543.2161. (Supplementary Fig. 29)

$S$-H ($C_{31}H_{30}N_3O_5$•, 524.22). Elemental analyses: calcd C, 70.98; H, 5.76; N, 8.01; O, 15.25. found C, 70.71; H, 5.15; N, 8.01; O, 15.28. HRMS (ESI, $m/z$) calcd: 525.2219 $[M + H]^+$; found 525.2257. (Supplementary Fig. 30)

## Thermal analyses measurements

DSC measurements were recorded on a NETZSCH DSC 200F3 instrument. The samples were loaded in aluminum crucibles and were heated and cooled repeatedly with a rate of 20 K min$^{-1}$ at the nitrogen atmosphere. TGA were carried out on a PerkinElmer TGA 8000 instrument by heating samples with a rate of 40 K min$^{-1}$ under the nitrogen atmosphere.

## SHG and dielectric measurements

SHG experiments were performed on an unexpanded laser beam with low divergence (pulsed Nd:YAG at a wavelength of 1064 nm, 5 ns pulse duration, 1.6 MW peak power, 10 Hz repetition rate). The instrument model is Ins1210058, INSTEC Instruments, while the laser is Vibrant 355 II, OPOTEK. Complex dielectric permittivity was measured with a TH2828A impedance analyzer. The conductive silver adhesive applied to surfaces of samples was used as top and bottom electrodes.

## IR, VCD, CD and UV/Vis Spectroscopy

The samples were mixed uniformly with potassium bromide at 1:400 (sample:KBr) ratio and then ground into powder. The powders were pressed into transparent sheets for IR and VCD measurements, which were performed on a Shimadzu model IR-60 spectrometer at room temperature and a Bruker PMA-50 instrument under nitrogen, respectively. The potassium bromide (KBr) tablets method was used for CD measurements in transmission mode. The CD spectra are recorded with SHIMADZU UV-3600Plus equipped with JASCO J-1700. The UV–vis absorption spectra were measured on a Shimadzu UV-3600Plus spectrophotometer equipped with the integrating sphere (ISR-603) with the wavelength range of 200–800 nm. BaSO$_4$ was used as a 100% reflectance reference. The reflectance data were then converted to absorbance data by using the Kubelka–Munk (K–M) equation.

## Single crystal and PXRD measurements

Crystallographic data were collected using a Rigaku Saturn 924 diffractometer equipped with temperature control device, by using Cu Kα (λ = 1.54184 Å) radiation. Data processing including empirical absorption correction, cell refinement, and data reduction was performed using the Crystal Clear software package. The structures were solved using the Olex2 program. PXRD data were measured on a Rigaku D/MAX 2000 PC X-ray diffractometer with Cu Kα radiation. Diffraction patterns were collected in the 2θ range of 5–35° with a step size of 0.02° from 253 K to 413 K.

## EPR and magnetic measurements

EPR spectra were recorded on a Bruker EMXplus-9.5/12 spectrometer. Magnetic measurements for the compound were performed on a Quantum Design MPMSXL5 SQUID system with powder samples.

## The interaction energy and Hirshfeld surfaces, fingerprint plots calculations, and experimental details of the calculation

Molecular Hirshfeld surface calculations were performed by using the CrystalExplorer17.5 program. All bond lengths to hydrogen were automatically modified to typical standard neutron values. In this study, all the Hirshfeld surfaces were generated using a standard (high) surface resolution. The intensity of molecular interaction is mapped onto the Hirshfeld surface by using a red-blue-white color scheme: the white regions exactly correspond to the distance of Van der Waals contact, the blue ones correspond to longer contacts, and the red ones represent closer contacts. The interaction energy calculation was using the CrystalExplorer17.5 program with the energy model of HF/3-21 G and the UNI force field as implemented in the Mercury program with the hydrogens all being normalized[42].

## Computational methods

The first principles calculations based on DFT were implemented by the Vienna ab initio simulation package[44]. The interaction between valence electrons and atomic cores was treated by the projector-augmented wave method[54], where the expansion term of the wave function ends at 450 eV. The generalized gradient approximation of Perdew-Burke-Ernzerhof (PBE) parameterization was used to deal with the electron-electron exchange-correlation potential[55]. The Monkhorst-Pack mesh with the density of 5 × 2 × 3 was used to integrate the Brillouin zone during optimization and static calculation. DFT-D3 method of Grimme with the zero-damping function was applied to describe properly van der Waals (vdW) interactions[56]. The convergence criterion of electron self-consistent iteration was set as 10$^{-5}$ eV/atom, and the ionic relaxation loop was not broken until the Hellman−Feynman force was less than 0.01 eV/Å. The transition barriers of HTP and LTP are calculated using the climbing image nudged elastic band method with three saddle points. Other calculation parameters are the same as static relaxation[57].

## PFM and $P−V$ hysteresis loop measurement

The PFM measurement was carried out on a commercial piezoresponse force microscopy (Cypher, Asylum Research) with a high-voltage package at room temperature. Conductive Pt/Ir-coated silicon probes (EFM, Nanoworld) were used for domain imaging and polarization switching studies, with a nominal spring constant of -2.8 nN/nm and a free-air resonance frequency of -75 kHz. Since the amplitude of the low-frequency vertical PFM was within the noise level of the quadrant photodetector of the AFM, we performed the PFM experiments at contact resonance. The polarization−voltage loop was obtained via the double-wave method. The sample film deposited on the ITO glass was connected to the testing system via two conductive probes, with one dipped with liquid InGa alloy placed on the sample and the other attached to the ITO coating. The testing system was composed of a voltage source (Trek 609E-6), a waveform generator (Keysight 33500B), and a current meter (Keithley 6517B). The typical $J−V$ curve has two peaks at voltages corresponding to the switching voltage $U_{switch}$ of the ferroelectric layer. Integration of the current over time gives the polarization curve:

$$P(U) = \frac{1}{S\nu_{sw}} \int IdU \qquad (1)$$

where $\nu_{sw}$ is the voltage sweep rate and $S$ is the area of the capacitor.

## Data availability

All data generated and analyzed in this study are included in the article and the Supplementary Information, and are also available from corresponding authors upon request. The crystal structures generated in this study have been deposited in the Cambridge Crystallographic Data Center under accession code CCDC: 2249888-2249895 and can

be obtained free of charge from the CCDC via www.ccdc.cam.ac.uk/data_request/cif.

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

## Acknowledgements

We thank Prof. Tao Liu and Prof. Yin-Shan Meng (Dalian University of Technology) for their help with magnetic measurements and thank the Big Data Center of Southeast University for providing the facility support on the numerical calculations. This work was supported by Seventh Youth Elite Scientist Sponsorship Program by the China Association for Science and Technology (H.-Y.Z.), Ten Science and Technology Problems of Southeast University (H.-Y.Z.), National Natural Science Foundation of China Grant No 21991142 (R.-G.X.), 21831004 (R.-G.X.), and 12274069 (S.D.), and Postgraduate Research & Practice Innovation Program of Jiangsu Province Grant No KYCX23_0222 (W. S.).

## Author contributions

N.Z., Y.Z., and H.-H.J. performed the experimental studies. N.Z. and H.-Y.Z. carried out the analysis. W.C.S. and S.D. performed the computational studies. R.-G.X. supervised the work. N.Z., W.C.S., S.D., and H.-Y.Z. wrote the manuscript. All authors commented on the manuscript.

## Competing interests

The authors declare no competing interests.
