## [Peer review file · Nature Communications]

REVIEWER COMMENTS

Reviewer #1 (Remarks to the Author):

In this manuscript, the authors report the first example of an organic ferroelectric compound that undergoes a martensitic transition accompanied by the very rare "thermosalient effect" with an unexpectedly large thermal hysteresis. They use a clever approach of asymmetric chemical substitution of a robust chemical core with two different substituents, one of which is chiral, while the other is a stable radical and acts as an unpaired electron-bearer. The material reported here is unique because it combines multiple physical facets, including a thermosalient effect, which is very rare and cannot be predicted or intentionally designed at the current state of knowledge. Given the special properties of the materials, I strongly recommend acceptance of this manuscript in Nature Communications. Having said that, after including all corrections, the authors must have their manuscript read by a native speaker. The poor language is distracting from the scientific content and makes the reading very difficult. Below listed are scientific and technical comments: I would suggest that the revised and substantially improved manuscript be re-reviewed before acceptance.

(1) The reported materials undergo structural phase transition at high temperature (399 K). This is rather high temperature, and not very compatible with the expected applications, which require transition around room temperature or slightly above and below. Can the authors comment on the advantages in comparison with materials that undergo ferroelectric transition close to room temperature (cite and compare with: Nat. Commun. (2022) 13, 2823)?

(2) On page 7, the thermal properties of the compounds are discussed. It is not clear whether these were recorded from powder samples, or from crystals. It is known that for the thermosalient compounds, these two states can give very different temperatures in the DSC and sometimes the thermosalient transition is even suppressed. Please provide DSC measurements on collection of single crystals and show them in the supporting information. If there are significant differences, please comment on that in the main text.

(3) From the sections discussing the electrical properties sections such as J-V, P-V, and PFM phase and amplitude, it is evident that the results confirm the ferroelectricity in the crystals. However, a thorough discussion regarding the physics of domain switching and its correlation to material structure seems to be lacking, and the discussion of that aspect is somewhat ambiguous. A more detailed description of the testing configuration for measurement of J-V, especially regarding the integration and display of P-V, would greatly improve the transparency and the reproducibility of the work presented.

(4) It is also worth noting that the thin films utilized in this study are not based on crystals, and appropriate structural confirmation is lacking. Additionally, the lack of a comparison to other published works on ferroelectricity of organic materials (see some references suggested in specific comments below) makes it very difficult to gauge the ferroelectric performance of this compound compared to its counterparts. Please include comparison and explain why this material is better or worse than the others ferroelectric and/or thermosalient materials.

(5) The article contains six composite images. I feel that some can be combined to decrease the total number of images in the main text, or shifted to the supplementary material. The quality of all figures should be improved.

Specific comments:

Abstract:

- "fluoridated" should be "fluorinated"

- Replace “experience reversible” with “exhibit reversible”
- I would suggest to tone down the expressions such as “ultra-large thermal hysteresis” to “large thermal hysteresis”
- Change “new type of martensitic ferroelectric compounds” to “rare martensitic ferroelectric compounds”

Introduction:

- The first sentence of the introduction starts with both “shape memory” and “a ferroelectric”, and it should be corrected.
- The phrases in the sentences “bio-friendliness, high energy efficiency and easy portability” does not read well. It should be modified to “biocompatibility, high energy conversion efficiency, and light weight” or similar.
- The first report of shape memory in organic crystal should be cited clearly: JACS, 2016, 138, 13298–13306, as well as a follow-up detailed study of the phenomenon: Nat. Commun., 2019, 10(1), 3723.
- Replace “The introducing of” with “The introduction of”
- Replace “is absent of” with “is devoid of”
- Page 4, line 57: relevant references should be added to support this statement. Here are some representative works that provide in-depth studies of these phenomena: thermosalient effect: Matter, 2019, 1, 1033; JACS, 2019, 141, 3371; photosalient effect: Crystal Growth and Design, 2015, 15, 1983–1990; negative thermal expansion due to thermosalient effect: Sci. Rep., 2016, 6, 29610.
- I was not able to find justification or support for the assignment of the several Raman bands on the bottom pf page 9. Are these assignments reliable? Please comment or remove this discussion if the assignments are empirical.
- Delete the phrase “To end this”. It has a different meaning from the use intended in this sentence.
- Replace “with unique superiority of” with “with advantageous”
- Replace “for chemically constructing” with “to chemically construct”
- Replace “to assemble a” with “to introduce a”
- Replace “the hit rate” with “the probability”
- Replace “ultra-large” and other terms containing the prefix “ultra” across the main text, the figures and the supplementary materials. Use “large” instead, it sounds more scientific.
- Replace “ultra-wide bistability” with “bistability over large temperature range”. Remove the prefix “ultra” throughout the text.
- The quality of the graphics in Figure 1 should be improved substantially.
- The chemical names of the compounds should be given in the caption, and the acronyms should be given on the figure to facilitate the identification of the chemical structures
- The sentence “the fluoridated compounds undergo a high-temperature structural phase transition at 399 K with an ultra-large thermal hysteresis (ΔT) of 132 K.” states “compounds”, yet only one temperature is given for the phase transition and for the hysteresis. Please check whether this is one compound or multiple compounds, and correct the sentence.

Results and discussion

- Change “infrared radiation (IR) analyses” to “Infrared (IR) spectroscopy”
- The phrase “isolated electron might be unstable” should be modified. It does not make sense.
- “Under the corresponding....level” should be “using the method”
- Page 7, line 132: replace “are absent from” with “are devoid of”
- Page 9, the sentence closing with “fluorinated molecular” is not complete.
- Replace “units are connecting” with “units are connected”
- Figure 3D: please show the orientation of the axes of both unit cells, using different colors.
- The caption of Figure 3 indicates “layers” but I am not able to see the layers in the figure, I can only see columns. Please revise the caption or change the figure.
- Figure 3E is missing errors bars with standard deviations. Please also include information on whether

these points were calculated based on limited number of reflections (mention how many) or on the full dataset. If it is the latter case, provide the full crystallographic information in a table in the supplementary information.

- Page 11: replace "their parent counterparts are absent of" with "their analogues are devoid of"
- Page 11, bottom: specify which temperatures (in K) do these changes refer to (before and after), it is difficult to find that information without the absolute values
- Line 234: The shape memory effect is not generally observed for thermosalient phase transitions. It has been observed only in a couple of cases, and it is very rare. Please separate this sentence into two, and include the relevant citations for the shape-memory effect (JACS, 2016, 138, 13298–13306; Nat. Commun., 2019, 10(1), 3723.)
- More movies are needed to support the observation of the "jumping crystal" effect (thermosalient effect)
- Replace "this mutual exclusion" with "this inconsistency"
- "KBr-doped powder sample" is not clear. Please rewrite.
- Replace "to carry out the solid-state UV/vis" with "to record the solid-state UV/vis"
- Replace "observe obvious" with "observe apparent"
- Lien 283: replace "chiral feature" with "chirality". Line 289: explain why the exposure to UV/vis light does not change the chirality
- Page 17: replace "domain switching ability should be investigated" with "the switching of the domains can be observed"

Reviewer #2 (Remarks to the Author):

In this paper, the authors report on a series of asymmetric 1,4,5,8-naphthalenediimide (NDI) derivatives. Among them, S and R-F compounds have been shown to exhibit martensitic transitions with ultra-large thermal hysteresis of 132 K and ferroelectric properties. However, there remains doubt about the authors' claim that they chemically designed materials that exhibit these properties. While introducing chirality into the molecule is effective in creating a polar structure, it is not effective in creating a ferroelectric material that requires polarization inversion under an electric field. In fact, many ferroelectric materials belong centrosymmetric space groups above their phase transition temperatures, making them difficult to discuss as the general strategy. Additionally, the authors claim that they aimed for multiferroic properties by introducing a TEMPO unit, but the compounds did not show ferromagnetic or antiferromagnetic behavior within the measured temperature. Furthermore, since there is no interaction between the important martensitic transition and ferroelectricity in S and R-F ferroelectric materials, these properties are considered independent, and it is difficult to consider them as new functional materials. On the other hand, this paper is based on very careful experiments and discussions, and a logical development is made. Therefore, it is recommended to submit it to another highly specialized journal.

Reviewer #3 (Remarks to the Author):

The authors present a detailed study of crystals formed from TEMPO based NDI derivatives, characterizing the phase transitions in each compound and ferroelectric properties of these systems. They discovered the enantiomerically fluorinated compound undergoes a martensitic phase transition as well as exhibit ferroelectric behavior. This is coupled with photochromism and paramagnetic behavior in the system. While certainly it is interesting to find this mix of different transitions and optoelectronic properties in rationally designed molecules, It is not clear how these properties benefit from each other with this system. As a result, I'm not sure if inspiring future materials with these properties is interesting enough to a broad audience to warrant publishing in nature communications. Additionally, the analysis and discussion seemed lacking in determining the transition mechanism. For these reasons, I must recommend against publishing this paper in Nature Communications. But with a more thorough analysis of these data and a deeper discussion of the results, I look forward

to reading a revised version elsewhere.

More detailed comments are discussed below:

1. In this study, the researchers investigate the inclusion of homochirality and fluorine substitution on the NDI-TEMPO molecule. While the reasons for including homochirality was discussed, the reasons for fluorination weren't mentioned and they found the hydrogen-based system also displayed ferroelectricity as well. I think discussion of the underlying reasons for this result as well as a comparison, if any, to the ferroelectric behavior would be important here.

2. Are crystal structures similar between the H- and F- molecules? While the R and S structures are compared, it would be nice to show a comparison between the H- and F- systems since the martensitic phase transition appears suppressed in those cases. I think it's mentioned later with respect to the ferroelectric properties but should be talked about in detail with the SCXRD.

3. The Isotope effect is presented as the reason for changes to the phase transition but that seems to neglect many important aspects of H/F substitution. Even in the paper, the effects of hydrogen bonding are discussed which don't occur in the H- based molecule. Additionally, the fluorine would act as an electron withdrawing group changing the pi-stacking interactions. All of these would likely have a stronger effect on the phase transition than just the isotope effect on rotation.

4. In the temperature dependent Raman spectroscopy, the authors discuss broadening of the CH stretching peak indicates increased disorder. But it is not clear if that is from the Tempo portion of the molecule or the methyl group at the chiral center. Ideally these peaks could be fit and show the broadening through the FWHM, though I recognize the difficulties with the noisy high temperature data. Analyzing C-C stretching from other parts of the spectra may be useful in deconvoluting these possibilities. Along with that, putting the full spectra at least in the SI would be important to see if any other aspects of the molecule change.

5. The authors suggest that this order-disorder effect in the TEMPO portion of the molecule is causing this phase transition, but I am not sure if it is the driving force or a result of increased volume in the new unit cell.

a. First, what's interesting here is that looking at the Raman spectra, the peak broadening is maintained even upon cooling (looking at a spectrum of the HTP trapped at ~273 K, just prior to the reverse transition would likely be helpful.). In an order-disorder transition, I would expect that disorder to be reduced upon cooling even if the reverse transition hasn't occurred yet.

b. Secondly, if the transition is based on the increasing disorder of the TEMPO motif, why would there be no transition in the racemic crystal? Looking at the structure, it looks like there may be some disorder in the TEMPO motif at room temperature suggesting the chirality/specific packing of the enantiomerically pure crystals is key to accessing this transition.

6. The dielectric constant measurement seems out of place to be discussed in figure 2 and is not well discussed. Not much is mentioned beyond confirming the transition and hysteresis which is already confirmed through both PXRD and Raman. Would this fit better with the optical properties? Is there anything else related to the ferroelectricity that is important from this data?

7. In Figure 3a, the authors present a comparison of the packing between the R and S crystals of the fluorinated molecule. But the rest of the figure is focused on the structural change. It would be much clearer to have a comparison of the LTP and HTP here instead (or in addition to) for an overview of what's shown in the rest of the figure.

8. Does the HTP change the ferroelectric behavior? Is this a order- disorder transition that switches from ferroelectric or paraelectric states?

Minor points:

- Curie temperature is listed in fig 1 as an important aspect that makes these molecules better but isn't discussed in the text.

- Could the Authors comment on the disorder observed in the SCXRD? Is it fully rotating, or in two distinct states or vibrating back and forth?

- Discussion of DSC jumps back and forth between interactions and the DSC data along with I suspect the simulation results discussed later which is confusing.

- More details of how the N values in the DSC discussion were obtained would be nice.

- Figure 2C it appears that 298/243 are swapped.

- When discussing hysteresis, listing all of the molecules is probably not helpful and listing the names

makes it difficult to read.

- The way the jumping crystal effect/optical properties/magnetic/ferroelectric properties sections are presented is confusing. Having so many disparate parts in a single figure makes it difficult to follow.
- The simulated structure transition should probably be discussed before the optical properties as this seems to add confirmation for the martensitic behavior.

Reviewer #4 (Remarks to the Author):

The molecular system developed by Zhang et al. is a highly interesting multi-functional system that exhibits unique martensitic transition and thermosalient effects as well as ferroelectricity. Additionally, the system displays properties such as photochromism and paramagnetism. All these properties are characterized through the adequate set of experiments. By reading the manuscript, this reviewer was fascinated by the material and could imagine its potential for unprecedented applications. However, I believe that the manuscript does not fully capture the appeal of the material system, considering its great potential. This is particularly because the research fails to establish an intriguing connection between martensitic transition behavior and ferroelectricity. In addition, there are several points that should be revised: 1) the paper includes so many properties of the system that are somewhat unrelated to the key findings (martensitic transition and ferroelectricity), 2) when the authors are describing their experimental results, there are several logical leaps, and 3) there are so many typographical errors. Despite these issues, the material system presented in this paper is still novel and very interesting, therefore, it seems reasonable to offer the authors an opportunity to revise their manuscript.

1. This system exhibits an enormous hysteresis associated with martensitic transition, which can be considered as the biggest advantage. This factor enables easy analysis of characteristics of the low-temperature phase (LTP) and high-temperature phase (HTP) at room temperature. This implies the possibility of forming one more state through phase transition and can be considered as a switch concept of a memory device. Therefore, this reviewer suggests that the authors explore the possibility of associating this characteristic with ferroelectricity.
2. In addition to the comment above, this reviewer recommends that the authors will associate the switch concept with not only ferroelectricity but also other properties of the material system.
3. The peculiar hysteresis behavior and bistability of LTP and HTP in the system reminded me of the mechanosalient behavior of the terephthalic acid crystal studied by the Naumov group (J. Am. Chem. Soc. 2016, 138, 40, 13298).
4. Due to attempting to cover many topics, the Introduction section appears to be scattered and somewhat not well-organized. In particular, while attempting to introduce multi-ferroic and TEMPO in the first paragraph, the connection between them seems disjointed or not smooth. Such an issue is present throughout the entirety of the Introduction section. Therefore, there is a need to revise the Introduction section to make it more organized and concise.
5. There are various contents that seem to deviate from the main discoveries - the martensitic transition and ferroelectricity, which can disrupt the main storyline. This issue is particularly noticeable in the sections on the thermosalient effect and optical properties. While the thermosalient effect is an important clue in proving the martensitic nature of the transition, I feel the separate section of thermosalient effect hinders the readers to follow the overall storyline. Therefore, it might be better to briefly mention the thermosalient effect in the section where structural analysis is performed to support the martensitic nature, and then move the entire section to the supplementary information.

Similarly, except for SHG which supports polar structure of the materials, most of the results in the optical properties section do not strongly support the main argument and may hinder the readers from following the paper's main storyline. Therefore, it might be worth considering moving the section to the supplementary information.

6. (Page 9, Line 177 ~ Page 10, Line 181)

The temperature-dependent Raman study described by the authors lacks supporting evidence. There is no clear basis for the assignment of the peak at around 3000 cm^{-1} . Also, although the authors claim a connection between the peak broadening and TEMPO's order-disorder states, their results do not provide sufficient evidence to support this claim. Although the authors confirmed the disordering of TEMPO in the structural analysis section later on, it should be noted that readers may not be aware of this information. Therefore, it is needed to reconsider the structure of the manuscript.

7. It appears difficult to support the absence of a phase transition in parent counterparts solely based on the presence or absence of C-H...F interactions. Can you provide additional reasons for this? Also, in the transition of S-/R-F crystal, the authors pointed out that disappearance of this interaction causes disorder in TEMPO, but I believe that the opposite would make more sense.

8. (Page 12, Line 225-227)

It is not sufficient to support the enormous window of bistable states based solely on cohesive energy, especially considering only one neighboring molecule. As the authors already know, this is related to Gibbs free energy and activation energy barrier between the LTP and HTP, which is described by the authors in the last section. Thus, readers may not accept this conclusion while reading this part. Therefore, it would be helpful to indicate that a more detailed description will be provided through DFT study later in the manuscript.

9. (Page 11 Line 216-219)

The change in lattice parameters is remarkably surprising. Typically, for the occurrence of martensitic transitions, there should be an invariant plane. Are there any clues regarding this? Also, was there any evidence of interface motion or phase front sweeping associated with the phase transition?

10. Because the materials are new, the chemical structure should be characterized by NMR, EA, and MS.

11. Thin films are used for characterizing the ferroelectricity of the materials. However, the structure or the phase of the films is not characterized yet.

12. (page 3, Line 29) Is 'shape memory a ferroelectric' a typo?

13. (page 9, Line 175) I think broadened or similar word is missing in 'since the bandwidths in Raman spectra arises from...'

14. (page9, Line 177-178) do you mean molecule not molecular in the sentence 'One is that the variation of Raman bands under 200 cm^{-1} could be caused by the external modes of the fluorinated molecular.'?

15. (Line 189) crystalized is a typo.

16. (Page 12, Line 222-224) Fig. 3C should be changed to Fig. 3b.

17. all the abbreviations should be defined when the terminologies first appear. For instance, DFT in page 7, line 126 is not defined before.

18. There are many typos throughout the manuscript. The authors are encouraged to correct them

diligently.

Response to Reviewer 1

Comments:

In this manuscript, the authors report the first example of an organic ferroelectric compound that undergoes a martensitic transition accompanied by the very rare “thermosalient effect” with an unexpectedly large thermal hysteresis. They use a clever approach of asymmetric chemical substitution of a robust chemical core with two different substituents, one of which is chiral, while the other is a stable radical and acts as an unpaired electron-bearer. The material reported here is unique because it combines multiple physical facets, including a thermosalient effect, which is very rare and cannot be predict or intentionally design at the current state of knowledge. Given the special properties of the materials, I strongly recommend acceptance of this manuscript in Nature Communications. Having said that, after including all corrections, the authors must have their manuscript read by a native speaker. The poor language is distracting from the scientific content and makes the reading very difficult. Below listed are scientific and technical comments: I would suggest that the revised and substantially improved manuscript be re-reviewed before acceptance.

Response: We sincerely thank the reviewer for spending his/her valuable time on carefully reviewing our manuscript, the in-depth comments, as well as his/her compliments on the impact and significance of our work. We have addressed all problems and made substantial modifications to our manuscript. The Tables and Figures shown in the revised manuscript and Supplementary Information are marked with purple, and the response are in blue color.

(1) The reported materials undergo structural phase transition at high temperature (399 K). This is rather high temperature, and not very compatible with the expected applications, which require transition around room temperature or slightly above and below. Can the authors comment on the advantages in comparison with materials that undergo ferroelectric transition close to room temperature (cite and compare with: Nat.

Commun. (2022) 13, 2823)?

Response: We thank the reviewer for the valuable comments and suggestions. It is known that ferroelectrics with high phase transition temperature (T_c) are indispensable for practical applications (*Adv. Mater.* 2019, 31, 1902163). First, as ferroelectric materials, *S*- and *R*-F possess high T_c of 399 K and have ferroelectric behaviors in a broad temperature range. This makes them have a wide working temperature window for ferroelectricity and be attractive for various applications. For thermosalient materials, we do agree with the reviewer that most application scenarios in smart devices for daily use need the phase transition to occur around room temperature (Figure R1). It should be noted that Naumov *et al.* reported an organic ferroelectric material guanidinium nitrate (GN) with an exceedingly large stroke that can reversibly convert energy into work around room temperature (*Nat. Commun.* 2022, 13, 2823). This ferroelectric crystal exerts a linear stroke of 51%, which is the highest value observed with a reversible operation of an organic single-crystal actuator. Thus, the combination of ferroelectricity and exceptional performance of actuating makes GN an excellent material for reversible energy-work conversion around room temperature and shows great potential in the applications such as light-weight capacitors, dielectrics, ferroelectric tunnel junctions, and thermistors.

Figure R1. Selected thermoslient compounds and temperatures of their thermoslient and non-thermoslient transitions (*Adv. Funct. Mater.* 2022, 32, 2112117).

However, as shown in Figure R1, there are still many thermoslient compounds with T_c above 400 K or lower than 200 K, and they have aroused wide interest. These materials can fill in the gap of applications under extreme conditions. In this work, the high T_c of 399 K enables ferroelectrics *S*- and *R*-F suitable for some extreme application scenarios, such as overheating alarm systems, power systems, and aerospace systems. In addition, the large ΔT of 132 K realize wide temperature windows of bistable ferroelectric states beyond all of the molecular ferroelectrics. Thus, *S*- and *R*-F have different ferroelectric behaviors at LTP and HTP, and both two phases can be investigated and utilized at room temperature, which is contrary to the traditional high-temperature ferroelectrics with only one ferroelectric phase at room temperature (Fig. 1). This is the biggest advantage of our work. As mentioned by reviewer 4, this feature would provoke many interesting applications, such as the switch concept of memory devices.

(2) On page 7, the thermal properties of the compounds are discussed. It is not clear

whether these were recorded from powder samples, or from crystals. It is known that for the thermosalient compounds, these two states can give very different temperatures in the DSC and sometimes the thermosalient transition is even suppressed. Please provide DSC measurements on collection of single crystals and show them in the supporting information. If there are significant differences, please comment on that in the main text.

Response: We thank the reviewer for instructive suggestions. The DSC results shown in previous version are recorded from the corresponding crystalline powder samples. Thus, according to the instructive suggestions of the reviewer, we performed the DSC measurements both on the single crystals and crystalline powder samples of *S-F*. The DSC results show that they are basically the same except that the peak intensity of the single crystal samples is stronger than that of the powder ones.

The corresponding changes in the revised version:

As expected, the fluorinated compounds showed thermal anomalies at 399 K upon heating and 267 K upon cooling in both powder and single-crystalline forms (Fig. 2a and Supplementary Fig. 7b).

Fig. 2 Phase transition of the fluorinated compounds. (a) DSC curves of *S-* and *R-F* powders.

Supplementary Fig. 7: (b) The DSC curves of S-F single crystal. The DSC results of single crystals and crystalline powder are basically the same except that the peak intensity of the single crystal samples is stronger than that of the powder ones.

(3) From the sections discussing the electrical properties sections such as J-V, P-V, and PFM phase and amplitude, it is evident that the results confirm the ferroelectricity in the crystals. However, a thorough discussion regarding the physics of domain switching and its correlation to material structure seems to be lacking, and the discussion of that aspect is somewhat ambiguous. A more detailed description of the testing configuration for measurement of J-V, especially regarding the integration and display of P-V, would greatly improve the transparency and the reproducibility of the work presented.

Response: We thank the reviewer for instructive suggestions. We have supplemented the discussion about the physics of domain switching and the relation between the domain switching and crystal in the revised manuscript.

Physics of domain switching: Typically, the polarization switching process contains the following steps (Figure R2): (1) domain nucleation, (2) forward extension, (3) lateral expansion, and (4) domain merging (*Phys. Rev. Lett.* 2014, 112, 247603; *Adv. Electron. Mater.* 2016, 2, 1600038). The domain switching process for molecular ferroelectrics has unique characteristics. During the domain switching process, when a uniform electric field is applied, some domains are easy to diffuse owing to the following aspects: (i) the thin film of the sample is usually inhomogeneous so the V_c varies in the different

regions; (ii) the energy required for domain nucleation is greater than the one for the motion of the domain walls (*J. Am. Chem. Soc.* 2017, 139, 3954–3957). As shown in Fig. 5, the domain switching process of compound *S-F* in the LTP and HTP only involves the domain wall movement instead of the domain nucleation because of its relatively high V_c . However, the investigation of domain switching dynamics is very complex and usually depends on the *in situ* electromechanical TEM measurements with high spatial and time resolution (*Nat. Nanotech.* 2011, 6, 98–102). Thus, for most molecular ferroelectrics, we prefer the demonstration of ferroelectricity through the PFM domain switching test to the study of domain switching dynamics.

Figure R2. The containing steps of the domain growth in a general polarization switching process (*Chem. Soc. Rev.* 2021, 50, 8248–8278).

Domain switching and its correlation to material structures: Structurally, the domain switching process involves the reverse or reorientation of dipole moments of the surface sample under the electric field. Interestingly, the emerging domain pattern under an external electric field can conform to the symmetry of the ferroelectric phase. Taking MDABCORbI₃ as an example, it undergoes a cubic-to-trigonal phase transition with an Aizu notation of 432F3 (*J. Am. Chem. Soc.* 2017, 139, 10897–10902). The emerging domain shape after the domain switching process is a hexagon instead of the written rectangle pattern, which indicates the growth of domains abiding by the ferroelectric

point-group symmetry of the hexagonal phase. However, for most of the ferroelectric compounds, the domain structure is random, which does not reflect the structural symmetry. In this work, the observed domain pattern displays an irregular shape and we cannot obtain the structural symmetry of compound *S*-F during the domain switching process.

We have also provided a more detailed description of the testing configuration for the measurement of *P*–*V* hysteresis loop measurement.

Ferroelectric loop measurement. The polarization–voltage loop was obtained via the double-wave method. The sample film deposited on the ITO glass was connected to the testing system via two conductive probes, with one dipped with liquid InGa alloy placed on the sample and the other attached to the ITO coating. The testing system was composed of a voltage source (Trek 609E-6), a waveform generator (Keysight 33500B), and a current meter (Keithley 6517B).

The typical *J*–*V* curve has two peaks at voltages corresponding to the switching voltage U_{switch} of the ferroelectric layer. Integration of the current over time gives the polarization curve:

$$P(U) = \frac{1}{Sv_{\text{sw}}} \int IdU$$

where v_{sw} is the voltage sweep rate and *S* is the area of the capacitor. A polarization curve obtained by integration of the *J*–*V* curve is presented in Fig. 5a. (*J. Appl. Phys.* 2009, 105, 054110).

The corresponding changes in the revised version:

Typically, the polarization switching process involves four steps: (1) domain nucleation, (2) forward extension, (3) lateral expansion, and (4) domain merging⁵¹. The switchable ferroelectric behavior was visualized on the thin films of *S*- and *R*-F, where their ferroelectric domain can be reversed and switched back with the application of opposite voltage (Figs. 5d–f and Supplementary Fig. 21). As shown in Fig. 5, the domain

switching process of compound S-F only involves the domain wall movement instead of the domain nucleation because of the relatively high coercive voltage (V_c).

PFM and P - V hysteresis loop measurement

The PFM measurement was carried out on a commercial piezoresponse force microscopy (Cypher, Asylum Research) with a high-voltage package at room temperature. Conductive Pt/Ir-coated silicon probes (EFM, Nanoworld) were used for domain imaging and polarization switching studies, with a nominal spring constant of ~ 2.8 nN/nm and a free-air resonance frequency of ~ 75 kHz. Since the amplitude of the low-frequency vertical PFM was within the noise level of the quadrant photodetector of the AFM, we performed the PFM experiments at contact resonance. The polarization-voltage loop was obtained via the double-wave method. The sample film deposited on the ITO glass was connected to the testing system via two conductive probes, with one dipped with liquid InGa alloy placed on the sample and the other attached to the ITO coating. The testing system was composed of a voltage source (Trek 609E-6), a waveform generator (Keysight 33500B), and a current meter (Keithley 6517B). The typical J - V curve has two peaks at voltages corresponding to the switching voltage U_{switch} of the ferroelectric layer. Integration of the current over time gives the polarization curve:

$$P(U) = \frac{1}{Sv_{\text{sw}}} \int IdU$$

where u_{sw} is the voltage sweep rate and S is the area of the capacitor.

4) It is also worth noting that the thin films utilized in this study are not based on crystals, and appropriate structural confirmation is lacking.

Response: We thank the reviewer for instructive suggestions. We have investigated the structure of the R-F thin film through PXRD measurements. The PXRD pattern of the thin film is consistent with the simulated one derived from the single-crystal structure at 298 K, indicating that the thin film and single crystal of compound R-F have the same structure.

The corresponding changes in the revised version:

Supplementary Fig. 18: The measured PXR D pattern of the *S-F* thin film matches well with the simulated one derived from the single-crystal structure at 298 K, indicating that the thin film and single crystal of compound *S-F* have the same structure.

Additionally, the lack of a comparison to other published works on ferroelectricity of organic materials (see some references suggested in specific comments below) makes it very difficult to gauge the ferroelectric performance of this compound compared to its counterparts. Please include comparison and explain why this material is better or worse than the others ferroelectric and/or thermosolient materials.

Response: We thank the reviewer for reminding us for that. To our knowledge, organic radical ferroelectric has rarely been found to date (*Angew. Chem. Int. Ed.* 2021, 60, 16668; *Ferroelectrics* 1992, 125, 11–16; *Phys. Stat. Sol. b* 1994, 184, 471–482; *Rev. Roum. Phys.* 1988, 33, 375–380; *Physica B* 2000, 289, 607–611; *Inorg. Chem.* 2023, 62, 14, 5543–5552; *Ferroelectrics* 2005, 323, 151–156; *Sci. Rep.* 2016, 6, 19682), let alone the single-component counterpart. In 1973, the first organic single-component radical ferroelectric TEMPO has been reported with ferroelectricity and ferroelasticity, but this compound has a limited working-temperature range of a relatively low T_c of 287 K and a low melting point of 311 K. These poor features severely hinder its applications. Half a century later, we synthesized a pair of high-temperature organic single-component radical ferroelectrics *S-* and *R-F*. This work presents the first high-temperature organic

single-component radical ferroelectrics, offering guidance for designing organic radical ferroelectric compounds with excellent physical properties.

Following the reviewer's suggestion, we have made a detailed summary and comparison of some physical properties (such as P_s , T_c , and ΔT) of the reported organic single-component ferroelectrics, as shown in Supplementary Table 1. For practical applications, high T_c and large P_s are critical to the performance of ferroelectric materials. In our work, compounds *S*- and *R*-F show high T_c of 399 K, larger than most of the organic ferroelectrics (such as nopinic acid and 1,3,5-trimethylnitrobenzene), but still lower than some counterparts (such as 2-amino-2',4,4',6,6'-pentafluoroazobenzene and 2-(*p*-tolyl)-1*H*-phenanthro[9,10-*d*]imidazole). Such a high T_c could be attributed to the introduction of F atoms where the increase in the energy barrier of the molecular motion. Besides, most of the organic single-component ferroelectrics have relatively low P_s values ($< 10 \mu\text{C}/\text{cm}^2$) except that croconic acid has a large P_s value of $21 \mu\text{C}/\text{cm}^2$. For comparison, although fluorination enhances ferroelectricity, the fluorinated compounds *S*- and *R*-F still possess relatively small P_s values.

Supplementary Table 1: Comparison of P_s , T_c , and ΔT of the reported organic single-component ferroelectrics.

Compound or acronym in the original publication	P_s ($\mu\text{C}/\text{cm}^2$)	T_c (K)	ΔT (K)	References
S- and R-F	~0.44	399	132	This work
2-methylbenzimidazole	5–7	-	-	Crystals 2021, 11, 1278
N -salicylidene-2,3,4,5,6-pentafluoroaniline	0.84	-	-	Adv. Sci. 2021, 8, 2102614
(S) - N -3,5-di- tert -butylsalicylidene-1–4-bromophenylethylamine	-	242	5	J. Am. Chem. Soc. 2021, 143, 21685–21693
		336	4	
salicylideneaniline	1.95	-	-	Chem. Eur. J. 2021, 27, 14831
3,4,5-trifluoro- N -(3,5-di- tert -butylsalicylidene)aniline	0.87	-	-	J. Am. Chem. Soc. 2021, 143, 13816–13823
2-(hydroxymethyl)-2-nitro-1,3-propanediol	8.4	345 362	15 27	J. Am. Chem. Soc. 2020, 142, 13989–13995
nopinic acid		161	4	Appl. Mater. Today 2020, 20, 100687

2-(p -tolyl)-1 H -phenanthro[9,10- d]imidazole	3	521	-	Chem. Commun. 2019, 55, 9610–9613
(R)-3-quinuclidinol (S)-3-quinuclidinol	6.96(R) 6.72(S)	400 (R) 398 (S)	-	Proc. Natl. Acad. Sci. U. S. A. 2019, 116, 13, 5878–5885
4-(4-(methylthio)phenyl)-2,6-di(1 H -pyrazol-1-yl)pyridine	0.715	-	-	J. Mater. Chem. C 2018, 6, 9330–9335
6TP	-	301 505	-	ChemistrySelect 2018, 3, 10608
5,6-dichloro-2-methylbenzimidazole	-	399	~30	Nat. Commun. 2012, 3, 1308
croconic acid	21	-	-	Nature 2010, 463, 789–792
2-phenylmalondialdehyde	9	363	-	Adv. Mater. 2011, 23, 2098–2103
3-HPLN	3	-	-	
CBDC	2.9	400	-	
1,3,5-trimethylnitrobenzene	2.0	152 161	-	CrystEngComm 2021, 23, 4005–4012
β -sitosteryl 4-iodocinnamate	4	342	14	Nat. Commun. 2022, 13, 6150
tetrakis(4-fluorophenylethynyl)silane	~0.1	475	-	JACS Au 2023, 3, 2, 603–609
(S,S)-4,4'-(3,3,4,4,5,5-hexafluorocyclopent-1-ene-1,2-diyl)bis[5-methyl- N -(1-phenylethyl)thiophene-2-carboxamide]	1.49	-	-	J. Am. Chem. Soc. 2022, 144, 19, 8633–8640
ortho -I-OA	3.43	-	-	Adv. Sci. 2022, 9, 2201702
(7 aR ,10 R ,11 aS)-12,12-dimethyl-6,6-dioxo-3,4,9,10-tetrahydro-7 H -7 a ,10-methano-2 H -1,3-oxazino[2,3- i][2,1]benzothiazol-11(8 H)-one	2.2	460	-	Chem. Commun. 2022, 58, 10361–10364
(R,R)-(E,E)- 1 (S,S)-(E,E)- 1	0.16 0.17	343	40	Angew. Chem. Int. Ed. 2022, 61, e202200135
2-amino-2',4,4',6,6'-pentafluoroazobenzene	1.83	433	-	Chem. Sci. 2022, 13, 4936–4943

Notably, compounds *S*- and *R*-F have large thermal hysteresis of 132 K exceeding all of the molecular ferroelectrics. This large hysteresis endows the fluorinated compounds with wide bistability beyond all of the molecular ferroelectrics

(Supplementary Tables 1 and 2). For most high-temperature ferroelectric compounds, their high-temperature ferroelectric or paraelectric states can only be obtained at high temperature. However, for martensitic ferroelectrics *S*- and *R*-F, their physical properties especially ferroelectricity of HTP can be retained at room temperature. As suggested by reviewer 4, we have confirmed the ferroelectricity of HTP by the P - V hysteresis loop and PFM measurements (Supplementary Fig. 20 and Fig. 5). Therefore, both the high- and low-temperature ferroelectric states for *S*- and *R*-F can be investigated and utilized at room temperature, which is seldom found in other high-temperature ferroelectric materials (Fig. 1). The feature of bistable ferroelectric state at room temperature makes *S* and *R*-F a new type of functional material with great application potential in next-generation smart memory devices. Meanwhile, ferroelectric compounds *S*- and *R*-F undergo martensitic phase transition accompanied by thermosalient effect, which could show larger actuation by an order of magnitude in comparison with the conventional converse piezoelectric effect. Thus, the actuation of *S*- and *R*-F can realize the modulation between the millimeter-scale (thermodynamic control) and the picometer-scale movement (electric control). This rare feature would satisfy the specific application scenarios for actuation.

Fig. 1 Comparison between traditional high-temperature ferroelectrics and *S*-/*R*-F. (a) Traditional high-temperature ferroelectrics show only one ferroelectric phase at

room temperature. (b) The combination of martensitic phase transition and ferroelectricity in *S*-/*R*-F brings forth bistability with two different ferroelectric phases at room temperature. LTP and HTP represent low-temperature phase and high-temperature phase respectively. RT FE, RT FE1, and RT FE2 represent different ferroelectric phases at room temperature.

Supplementary Table 2: Molecular ferroelectrics with large thermal hysteresis.

Compound or acronym in the original publication	ΔT	References
diisopropylammonium bromide	8 K	Science 2013, 339, 425–428
D -chiro-inositol-SiMe ₃	15 K	Angew. Chem. Int. Ed. 2022, 61, e202210809
TMCM-MnCl ₃	~18 K	Science 2017, 357, 306–309
[3-oxoquinuclidinium]ClO ₄	35 K	J. Am. Chem. Soc. 2019, 141, 1781–1787
MDABCO-NH ₄ I ₃	~56 K	Science 2018, 361, 151–155
[3.2.1-dabco]BF ₄	70 K	J. Am. Chem. Soc. 2020, 142, 1995–2000

Fig. 5 Ferroelectricity of *S*-F. (a) The J - V (dotted) and P - V (solid) curves of *S*-F showing a typical ferroelectric hysteresis loop. Vertical PFM phase (b) and amplitude

(c) images of *S*-F overlaid on the 3D topographic image. (d–f) The electrical switching of the ferroelectric domain at LTP by applying ± 150 V voltage on the white dot for 20 s: (d) pristine, (e) after applying a positive bias voltage, and (f) after applying a negative bias voltage. (g–i) The electrical switching of the ferroelectric domain at HTP by applying ± 120 V voltage in the white boxes: (g) pristine, (h) after applying a positive bias voltage, and (i) after applying a negative bias voltage.

Supplementary Fig. 20: The J - V (blue) and P - V (red) curves of *S*-F at HTP measured at room temperature.

The corresponding changes in the revised version:

To our knowledge, this large thermal hysteresis of *S*-/*R*-F is extremely rare in molecular ferroelectrics whose ΔT do not exceed 70 K ³⁹ (Supplementary Tables 1 and 2).

This martensitic phase transition brings forth a large ΔT crossing the high-temperature and low-temperature regions and thus realizes the wide bistability beyond all of the molecular ferroelectrics (Supplementary Tables 1 and 2).

(5) The article contains six composite images. I feel that some can be combined to decrease the total number of images in the main text, or shifted to the supplementary material. The quality of all figures should be improved.

Response: We thank the reviewer for instructive suggestions. According to the reviewer's suggestion, we have reduced the number of figures in the main text and improved their resolution.

Specific comments:

Response: We thank the reviewer for instructive suggestions. We have carefully revised our manuscript according to the reviewer's comments, and point-by-point replies to the comments listed as follows.

Abstract:

- “fluoridated” should be “fluorinated”
- Replace “experience reversible” with “exhibit reversible”
- I would suggest to tone down the expressions such as “ultra-large thermal hysteresis” to “large thermal hysteresis”

The corresponding changes in the revised version:

Upon H/F substitution, the fluorinated compounds exhibit reversible ferroelectric and martensitic transitions at 399 K accompanied by a large thermal hysteresis (ΔT) of 132 K.

- Change “new type of martensitic ferroelectric compounds” to “rare martensitic ferroelectric compounds”

The corresponding changes in the revised version:

The rare combination of martensitic phase transition and ferroelectricity realizes the bistability with two different ferroelectric phases at room temperature. Our finding provides insight into the exploration of martensitic ferroelectric compounds with potential applications in switchable memory devices, soft robotics, and smart actuators.

- The first sentence of the introduction starts with both “shape memory” and “a ferroelectric”, and it should be corrected.

The corresponding changes in the revised version:

Ferroelectric is a type of important functional material whose ferroelectric state can be modulated under various external stimuli including electric, light, magnetic, and thermal fields.

- The phrases in the sentences “bio-friendliness, high energy efficiency and easy portability” does not read well. It should be modified to “biocompatibility, high energy conversion efficiency, and light weight” or similar.

The corresponding changes in the revised version:

In pursuit of biocompatibility, high energy conversion efficiency, and light weight, organic multi-functional ferroelectric materials are highly desirable for smart medicine and microelectronics.

- The first report of shape memory in organic crystal should be cited clearly: JACS, 2016, 138, 13298—13306, as well as a follow-up detailed study of the phenomenon: Nat. Commun., 2019, 10(1), 3723.

The corresponding changes in the revised version:

During the martensitic phase transition, many intriguing phenomena would occur including the shape memory effect^{18,19}, superelasticity²⁰, negative thermal expansion²¹, and thermosalient effect^{22, 23, 24, 25}.

18. Karothu DP, Weston J, Desta IT, Naumov P. Shape-Memory and Self-Healing Effects in Mechanosalient Molecular Crystals. *J. Am. Chem. Soc.* **138**, 13298–13306 (2016).
19. Ahmed E, Karothu DP, Warren M, Naumov P. Shape-memory effects in molecular crystals. *Nat. Commun.* **10**, 3723 (2019).

- Replace “The introducing of” with “The introduction of”

Response: According to the suggestion by the reviewers, we have substantially modified the manuscript. In the revised version, this sentence has been deleted.

- Replace “is absent of” with “is devoid of”

The corresponding changes in the revised version:

However, this compound is devoid of structural phase transition, and thereby its ferroelectric state cannot be thermally modulated.

- Page 4, line 57: relevant references should be added to support this statement. Here are some representative works that provide in-depth studies of these phenomena: thermosalient effect: *Matter*, 2019, 1, 1033; *JACS*, 2019, 141, 3371; photosalient effect: *Crystal Growth and Design*, 2015, 15, 1983–1990; negative thermal expansion due to thermosalient effect: *Sci. Rep.*, 2016, 6, 29610.

The corresponding changes in the revised version:

During the martensitic phase transition, many intriguing phenomena would occur including the shape memory effect^{18,19}, superelasticity²⁰, negative thermal expansion²¹, and thermosalient effect^{22, 23, 24, 25}.

21. Panda MK, Centore R, Causa M, Tuzi A, Borbone F, Naumov P. Strong and Anomalous Thermal Expansion Precedes the Thermosalient Effect in Dynamic Molecular Crystals. *Sci. Rep.* **6**, 29610 (2016).
23. Commins P, *et al.* Structure–Reactivity Correlations and Mechanistic Understanding of the Photorearrangement and Photosalient Effect of α -Santonin and Its Derivatives in Solutions, Crystals, and Nanocrystalline Suspensions. *Cryst. Growth Des.* **15**, 1983–1990 (2015).
24. Colin-Molina A, *et al.* Thermosalient Amphidynamic Molecular Machines: Motion at the Molecular and Macroscopic Scales. *Matter* **1**, 1033–1046 (2019).

- I was not able to find justification or support for the assignment of the several Raman bands on the bottom pf page 9. Are these assignments reliable? Please comment or remove this discussion if the assignments are empirical.

Response: We thank the reviewer for instructive suggestions. Considering the overall comments of reviewers 1, 3 and 4, we have remeasured the Raman spectra and tried to address the problems. We mainly intended to prove the occurrence of structural phase transitions through the measurement of temperature-dependent Raman spectra. As suggested by the reviewers, to highlight the main discovery of our work, we have finally deleted the discussion of Raman spectra since phase transition has been proved by many physical measurements such as DSC, dielectric constant and PXRD.

Assignment of Raman bands: As shown in Figure R3A, there are some strong Raman bands below 200 cm^{-1} and in the range of $1400\text{--}1800\text{ cm}^{-1}$. The bands below 200 cm^{-1} could be attributed to the lattice vibration modes of the fluorinated molecules (*J. Phys.: Condens. Matter* 2013, 25, 125404; *J. Mol. Struct.* 1994, 325, 71–75). According to the cited references and the calculated Raman spectra by using B3LYP/6–31G(d) method with Gaussian 16 software, the strong bands located at 1422 cm^{-1} and 1605 cm^{-1} could be mainly assigned to the bending vibration of C–H bonds and the stretching vibration of C=C bonds, respectively (Figure R3B–D, *Chem. Sci.* 2017, 8, 1927–1935 and *J. Phys. Chem. A* 2018, 122, 44, 8738–8744). It should be noted that the calculated Raman spectra show a slight peak shift compared with the measured counterparts. Such a misfit can be attributed to the different molecular configurations under experiment and DFT calculation since the calculations of Raman spectra are based on the geometry preoptimization under the corresponding B3LYP/6–31G(d) level. (*Proc. Natl. Acad. Sci. U. S. A.* 2019, 116, 5878–5885)

Investigation of specific motion related to TEMPO portion: According to the calculated Raman spectra, the bands located at around 3000 cm^{-1} can be assigned to the C–H stretching vibration, and they broadened/narrowed as the temperature rose/dropped. As suggested by reviewers 3 and 4, the broadening of the C–H stretching peak indicated the increase in disorder. However, it is unclear whether this disorder comes from the TEMPO portion of the molecule or the methyl group at the chiral center. We do agree with the reviewers' suggestion, and think it is improper to analyze this order-disorder transition related to the TEMPO component by the C–H stretching peak. Thus, we have

ultimately deleted the discussion of this part in the revised manuscript.

Proof of structural phase transition: When the temperature rose/dropped to T_c , the apparent variations of Raman bands around 1422 and 1605 cm^{-1} were observed (Figure R3E–F), indicating the occurrence of this reversible structural phase transition.

Figure R3: (A) The temperature-dependent Raman spectra of LTP and HTP for *R-F* in the range of 45–3500 cm^{-1} . (B) The calculated Raman spectra of *R-F* molecule by using B3LYP/6–31G(d) method with Gaussian 16 software. In the calculated Raman spectra, the schematic of the vibration mode of C–H (C) and C=C (D) corresponding to the peak at 1422 cm^{-1} and 1650 cm^{-1} , respectively. (E) Temperature dependence of the Raman spectra in the frequency range of 1350–1650 cm^{-1} . (F) Temperature dependence of full width at half maximum (FWHM) of the modes in the 1422 and 1605 cm^{-1} spectral

range.

- Delete the phrase “To end this”. It has a different meaning from the use intended in this sentence.

Response: According to the suggestion by the reviewer, this sentence has been deleted.

- Replace “with unique superiority of” with “with advantageous”

The corresponding changes in the revised version:

1,4,5,8-naphthalenediimide (NDI) derivatives, as a type of electron-deficient planar aromatic compound, provide a versatile platform for the design of martensitic ferroelectrics.

- Replace “for chemically constructing” with “to chemically construct”

The corresponding changes in the revised version:

Notably, NDI has more than two modifiable sites, and thereby it is suitable to chemically construct asymmetric structures.

- Replace “to assemble a” with “to introduce a”

The corresponding changes in the revised version:

Firstly, to induce ferroelectricity, we decided to introduce a homochiral α -methyl benzylamine to one side of NDI according to the design strategy of introducing homochirality.

- Replace “the hit rate” with “the probability”

The corresponding changes in the revised version:

This strategy can largely increase the probability of one compound crystallizing in one

of five chiral-polar point groups.

- Replace “ultra-large” and other terms containing the prefix “ultra” across the main text, the figures and the supplementary materials. Use “large” instead, it sounds more scientific.

Response: According to the reviewer’s suggestion, we have deleted the prefix “ultra” across the main text.

The corresponding changes in the revised version:

Upon H/F substitution, the fluorinated compounds exhibit reversible ferroelectric and martensitic transitions at 399 K accompanied by a large thermal hysteresis (ΔT) of 132 K. This large ΔT with two competing (meta)-stable phases is further confirmed by density functional theory calculations.

Such a large ΔT could be attributed to the martensitic characteristic, where large shear strain would occur by lattice deformation during the transition.

- Replace “ultra-wide bistability” with “bistability over large temperature range”. Remove the prefix “ultra” throughout the text.

Response: According to the reviewers’ suggestion, we have made substantial modifications to our manuscript. We have deleted the prefix “ultra” throughout the main text and changed the description of bistability.

The corresponding changes in the revised version:

We found that the HTP of the fluorinated compounds remained stable when the temperature dropped to room temperature, resulting in the bistability over large temperature range.

These features realize wide temperature windows of bistable ferroelectric states beyond all of the molecular ferroelectrics. Contrary to the traditional high-temperature

ferroelectrics with only one ferroelectric phase at room temperature^{12, 13}, *S*- and *R*-F show bistability with two different ferroelectric phases, thereby both of the low- and high-temperature ferroelectric phases can be investigated and utilized at room temperature (Fig. 1).

This martensitic phase transition brings forth a large ΔT crossing the high-temperature and low-temperature regions and thus realizes the wide bistability beyond all of the molecular ferroelectrics (Supplementary Tables 1 and 2).

- The quality of the graphics in Figure 1 should be improved substantially.

Response: According to the reviewer's suggestion, we have improved the resolution of Fig. 1 and moved it to the Supplementary Information.

The corresponding changes in the revised version:

Supplementary Fig. 1: (A) Ferroelectrics with TEMPO radicals. (B) Chemical design strategy. (C) Effects of H/F substitution in NDI derivatives.

- The chemical names of the compounds should be given in the caption, and the acronyms should be given on the figure to facilitate the identification of the chemical structures.

Response: We have added the chemical names of the compounds as well as their acronyms in the captions and figures in the revised manuscript.

The corresponding changes in the revised version:

Fig. 2 Phase transition of the fluorinated compounds. (a) DSC curves of *S*- and *R*-F powders. (b) Temperature-dependent PXRD results of *R*-F. (c) Temperature-dependent dielectric constant curves of *S*-F. (d) Optical images of the “jumping crystal” effect in *R*-F crystals upon heating.

Fig. 5 Ferroelectricity of *S*-F. (a) The J - V (dotted) and P - V (solid) curves of *S*-F showing a typical ferroelectric hysteresis loop. Vertical PFM phase (b) and amplitude

(c) images of *S*-F overlaid on the 3D topographic image. (d–f) The electrical switching of the ferroelectric domain at LTP by applying ± 150 V voltage on the white dot for 20 s: (d) pristine, (e) after applying a positive bias voltage, and (f) after applying a negative bias voltage. (g–i) The electrical switching of the ferroelectric domain at HTP by applying ± 120 V voltage in the white boxes: (g) pristine, (h) after applying a positive bias voltage, and (i) after applying a negative bias voltage.

- The sentence “the fluoridated compounds undergo a high-temperature structural phase transition at 399 K with an ultra-large thermal hysteresis (ΔT) of 132 K.” states “compounds”, yet only one temperature is given for the phase transition and for the hysteresis. Please check whether this is one compound or multiple compounds, and correct the sentence.

The corresponding changes in the revised version:

In comparison with the parent compounds *S*-/*R*-H, the fluorinated compounds undergo high-temperature structural phase transitions at 399 K with large thermal hysteresis (ΔT) of 132 K.

Results and discussion

- Change “infrared radiation (IR) analyses” to “Infrared (IR) spectroscopy”

The corresponding changes in the revised version:

Their phase purities and high thermal stabilities (up to 557 K) were confirmed by powder X-ray diffraction (PXRD) measurements (Supplementary Fig. 2), infrared (IR) spectroscopy (Supplementary Fig. 3), and thermogravimetric analyses (TGA) (Supplementary Fig. 4).

- The phrase “isolated electron might be unstable” should be modified. It does not make sense.

Response: According to the suggestion by the reviewer, this sentence has been deleted.

- “Under the corresponding....level” should be “using the method”

The corresponding changes in the revised version:

After geometric optimization using the B3LYP/6-31G(d) method, the DFT-calculated VCD spectra are consistent with the measured one.

- Page 7, line 132: replace “are absent from” with “are devoid of”

Response: According to the suggestion by the reviewers, we have substantially modified the manuscript. In the revised version, this description has been deleted.

The corresponding changes in the revised version:

As shown in Supplementary Fig. 7, we did not observe the structural phase transitions of the parent compounds *S-/R-H* and racemic compounds *Rac-H/F*.

- Page 9, the sentence closing with “fluorinated molecular” is not complete.

Response: According to the suggestion by the reviewers, we have substantially modified the manuscript. In the revised version, this sentence has been deleted.

- Replace “units are connecting” with “units are connected”

The corresponding changes in the revised version:

The parallel molecules form a one-dimensional column along the *a*-axis, where the NDI units are connected through the weak interlayer π - π interactions (Supplementary Fig. 13).

- Figure 3D: please show the orientation of the axes of both unit cells, using different colors.

Response: As shown in the previous version, the orientation of the axes of both unit cells is the same, and thus we used only one coordinate axis to signify their orientations.

The corresponding changes in the revised version:

Fig. 3 Crystal structures of *S-F*. (c) Comparison of the cell lattices in the LTP (blue) and HTP (red).

• The caption of Figure 3 indicates “layers” but I am not able to see the layers in the figure, I can only see columns. Please revise the caption or change the figure.

Response: We have changed the caption of Fig. 3 in our revised version.

The corresponding changes in the revised version:

Fig. 3 Crystal structures of *S-F*. (e) Schematic diagrams of the distance and interaction energy between adjacent molecules at LTP and HTP. H atoms are partly omitted for clarity.

- Figure 3E is missing errors bars with standard deviations. Please also include information on whether these points were calculated based on limited number of reflections (mention how many) or on the full dataset. If it is the latter case, provide the full crystallographic information in a table in the supplementary information.

Response: Crystallographic data were collected using a Rigaku Saturn 924 diffractometer equipped with temperature control device, by using Cu K α ($\lambda = 1.54184$ Å) radiation. Data processing including empirical absorption correction, cell refinement, and data reduction was performed using the Crystal Clear software package. We have remeasured the temperature-dependent change of the three axes of the crystallographic axes of crystal *S-F* and produced Fig. 3d with error bars in the revised version. Detailed information as well as the standard deviation are listed in Supplementary Table 5. The unit cell parameters of crystal *S-F* were calculated based on the limited number of diffraction points (~300).

Fig. 3 Crystal structures of *S-F*. (d) Variation of unit cell parameters in the heating-cooling mode. Abrupt changes correspond to the phase transition.

Supplementary Table 5: Temperature-dependent change of the three axes of the crystallographic axes of *S-F*.

Temperature / K	$a / \text{Å}$	sd (a) ^a	$b / \text{Å}$	sd (b) ^a	$c / \text{Å}$	sd (c) ^a
300	5.986	0.006	15.79	0.02	14.545	0.018
330	6.008	0.006	15.74	0.02	14.487	0.015
360	6.028	0.005	15.731	0.017	14.48	0.017
390	6.097	0.006	15.86	0.02	14.33	0.02
420	6.607	0.009	16.81	0.02	12.687	0.017

450	6.65	0.02	16.88	0.05	12.79	0.04
420	6.63	0.02	16.89	0.05	12.59	0.04
390	6.53	0.03	16.96	0.07	12.66	0.05
360	6.53	0.03	16.98	0.07	12.63	0.05
330	6.46	0.04	16.89	0.1	12.86	0.07
300	6.44	0.03	17.04	0.08	12.69	0.07
270	6.15	0.02	15.86	0.06	14.21	0.04
240	6.04	0.02	15.72	0.05	14.37	0.04
270	6.1	0.02	15.79	0.05	14.39	0.04
300	6.02	0.02	15.87	0.05	14.34	0.05

^a Standard deviation

The corresponding changes in the revised version:

This transition from LTP (298 K) to HTP (418 K) is accompanied by dramatic mechanical movements of *S*-/*R*-F, which could be associated with the expansion of the unit cell along *a* by 10.4/10% and *b* by 7.2/7.4% and contraction along *c* by 12.6/12.5%, while the overall volume barely changes (Figs. 3C–D and Supplementary Table 5).

- Page 11: replace “their parent counterparts are absent of” with “their analogues are devoid of”

Response: According to the suggestion by the reviewers, we have substantially modified the manuscript. In the revised version, this description has been deleted.

The corresponding changes in the revised version:

As shown in Supplementary Fig. 7, we did not observe the structural phase transitions of the parent compounds *S*-/*R*-H and racemic compounds *Rac*-H/F.

- Page 11, bottom: specify which temperatures (in K) do these changes refer to (before and after), it is difficult to find that information without the absolute values

Response: Based on the single-crystal structures of crystals *S*- and *R*-F at 298 K and 418 K, the changes in unit cell parameters resulting from the martensitic transition have been investigated.

The corresponding changes in the revised version:

This transition from LTP (298 K) to HTP (418 K) is accompanied by dramatic mechanical movements of *S*-/*R*-F, which could be associated with the expansion of the unit cell along *a* by 10.4/10% and *b* by 7.2/7.4% and contraction along *c* by 12.6/12.5%, while the overall volume barely changes (Figs. 3C–D and Supplementary Table 5).

- Line 234: The shape memory effect is not generally observed for thermosalient phase transitions. It has been observed only in a couple of cases, and it is very rare. Please separate this sentence into two, and include the relevant citations for the shape-memory effect (JACS, 2016, 138, 13298—13306; Nat. Commun., 2019, 10(1), 3723.)

Response: According to the suggestion by the reviewer, the description of the thermosalient effect and shape memory effect has been revised, and the relevant references have been cited.

The corresponding changes in the revised version:

During the martensitic phase transition, shape memory^{18, 19}, self-actuation⁴¹ or explosion²⁹ could occur with the release of elastic energy. A common phenomenon observed in the thermally induced phase transition of organic compounds is thermosalient effect¹³.

18. Karothu DP, Weston J, Desta IT, Naumov P. Shape-Memory and Self-Healing Effects in Mechanosalient Molecular Crystals. *J. Am. Chem. Soc.* **138**, 13298–13306 (2016).

19. Ahmed E, Karothu DP, Warren M, Naumov P. Shape-memory effects in molecular crystals. *Nat. Commun.* **10**, 3723 (2019).

- More movies are needed to support the observation of the “jumping crystal” effect (thermosalient effect)

Response: We have provided the movie of the “jumping crystal” effect of the

fluorinated compounds as Supplementary Movie 1.

The corresponding changes in the revised version:

Supplementary Movie 1: The ‘jumping crystal’ phenomenon of the fluorinated compounds observed under the microscope.

- Replace “this mutual exclusion” with “this inconsistency”

The corresponding changes in the revised version:

This inconsistency might result from a forbidden transition of photons and could be understood through the optical absorption coefficient.

- “KBr-doped powder sample” is not clear. Please rewrite.

Response: We used the KBr-doped powder sample to record the solid-state UV/vis absorption, and the samples were mixed uniformly with potassium bromide at 1:400 (sample:KBr) ratio.

The corresponding changes in the revised version:

Thus, we used the KBr-doped powder sample to record the solid-state UV/Vis absorption with prolonged irradiation time, and the samples were mixed uniformly with potassium bromide at 1:400 (sample:KBr) ratio.

- Replace “to carry out the solid-state UV/vis” with “to record the solid-state UV/vis”

The corresponding changes in the revised version:

Thus, we used the KBr-doped powder sample to record the solid-state UV/Vis absorption with prolonged irradiation time, and the samples were mixed uniformly with potassium bromide at 1:400 (sample:KBr) ratio.

- Replace “observe obvious” with “observe apparent”

The corresponding changes in the revised version:

Meanwhile, we can observe apparent photochromic phenomena, where the sample color changes from pale pink to pale yellow.

- Lien 283: replace “chiral feature” with “chirality”.

Response: According to the suggestion by the reviewers, we have substantially modified the manuscript. In the revised version, this sentence has been deleted.

Line 289: explain why the exposure to UV/vis light does not change the chirality

Response: Circular dichroism (CD) measurement has been testified as a powerful technique in the structural analysis of chiral molecules (*Science* 2018, 361, 151–155). CD signal is the difference in the absorption of left-handed circularly polarized light (L-CPL) and right-handed circularly polarized light (R-CPL) and occurs when a molecule contains one or more chiral chromophores (light-absorbing groups). A CD signal can be positive or negative, depending on whether L-CPL is absorbed to a greater extent than R-CPL (CD signal positive) or to a lower extent (CD signal negative). (*Proc. Natl. Acad. Sci. U. S. A.* 2019, 116, 5878–5885) The free radical [NDI•]⁻ state can be easily produced by photoexciting at the wavelengths of 365 nm. This process is a photoinduced electron-transfer one which does not affect the absolute configuration of chiral compounds (*Chem. Eur. J.* 2014, 20, 7309–7316). Here, CD spectra of compounds *S*- and *R*-F are nearly mirror images. We have also investigated the CD spectra before and after the irradiation of 365 nm for 15 min. As expected, they do not show any obvious difference, verifying that this photoinduced electron-transfer process would not affect the chiral features of the synthesized compounds (Figure R4). As suggested by reviewers 1 and 4, we have deleted the discussion of this part which is partially unrelated to the main storyline of our work.

Figure R4. The CD spectra of *S*- and *R*-F before and after the irradiation of 365 nm for 15 min.

- Page 17: replace “domain switching ability should be investigated” with “the switching of the domains can be observed”

The corresponding changes in the revised version:

Then, as the essential properties of ferroelectric materials, domain switching ability can be observed by the switching spectroscopy PFM (SS-PFM).

Response to Reviewer 2

Comments:

In this paper, the authors report on a series of asymmetric 1,4,5,8-naphthalenediimide (NDI) derivatives. Among them, S and R-F compounds have been shown to exhibit martensitic transitions with ultra-large thermal hysteresis of 132 K and ferroelectric properties. However, there remains doubt about the authors' claim that they chemically designed materials that exhibit these properties.

Response: We would like to thank the reviewer for the valuable comments and suggestions. We also would like to thank him/her for the seriousness and carefulness in the reviewing process. We have addressed all problems and made substantial modifications to our manuscript. According to the reviewers' comments and suggestions, the manuscript is revised in a very serious and deliberate way. The Tables and Figures shown in the revised manuscript and Supplementary Information are marked with purple, and the response are in blue color.

While introducing chirality into the molecule is effective in creating a polar structure, it is not effective in creating a ferroelectric material that requires polarization inversion under an electric field. In fact, many ferroelectric materials belong centrosymmetric space groups above their phase transition temperatures, making them difficult to discuss as the general strategy.

Response: As mentioned by the reviewer that many ferroelectric materials belong to centrosymmetric space groups above their phase transition temperatures, we do admit that many ferroelectric compounds adopt centrosymmetric structures at their high-temperature paraelectric phase. However, one should not ignore that half of the possible ferroelectric phase transitions (45/88) have non-centrosymmetric paraelectric phases, let alone the 22 species of chiral-to-chiral ferroelectric phase transitions which can be rationally realized by chemical design (Table R1). Notably, Rochelle salt (potassium sodium *L*-tartrate tetrahydrate), as the first example of ferroelectric, is also a homochiral

compound that undergoes a paraelectric-to-ferroelectric phase transition with Aizu notation of 222F2 (*Phys. Rev.* 1921, 17, 475–481). The connection between homochirality and ferroelectricity started from Rochelle salt, while the importance of homochirality toward the generation and application of ferroelectricity was overlooked for nearly a century.

Table R1. The 22 species of chiral-to-chiral ferroelectric phase transitions.

Crystal System	Aizu Notation
Monoclinic	2F1
Orthorhombic	222F1; 222F2
Tetragonal	4F1; 422F1; 422F2 (s); 422F4
Trigonal	3F1; 32F1; 32F2; 32F3
Hexagonal	6F1; 622F1; 622F2 (s); 622F6
Cubic	23F1; 23F2; 23F3; 432F1; 432F2 (s); 432F4; 432F3

As indicated by the reviewer, the chemical design of ferroelectrics is an extremely difficult task, and nobody can assert one compound must have ferroelectricity before some indispensable physical measurements such as P - E hysteresis loop or PFM measurements. To be ferroelectrics, crystalline materials must crystallize in one of the 10 polar point groups: 1 (C_1), 2 (C_2), m (C_s), $mm2$ (C_{2v}), 4 (C_4), $4mm$ (C_{4v}), 3 (C_3), $3m$ (C_{3v}), 6 (C_6), and $6mm$ (C_{6v}). This is the essential condition to be ferroelectric. Among the 10 polar point groups, five enantiomorphic point groups (C_1 , C_2 , C_4 , C_3 , and C_6) can be achieved by introducing homochirality and the others (C_s , C_{2v} , C_{4v} , C_{3v} , and C_{6v}) by molecular tailoring. That is where chemists come in, fabricating ferroelectric materials via the knowledge of physics, chemistry, materials, etc. and raising practicable and efficient design theories. It should be highlighted that based on Neumann's principle, Landau's phenomenological theory and the Curie symmetry principle, molecular design principles for ferroelectrics have been proposed by Prof. Xiong from the chemical point of view. There are three chemical design approaches (phenomenological theories) that provide invaluable guidance toward the targeted design of molecular ferroelectrics, including quasi-spherical theory, introducing homochirality, and H/F substitution. (*J. Am. Chem. Soc.* 2020, 142, 15205–15218; *Acc. Chem. Res.* 2019, 52,

1928–1938; *APL Mater.* 2021, 9, 051112).

In terms of crystallography, homochiral materials must crystallize in 11 chiral point groups: 1 (C_1), 2 (C_2), 222 (D_2), 4 (C_4), 422 (D_4), 3 (C_3), 32 (D_3), 6 (C_6), 622 (D_6), 23 (T), and 432 (O). Among the 11 chiral point groups, five of them (C_1 , C_2 , C_4 , C_3 , and C_6) are also polar, which permits ferroelectricity. There is a higher probability (5/11) for homochiral materials to crystallize in one of the polar point groups in comparison to achiral materials (10/32). Thus, using a homochiral component makes the molecules easier to crystallize in the polar-chiral point group to meet the essential requirement for ferroelectricity. As mentioned by the reviewer this strategy may not be effective in realizing ferroelectric polarization switching under the electric field, but other chemical design strategies can also be involved to solve this problem. Nevertheless, several successful work of designing molecular ferroelectrics have been reported, including (*R*)- and (*S*)-3-quinuclidinol (*Proc. Natl. Acad. Sci. U. S. A.* 2019, 116, 5878–5885), (*R*)- and (*S*)-1-(4-cyclohexylammonium)ethylammonium (*Adv. Mater.* 2019, 31, 1808088), [*R*- and *S*-*N*-fluoromethyl-3-quinuclidinol]Ni(NO₂)₃ (*J. Am. Chem. Soc.* 2020, 142, 4756–4761), (*R*)- and (*S*)-3-(fluoropyrrolidinium)MnBr₃ (*Angew. Chem., Int. Ed.* 2012, 51, 6048–6050), *D*-chiro-inositol-SiMe₃ (*Angew. Chem. Int. Ed.* 2022, 61, e202210809) and (*R*)- and (*S*)-BINOL–DIPASi (*J. Am. Chem. Soc.* 2022, 144, 19559–19566). Thus, introducing homochirality has been proven to be a practical and efficient design strategy for molecular ferroelectrics.

The corresponding changes in the revised version:

First, to induce ferroelectricity, we decided to introduce a homochiral α -methyl benzylamine to one side of NDI according to the design strategy of introducing homochirality. This strategy can largely increase the probability of one compound crystallizing in one of five chiral-polar point groups^{12, 34}.

Additionally, the authors claim that they aimed for multiferroic properties by introducing a TEMPO unit, but the compounds did not show ferromagnetic or

antiferromagnetic behavior within the measured temperature.

Response: Organic radical compounds composed of an odd number of electrons generally show paramagnetism, but may exhibit magnetism in some cases. For example, galvinoxyl shows ferromagnetic interaction. Besides, Ren *et al.* reported an all-molecular radical magnetoelectric material by utilizing potassium-doped radical terphenyl ($K_{1.5}C_{18}H_{14}$) PAH with bulk anti-ferromagnetic spin coupling and ferroelectric imidazolium perchlorate ($ImClO_4$). (*Adv. Mater.* 2019, 31, 1806263) Thus, as mentioned by the reviewer, we do anticipate inducing multiferroic properties by introducing an organic radical component. However, the introduction of a radical TEMPO component only results in the paramagnetic behavior in compounds *S* and *R-F*. Following the suggestions of reviewers 2 and 4, we have changed the motivation from multiferroics to multi-functional ferroelectric materials and made substantial modifications to the Introduction section. In the revised manuscript, we have highlighted the contribution of introducing radical component is to induce multifunctionality including the hysteretic responses to external stimuli (ferroelectric, ferromagnetic, or ferroelastic behaviors). In our work, *S* and *R-F* show ferroelectric and paramagnetic properties. Paramagnetic property would show great potential in paramagnetic probes (*Acc. Chem. Res.* 2019, 52, 1675–1686; Pulsed electron-electron double resonance; Springer, 2019; EPR spectroscopy: fundamentals and methods; Wiley, 2018), cell proliferation and differentiation (*Bioengineering* 2020, 7, 153; *Nanoscale* 2020, 12, 8720–8726), paramagnetic smart materials (*J. Am. Chem. Soc.* 2019, 141, 16915–16922), magnetic refrigeration (*Nat. Mater.* 2005, 4, 450–454), and so on.

To our knowledge, organic radical ferroelectric has rarely been found to date (*Angew. Chem. Int. Ed.* 2021, 60, 16668; *Ferroelectrics* 1992, 125, 11–16; *phys. stat. sol. (b)* 1994, 184, 471–482; *Rev. Roum. Phys.* 1988, 33, 375–380; *Physica B* 2000, 289, 607–611; *Inorg. Chem.* 2023, 62, 14, 5543–5552; *Ferroelectrics* 2005, 323, 151–156; *Sci. Rep.* 2016, 6, 19682), let alone the single-component counterpart. In 1973, the first organic single-component radical ferroelectric TEMPO has been reported with

ferroelectricity and ferroelasticity, but this compound has a limited working-temperature range of a relatively low T_c of 287 K and a low melting point of 311 K. These poor features severely hinder its applications. Half a century later, we synthesized a pair of high-temperature organic single-component radical ferroelectrics *S*- and *R*-F. This work presents the first high-temperature organic single-component radical ferroelectrics, offering guidance for designing organic radical ferroelectric compounds with excellent physical properties.

The corresponding changes in the revised version:

In pursuit of biocompatibility, high energy conversion efficiency, and light weight, organic multi-functional ferroelectric materials are highly desirable for smart medicine and microelectronics^{6, 7}. Interestingly, materials with organic radicals usually exhibit hysteretic responses to external stimuli^{8, 9}. For example, Ren et al. reported a molecular radical hydrocarbon solid with the combination of ferroelectricity and magnetic spin exchange coupling¹⁰.

Furthermore, since there is no interaction between the important martensitic transition and ferroelectricity in *S* and *R*-F ferroelectric materials, these properties are considered independent, and it is difficult to consider them as new functional materials.

Response: The interaction between the important martensitic phase transition and ferroelectricity in *S* and *R*-F ferroelectric materials can be concluded as follow: 1) This martensitic phase transition brings forth a large ΔT crossing the high-temperature and low-temperature region, and thus realizes the bistability over large temperature range. For most high-temperature ferroelectric compounds, their high-temperature ferroelectric or paraelectric states can only be obtained at high temperature and they usually recover to the LTP at room temperature (Fig. 1). The study and application of their high-temperature ferroelectric or paraelectric state are difficult because of the high energy cost as well as the possible damage to crystal quality. However, for martensitic ferroelectrics *S*- and *R*-F, this wide bistability makes the physical properties especially

ferroelectricity of HTP can be retained at room temperature. As suggested by reviewer 4, we have confirmed the ferroelectricity of HTP by the $P-V$ hysteresis loop and PFM measurements (Supplementary Fig. 20 and Fig. 5). Thus, both the high- and low-temperature ferroelectric states for S - and R -F can be investigated and utilized at room temperature (Fig. 1). Accordingly, the feature of bistable ferroelectric states at room temperature makes S and R -F a new type of functional material with great application potential, which is seldom found in other ferroelectric materials. As indicated by reviewer 4, the combination of ferroelectricity and martensitic phase transition makes these materials desirable candidates for application in next-generation smart memory devices with multiple physical channels.

Fig. 1 Comparison between traditional high-temperature ferroelectrics and S -/ R -F. (a) Traditional high-temperature ferroelectrics show only one ferroelectric phase at room temperature. (b) The combination of martensitic phase transition and ferroelectricity in S -/ R -F brings forth bistability with two different ferroelectric phases at room temperature. LTP and HTP represent low-temperature phase and high-temperature phase respectively. RT FE, RT FE1, and RT FE2 represent different ferroelectric phases at room temperature.

Fig. 5 Ferroelectricity of S-F. (a) The J - V (dotted) and P - V (solid) curves of S-F showing a typical ferroelectric hysteresis loop. Vertical PFM phase (b) and amplitude (c) images of S-F overlaid on the 3D topographic image. (d-f) The electrical switching of the ferroelectric domain at LTP by applying ± 150 V voltage on the white dot for 20 s: (d) pristine, (e) after applying a positive bias voltage, and (f) after applying a negative bias voltage. (g-i) The electrical switching of the ferroelectric domain at HTP by applying ± 120 V voltage in the white boxes: (g) pristine, (h) after applying a positive bias voltage, and (i) after applying a negative bias voltage.

Supplementary Fig. 20: The J - V (blue) and P - V (red) curves of S -F at HTP measured at room temperature.

2) For S and R -F ferroelectric materials, mechanical strain can be induced by an electric field. Meanwhile, this martensitic phase transition causes a thermosalient effect, which could show larger actuation by an order of magnitude in comparison with the conventional converse piezoelectric effect. Thus, the actuation of S and R -F ferroelectric materials can realize the modulation between the millimeter-scale (thermodynamic control) and the picometer-scale movement (electric control) because of the combination of ferroelectricity and martensitic phase transition. This rare feature would satisfy the specific application scenarios for actuation, and thereby make S and R -F ferroelectric materials good candidates for lightweight actuators including sensors, artificial muscles, soft robotics, and energy harvesters.

Consequently, we think the interaction of ferroelectricity and martensitic phase transition endows S and R -F with many modulable physical properties which are seldom found in other pure ferroelectric materials or jumping crystals. These attributes make them a new type of functional materials with broad academic interests and high application values.

The corresponding changes in the revised version:

The combination of the martensitic phase transition and ferroelectricity endows S - and R -F with many advantages over the other ferroelectric and thermosalient materials. This martensitic phase transition brings forth a large ΔT crossing the high-temperature

and low-temperature regions and thus realizes the wide bistability beyond all of the molecular ferroelectrics (Supplementary Tables 1 and 2). For most high-temperature ferroelectric compounds, their high-temperature ferroelectric or paraelectric states can only be obtained at high temperature, which somewhat impede their study and application (Fig 1)¹². However, for martensitic ferroelectrics *S*- and *R*-F, their physical properties especially ferroelectricity of HTP can be retained at room temperature. Therefore, both the high- and low-temperature ferroelectric states for *S*- and *R*-F can be investigated and utilized at room temperature, which is seldom found in other high-temperature ferroelectric materials¹² (Fig. 1). The feature of bistable ferroelectric states at room temperature makes *S* and *R*-F a new type of functional material with great application potential in next-generation smart memory devices with multiple physical channels. Besides, ferroelectric compounds can induce mechanical strain under the electric field⁴². Meanwhile, this martensitic phase transition causes thermosalient effect, which could show larger actuation by an order of magnitude in comparison with the conventional converse piezoelectric effect. Thus, the actuation of *S*- and *R*-F can realize the modulation between the millimeter-scale (thermodynamic control) and the picometer-scale movement (electric control)⁵². This rare feature would satisfy specific application scenarios for actuation, thereby making *S*- and *R*-F good candidates for lightweight actuators including sensors, artificial muscles, soft robotics, and energy harvesters¹⁴.

On the other hand, this paper is based on very careful experiments and discussions, and a logical development is made. Therefore, it is recommended to submit it to another highly specialized journal.

Response: We have carried out new experiments, made substantial revisions to the manuscript and adequately addressed all the concerns proposed by the reviewers. With our tremendous efforts, we hope this revision would change reviewer 2's opinion.

Significance:

Organic multi-functional materials are highly desirable for smart medicine and

microelectronics because of their unique advantages of biocompatibility, high energy conversion efficiency, and light weight. Among them, organic radical ferroelectric materials have attracted wide attention because they usually exhibit hysteretic responses to external stimuli. For example, Ren *et al.* reported a molecular radical hydrocarbon solid with the combination of ferroelectricity and magnetic spin exchange coupling. In 1973, the first organic single-component radical ferroelectric TEMPO has been reported with ferroelectricity and ferroelasticity, but this compound has a limited working-temperature range of a relatively low T_c of 287 K and a low melting point of 311 K. These poor features severely hinder its applications. To our knowledge, organic radical ferroelectric has rarely been found to date, let alone the single-component counterpart. Thus, it is challenging to construct organic single-component radical ferroelectric compounds with high T_c and other interesting properties.

In our work, we successfully designed a pair of organic single-component ferroelectric compounds *S*- and *R*-F, exhibiting a high-temperature martensitic phase transition at 399 K with a large thermal hysteresis (ΔT) of 132 K. The combination of the martensitic phase transition and ferroelectricity brings forth a large ΔT crossing the high-temperature and low-temperature region, and thus realizes the wide bistability beyond all of the molecular ferroelectrics. Contrary to the traditional high-temperature ferroelectrics with only one ferroelectric phase at room temperature, *S*- and *R*-F show bistability with two different ferroelectric phases and thereby both of the low- and high-temperature ferroelectric phases can be investigated and utilized at room temperature, enabling the applications in next-generation smart memory devices with multiple physical channels. To our knowledge, this work presents the first high-temperature organic single-component radical ferroelectrics, offering guidance for designing organic radical ferroelectric compounds with excellent physical properties. These new ferroelectric materials combined with martensitic phase transition bring forth many interesting but rare physical properties, and will attract broad interests and wide readerships in the field of material science, chemistry, physics and electronics.

Response to Reviewer 3

Comments:

The authors present a detailed study of crystals formed from TEMPO based NDI derivatives, characterizing the phase transitions in each compound and ferroelectric properties of these systems. They discovered the enantiomerically fluorinated compound undergoes a martensitic phase transition as well as exhibit ferroelectric behavior. This is coupled with photochromism and paramagnetic behavior in the system. While certainly it is interesting to find this mix of different transitions and optoelectronic properties in rationally designed molecules, It is not clear how these properties benefit from each other with this system. As a result, I'm not sure if inspiring future materials with these properties is interesting enough to a broad audience to warrant publishing in nature communications. Additionally, the analysis and discussion seemed lacking in determining the transition mechanism. For these reasons, I must recommend against publishing this paper in Nature Communications.

Response: We sincerely thank the reviewer for spending his/her valuable time on carefully reviewing our manuscript, the in-depth comments, as well as his/her compliments on the impact and significance of our work. It is a great opportunity for us to communicate with excellent reviewers and learn from them a lot.

Highlights:

1. *S*- and *R*-F are the first high-temperature organic single-component radical ferroelectrics.
2. *S*- and *R*-F have large thermal hysteresis of 132 K exceeding all of the molecular ferroelectrics.
3. The rare combination of martensitic phase transition and ferroelectricity realizes the bistability with two different ferroelectric phases at room temperature.

The photochromism and paramagnetic behaviors mainly come from the NDI and

TEMPO components in *S*- and *R*-F, respectively. We cannot find the coupling between these properties with ferroelectricity or martensitic phase transition. SHG effect is closely related to ferroelectricity. The dependence of $\chi^{(2)}$ (second-order nonlinear coefficient) and P_s follows the Landau expression, $\chi^{(2)} = 6\epsilon_0\beta P_s$, where ϵ_0 is the vacuum dielectric constant and β is nearly independent of the temperature (*Phys. Rev. Lett.* **2011**, *107*, 147601). Notably, the coupling is found in ferroelectricity and martensitic phase transition, which is the biggest advantage of our work (Fig. 1). This coupling in *S* and *R*-F ferroelectric materials can be concluded as follow: 1) This martensitic phase transition brings forth a large ΔT crossing the high-temperature and low-temperature region, and thus realizes the wide bistability with two different ferroelectric phases at room temperature. 2) For *S* and *R*-F ferroelectric materials, mechanical strain can be induced by the electric field and thermosalient effect. Thus, the actuation of *S* and *R*-F ferroelectric materials can realize the modulation between the millimeter-scale (thermodynamic control) and the picometer-scale movement (electric control) because of the combination of ferroelectricity and martensitic phase transition.

Fig. 1 Comparison between traditional high-temperature ferroelectrics and *S*-/*R*-F. (a) Traditional high-temperature ferroelectrics show only one ferroelectric phase at room temperature. (b) The combination of martensitic phase transition and ferroelectricity in *S*-/*R*-F brings forth bistability with two different ferroelectric phases

at room temperature. LTP and HTP represent low-temperature phase and high-temperature phase respectively. RT FE, RT FE1, and RT FE2 represent different ferroelectric phases at room temperature.

These new ferroelectric materials combined with martensitic phase transition bring forth many interesting but rare physical properties. As indicated by reviewer 4, this will show great potential in the switch concept of memory devices with multiple physical channels. We have also elaborated the mechanism of this rare phase transition based on the experiment analyses and DFT calculations (details are shown below). The significance of our manuscript especially the combination of ferroelectricity and martensitic phase transition has also been highlighted in the revised manuscript. These attributes make them a new type of functional materials with broad academic interests and high application values. We think our work deserves the publication in Nature Communications since they will attract broad interests and wide readerships in the field of material science, chemistry, physics and electronics.

But with a more thorough analysis of these data and a deeper discussion of the results, I look forward to reading a revised version elsewhere.

Response: We believe that we have made substantial revisions to the manuscript and adequately addressed all the concerns proposed by the reviewer, and we hope that the revised manuscript would change reviewer 3's opinion. The Tables and Figures shown in the revised manuscript and Supplementary Information are marked with purple, and the response are in blue color.

More detailed comments are discussed below:

1. In this study, the researchers investigate the inclusion of homochirality and fluorine substitution on the NDI-TEMPO molecule. While the reasons for including homochirality was discussed, the reasons for fluorination weren't mentioned and they found the hydrogen-based system also displayed ferroelectricity as well. I think discussion of the underlying reasons for this result as well as a comparison, if any, to

the ferroelectric behavior would be important here.

Response: We greatly appreciate the reviewer's comments. Ferroelectric compounds *S*- and *R*-H do not have structural phase transitions, which indicates that their physical properties cannot be thermodynamically modulated. Thus, fluorination was applied in the hydrogen-based system to induce phase transition.

H/F substitution, resembling the isotope effect, has been proved as a practical and universal molecular design principle for ferroelectrics, and we have summarized the advantages as follows: 1) *Do not change the crystallography point group.* Replacement of H with an F atom leads to only a minor structural distortion, resulting from the similar van der Waals radii and steric hindrance between the H and F atoms. 2) *Enhance P_s .* Since F atom shows higher electronegativity, the introduction of an F atom may also enhance the strength of the dipole moment and ferroelectricity. 3) *Introducing homochirality.* Replacement of H with an F atom may sometimes introduce homochirality into the fluorinated compound, which increases the possibility of constructing molecular ferroelectric. 4) *Raise T_c .* F atoms have larger mass in comparison with the H counterpart. The increase in T_c can be explained by the increases in the potential energy barrier of the tumbling motions of molecules. 5) *Better hydrophobicity and liposolubility.* The fluorinated compounds gain better hydrophobicity and liposolubility.

In this work, compared with *S*- and *R*-H, H/F substitution maintained the crystallography point group and induced this unique martensitic phase transition in ferroelectric compounds *S*- and *R*-F (Figure R5). Besides, fluorination also enhance ferroelectricity; specifically, the fluorinated compounds have larger P_s values of 0.83 $\mu\text{C cm}^{-2}$ in comparison with that of the H counterparts (0.23 $\mu\text{C cm}^{-2}$).

Figure R5. Effects of H/F substitution in NDI derivatives.

The corresponding changes in the revised version:

H/F substitution, resembling the isotope effect, could produce new opportunities to induce phase transition without the change of the crystallography point group^{35, 36}.

H/F substitution provides new possibilities to induce phase transition through the regulation of molecular rotational energy barriers and intermolecular forces¹².

The higher electronegativity of F atoms may result in a larger dipole moment and further enhance ferroelectricity³⁵. To estimate the ferroelectric polarization of the crystal, we first calculated the vector sum of the corresponding molecular dipole moment in the unit cell (Supplementary Fig. 17). As expected, the estimated saturated polarization (P_s) value of *S*-F is about $0.83 \mu\text{C cm}^{-2}$ (along the *b*-axis), larger than that of the parent one (*S*-H, $0.23 \mu\text{C cm}^{-2}$).

2. Are crystal structures similar between the H- and F- molecules? While the R and S structures are compared, it would be nice to show a comparison between the H- and F-systems since the martensitic phase transition appears suppressed in those cases. I think it's mentioned later with respect to the ferroelectric properties but should be talked about in detail with the SCXRD.

Response: We thank the reviewer for instructive suggestions. In general, H/F substitution does not change the crystallography point group as well as the packing arrangement. Replacement of H with an F atom leads to only a minor structural distortion, resulting from the similar van der Waals radii and steric hindrance between

the H and F atoms. (*J. Am. Chem. Soc.* 2020, 142, 15205–15218; *Trends Chem.* 2021, 3, 1088–1099) As expected, according to the SCXRD results, the fluorinated compounds have similar crystal symmetry and packing arrangements with the H counterparts (Supplementary Fig. 14a and c). Differently, H/F substitution alters the crystal packing environment, namely, the intermolecular interactions. Specifically, fluorination brings new C–H···F interactions between TEMPO components and F atoms of the adjacent molecules (Supplementary Fig. 14a). Besides, the introduction of F atoms has also enhanced other intermolecular interactions such as π - π interactions between NDI planes (Supplementary Fig. 14b and d). The intensity of molecular interaction is mapped onto the Hirshfeld surface by using a red-blue-white color scheme, where the white regions exactly correspond to the distance of Van der Waals contact, the blue ones correspond to longer contacts, and the red ones represent closer contacts (Supplementary Fig. 15). According to the red regions shown in Hirshfeld surfaces and the distances between NDI layers, *S-F* has a stronger π - π stacked interaction in comparison with *S-H*. Besides, *S-F* has C–H···F interaction which does not exist in *S-H*.

To further confirm above discussion, we have appended the DFT structural relaxation for *S-R-H*. For reference, the optimized lattice constants *a*, *b*, *c* of LTP and HTP are significantly different by -5.58%, -7.43% and 13.19% for *S-R-F*, respectively, as shown in supplementary Table 6. However, the optimized lattice constants of LTP and hypothetical HTP (derived from the F-case) of *S-R-H* are very close: -0.05%, -1.75%, 1.24% for *a*, *b*, *c*, respectively. Considering the numerical precision and soft vdW packing between molecules, such tiny differences are in the tolerant range to conclude that no phase transition occurred for *S-R-H*, which is consistent with the experimental observation and can support above analysis.

Supplementary Fig. 14: (a) Packing view of *S-F* at LTP along the *a*-axis. Red dashed lines represent C–H···F interaction, which do not exist in *S-H*. (b) Schematic diagrams of the distance and interaction energy between adjacent *S-F* molecules. H atoms are partly omitted for clarity. (c) Packing view of *S-H* along the *a*-axis. (d) Schematic diagrams of the distance and interaction energy between adjacent *S-H* molecules.

Supplementary Fig. 15: The Hirshfeld d_{norm} surfaces and the 2D fingerprint plots of *S-F* at LTP (a) and *S-H* (b). The interactions between different atoms are marked in the

Figures. According to the red regions shown in Hirshfeld surfaces, *S*-F has C–H···F interactions as well as the enhanced interlayer π - π interactions in comparison with *S*-H.

The corresponding changes in the revised version:

Due to the close van der Waals radius and similar steric parameters of H and F atoms, the F-substituted compounds and the parent ones show similar crystal structures and crystallographic point groups³⁵. Their asymmetric units consist of one corresponding NDI derivatives molecule, where the chiral C atom has the “*R/S*” conformation, indicating their enantiomeric feature. The parallel molecules form a one-dimensional column along the *a*-axis, where the NDI units are connected through the weak interlayer π - π interactions (Supplementary Fig. 13). From the packing view, the NDI derivative molecules are symmetrically aligned along the 2₁ screw axes ([0 1 0] direction), resulting in the formation of spontaneous polarization as well as ferroelectricity (Fig. 3a). Differently, the introduction of F atoms changes the crystal packing environment, that is, the intermolecular interactions⁴³ (Supplementary Fig. 14). The molecular Hirshfeld surface and the related 2D-fingerprint plot calculations indicated that the fluorination brought forth C–H···F interactions and enhanced other intermolecular interactions such as π - π stacking between the NDI planes (Supplementary Fig. 15). Accordingly, the newly formed and enhanced intermolecular interactions may play an important role in inducing phase transition in the fluorinated compounds.

The optimized lattice constants are close to their experimental one, where the lattice constants *a*, *b*, *c* of LTP and HTP are significantly different by -5.58%, -7.43% and 13.19%, respectively, as compared in Supplementary Table 6. What’s more, the internal energy of HTP is indeed higher (51.8 meV/f.u.) than the LTP one, as expected. The above analysis shows that there is indeed a structural phase transition in *S*-/*R*-F. However, the optimized lattice constants of LTP and hypothetical HTP (derived from the F-case) of *S*-/*R*-H are very close: -0.05%, -1.75%, 1.24% for *a*, *b*, *c*, respectively. Considering the numerical precision and soft van der Waals (vdW) packing between molecules, such tiny differences are in the tolerant range to conclude that no phase

transition occurred for *S*-/*R*-H which is consistent with the experimental observation.

3. The Isotope effect is presented as the reason for changes to the phase transition but that seems to neglect many important aspects of H/F substitution. Even in the paper, the effects of hydrogen bonding are discussed which don't occur in the H- based molecule. Additionally, the fluorine would act as an electron withdrawing group changing the pi-stacking interactions. All of these would likely have a stronger effect on the phase transition than just the isotope effect on rotation.

Response: We greatly appreciate the reviewer's comments. Generally, isotope effect only shows great potential in phase transition inducing of hydrogen-bonded molecular ferroelectrics. H/F substitution resembles the isotope effect, but it has broader applications than the isotope effect. In this work, considering the synthesis difficulty and practicability of H/D substitution, H/F substitution has been applied to the ferroelectric compounds *S*- and *R*-H to induce phase transition by modulating the potential energy barrier of rotating motions. Thus, as mentioned by the reviewer, H/F substitution should be the main cause of phase transition inducing in ferroelectrics *S*- and *R*-F instead of the isotope effect.

The interaction between active groups and their hosting lattice is responsible for a structural change at a specific thermodynamic condition. (*The Physics of Structural Phase Transitions (2nd Edition)*, Springer Science+Business Media, Inc, 2009) F atoms have larger mass as well as electronegativity in comparison with the H counterpart. Thus, H/F substitution provides new possibilities to induce phase transition through the regulation of molecular rotational energy barriers and intermolecular forces. From the perspective of crystallography, H/F substitution alters the crystal packing environment, namely, the intermolecular interactions. Specifically, fluorination brings new C–H···F interactions between TEMPO components and F atoms of the adjacent molecules (Supplementary Fig. 14a). Besides, the introduction of F atoms has also enhanced other intermolecular interactions such as π - π interactions between NDI planes (Supplementary Fig. 14b and d). The intensity of molecular interaction is mapped onto

the Hirshfeld surface by using a red-blue-white color scheme (Supplementary Fig. 15). According to the red regions shown in Hirshfeld surfaces and the distances between NDI layers, *S-F* has a stronger π - π stacked interaction in comparison with *S-H*. Consequently, the changes of intermolecular interaction after fluorination such as the newly formed C-H \cdots F interactions and the enhanced π - π stacking could mainly result in the phase transition inducing in ferroelectrics *S-* and *R-F*.

Supplementary Fig. 14: (a) Packing view of *S-F* at LTP along the *a*-axis. Red dashed lines represent C-H \cdots F interaction, which do not exist in *S-H*. (b) Schematic diagrams of the distance and interaction energy between adjacent *S-F* molecules. H atoms are partly omitted for clarity. (c) Packing view of *S-H* along the *a*-axis. (d) Schematic diagrams of the distance and interaction energy between adjacent *S-H* molecules.

Supplementary Fig. 15: The Hirshfeld d_{norm} surfaces and the 2D fingerprint plots of *S*-F at LTP (a) and *S*-H (b). The interactions between different atoms are marked in the Figures. According to the red regions shown in Hirshfeld surfaces, *S*-F has C–H \cdots F interactions as well as the enhanced interlayer π - π interactions in comparison with *S*-H. The corresponding changes in the revised version:

As shown in Supplementary Fig. 7, we did not observe the structural phase transitions of the parent compounds *S*-/*R*-H and racemic compounds *Rac*-H/F. H/F substitution provides new possibilities to induce phase transition through the regulation of molecular rotational energy barriers and intermolecular forces¹². As expected, the fluorinated compounds showed thermal anomalies at 399 K upon heating and 267 K upon cooling in both powder and single-crystalline forms (Fig. 2a and Supplementary Fig. 7b).

Differently, the introduction of F atoms changes the crystal packing environment, that is, the intermolecular interactions⁴³ (Supplementary Fig. 14). The molecular Hirshfeld surface and the related 2D-fingerprint plot calculations indicated that the fluorination brought forth C–H \cdots F interactions and enhanced other intermolecular interactions such as π - π stacking between the NDI planes (Supplementary Fig. 15). Accordingly, the

newly formed and enhanced intermolecular interactions may play an important role in inducing phase transition in the fluorinated compounds.

4. In the temperature dependent Raman spectroscopy, the authors discuss broadening of the CH stretching peak indicates increased disorder. But it is not clear if that is from the Tempo portion of the molecule or the methyl group at the chiral center. Ideally these peaks could be fit and show the broadening through the FWHM, though I recognize the difficulties with the noisy high temperature data. Analyzing C-C stretching from other parts of the spectra may be useful in deconvoluting these possibilities. Along with that, putting the full spectra at least in the SI would be important to see if any other aspects of the molecule change.

Response: We thank the reviewer for instructive suggestions. Considering the overall comments of reviewers 1, 3 and 4, we have remeasured the Raman spectra and tried to address the problems. We have provided the full Raman spectra in the response letter (Figure R3A). We mainly intended to prove the occurrence of structural phase transitions through the measurement of temperature-dependent Raman spectra. According to the calculated Raman spectra, the bands located at around 3000 cm^{-1} can be assigned to the C–H stretching vibration, and they broadened/narrowed as the temperature rose/dropped. As suggested by reviewers 3 and 4, the broadening of the C–H stretching peak indicates the increase in disorder, while it is unclear whether this disorder comes from the TEMPO portion of the molecule or the methyl group at the chiral center. We do agree with the reviewers' suggestion, and think it is improper to analyze this order-disorder transition related to the TEMPO component by the C–H stretching peak. As suggested by reviewer 3, we have also analyzed the C–C stretching vibration bands. However, because of the relatively weak Raman intensity as well as the unapparent variation, we failed to assign the C–C stretching vibration from the TEMPO portion of the molecule. As suggested by the reviewer, we have investigated the FWHM of the bands related to the C–H and C=C stretching mode (Figure R3B–D) by making multiple peak fit, especially with the Lorentz function in the Origin 2021 program. The variation of temperature-dependent FWHM of the Raman bands indicates

the occurrence of phase transition (Figure R3E–F). As indicated by reviewer 3, considering the difficulty in analyzing data because of the noise of the high-temperature data and poor crystal quality, the temperature-dependent FWHM of the Raman bands do not ideally present the typical rectangle shape. As suggested by the reviewers, to highlight the main discovery of our work, we have finally decided to delete the discussion of Raman spectra since phase transition has been proved by many physical measurements.

Figure R3: (A) The temperature-dependent Raman spectra of LTP and HTP for *R-F* in the range of 45–3500 cm^{-1} . (B) The calculated Raman spectra of *R-F* molecule by using B3LYP/6–31G(d) method with Gaussian 16 software. In the calculated Raman spectra, the schematic of the vibration mode of C–H (C) and C=C (D) corresponding to the peak at 1422 cm^{-1} and 1650 cm^{-1} , respectively. (E) Temperature dependence of the Raman

spectra in the frequency range of 1350–1650 cm^{-1} . (F) Temperature dependence of full width at half maximum (FWHM) of the modes in the 1422 and 1605 cm^{-1} spectral range.

5. The authors suggest that this order-disorder effect in the TEMPO portion of the molecule is causing this phase transition, but I am not sure if it is the driving force or a result of increased volume in the new unit cell.

Response: We thank the reviewer for his/her valuable comments. We think the order-disorder movement in the TEMPO portion of compounds *S*- and *R*-F is the cause of this phase transition instead of the result of increased volume in the new unit cell. To address the concern of the reviewer, we have measured the single-crystal structure of *S*-F at 390 K which is close to but lower than its T_c of 399 K. As temperature increased from 298 to 390 K, the volume of the corresponding unit cell increased (Table R2). Although at 390 K (just lower than T_c) all anisotropic thermal factors especially those in the TEMPO moiety increased slightly in comparison with those at 298 K, we can still use an ordered model to solve the crystal structure (Figure R6). This indicates that TEMPO portion is one of a thermal nature, not one of disorder (*Journal of Chemical Crystallography*, 2021, 51, 71–81). When temperature rose to T_c , the thermal ellipsoid plot exhibited abnormal anisotropic changes and the atoms in the TEMPO moiety should be split (Figure R6). Therefore, we confirmed that the atoms of TEMPO shown in Figure R6 are in a disordered condition and the order-disordered phase transition happened. These confirm that the order-disorder effect in the TEMPO portion is not the result of increased volume in the new unit cell. Besides, according to the DSC results, N , the ratio of the number of possible configurations in the HTP and LTP, was deduced to be 4.3 and 5.1 for *S*- and *R*-F respectively, which confirms the occurrence of order-disorder phase transitions. Combined the above-mentioned results, the fluorinated compounds undergo phase transitions associated with the order-disorder movement of the TEMPO component.

Figure R6. Thermal ellipsoid plot (50% probability) of *S-F* at 298 K, 390 K, and 418 K.

Table R2. Crystallographic data of *S-F* at 298 K, 390 K, and 418 K.

Compound		S-F		
Formula		$C_{31}H_{29}FN_3O_5$		
Temperature	298 K	390 K	418 K	
Weight		542.57		
System		monoclinic		
Space group		$P2_1$		
a (Å)	5.98590(10)	6.1568(2)	6.6000(3)	
b (Å)	15.7126(4)	15.8315(4)	16.8407(7)	
c (Å)	14.4821(4)	14.1795(3)	12.6531(6)	
α (°)	90	90	90	
β (°)	92.349(2)	91.065(2)	92.536(4)	
γ (°)	90	90	90	
V (Å ³)	1360.96(6)	1381.86(6)	1405.00(11)	
Z	2	2	2	
R_1	0.0582	0.0443	0.0761	
wR_2	0.1713	0.1384	0.2206	
GOF	1.060	1.054	1.025	

The corresponding changes in the revised version:

As the temperature increased, the TEMPO radical units displayed a partially disordered state, exhibiting a back-and-forth vibrating motion, and thus the C–H \cdots F interactions

disappeared in the HTP (Fig. 3b).

Consequently, the order-disorder change of TEMPO units and the lattice deformation cooperatively promote the martensitic phase transition under thermodynamic drive.

a. First, what's interesting here is that looking at the Raman spectra, the peak broadening is maintained even upon cooling (looking at a spectrum of the HTP trapped at ~273 K, just prior to the reverse transition would likely be helpful.). In an order-disorder transition, I would expect that disorder to be reduced upon cooling even if the reverse transition hasn't occurred yet.

Response: We thank the reviewer for instructive suggestions. Line widths in experimental Raman spectra are a consequence of vibrational damping, which in turn arises from fluctuations in molecular conformations. The more disordered the system, the more molecular vibrations are damped. Damping shifts the mode frequency downward and broadens the peak in the Raman spectrum (*Macromolecules* 2017, 50 (24), 9773–9787). In our previous version, as indicated by reviewer 3, the peak broadening may be maintained even upon cooling because of the poor crystal quality after annealing as well as the resulting weak Raman peaks and their unapparent variation. Thus, we have remeasured the Raman spectrum of compound R-F especially the one at 273 K (Figure R3A). As suggested by the reviewer, we have investigated the FWHM of the bands related to the C–H and C=C stretching mode (Figure R3B–D) by making multiple peak fit, especially with the lorentz function in the Origin 2021 program. As expected by the reviewer, the reduction of disorder upon cooling has been found even if the reverse transition has not occurred yet (Figure R3E–F). When the temperature rose/dropped to T_c , the apparent variations of Raman bands at around 1422 and 1605 cm^{-1} were observed, indicating the occurrence of this reversible structural phase transition (Figure R3E). As indicated by reviewer 3, considering the difficulty in analyzing data because of the noise of the high-temperature data and poor quality of crystals, the temperature-dependent FWHM of the Raman bands do not ideally present the typical rectangle shape (Figure R3F). As suggested by the reviewers, to highlight

the main discovery of our work, we have finally deleted the discussion of Raman spectra since phase transition has been proved by many physical measurements such as DSC, dielectric constant and PXRD.

Figure R3: (A) The temperature-dependent Raman spectra of LTP and HTP for *R-F* in the range of 45–3500 cm^{-1} . (B) The calculated Raman spectra of *R-F* molecule by using B3LYP/6–31G(d) method with Gaussian 16 software. In the calculated Raman spectra, the schematic of the vibration mode of C–H (C) and C=C (D) corresponding to the peak at 1422 cm^{-1} and 1650 cm^{-1} , respectively. (E) Temperature dependence of the Raman spectra in the frequency range of 1350–1650 cm^{-1} . (F) Temperature dependence of full width at half maximum (FWHM) of the modes in the 1422 and 1605 cm^{-1} spectral range.

b. Secondly, if the transition is based on the increasing disorder of the TEMPO motif, why would there be no transition in the racemic crystal? Looking at the structure, it looks like there may be some disorder in the TEMPO motif at room temperature suggesting the chirality/specific packing of the enantiomerically pure crystals is key to accessing this transition.

Response: We thank the reviewer for kind suggestion. It is known that the interaction between active groups and their hosting lattice is responsible for a structural change at a specific thermodynamic condition. (*The Physics of Structural Phase Transitions (2nd Edition)*, Springer Science+Business Media, Inc, 2009) Apparently, the crystal structures of the homochiral and racemic fluorinated compounds are different (Supplementary Tables 3 and 4). The homochiral compound crystallizes in the polar-chiral space group $P2_1$ at room temperature, while the racemic one adopts the space group Pc . The difference in the crystal structure makes them have different interaction between the active groups and their hosting lattice, thereby resulting in the difference in phase transition behaviors. This should be the main reason why the phase transition of the racemic compound has not been detected. As mentioned by the reviewer, the TEMPO component in the racemic crystal is partially disordered and the molecular motion cannot be frozen at room temperature. Thus, we can rationally speculate that this compound might have a structural phase transition below -180 °C (the detection limit of our DSC measurement), while this inference cannot be proved in the current experimental condition.

6. The dielectric constant measurement seems out of place to be discussed in figure 2 and is not well discussed. Not much is mentioned beyond confirming the transition and hysteresis which is already confirmed through both PXRD and Raman. Would this fit better with the optical properties? Is there anything else related to the ferroelectricity that is important from this data?

Response: We thank the reviewer for constructive comment. As mentioned by the editor and reviewers, the combination of martensitic phase transition and

ferroelectricity is the key finding of our work. We have emphasized that this combination brings forth wide bistability beyond all of the molecular ferroelectrics. This wide bistability makes both of the low- and high-temperature ferroelectric phases can be investigated and utilized at room temperature. Such a rare bistable feature was confirmed by the dielectric measurement (Fig. 2c and Supplementary Fig. 9). Meanwhile, the dielectric measurement can also be used to confirm the occurrence of martensitic transformation.

Furthermore, temperature-dependent dielectric behavior can reflect ferroelectric feature. For proper ferroelectrics, the curve of the dielectric reciprocal value ϵ_c^{-1} as a function of temperature is linear. The Curie–Weiss constant C can be divided into a paraelectric phase constant C_{para} and a ferroelectric phase constant C_{ferro} . The order of phase transition can be assigned according to the ratio $k = C_{\text{para}}/C_{\text{ferro}}$, that is, a second-order one if k is close to 2 and a first-order one if k is close to 8. The relationship between α_0 and C obeys $\alpha_0 = 1/(\epsilon_0 C)$. (*Chem. Soc. Rev.* 2011, 40, 3577–3598) The characteristic step-like dielectric anomalies of *S*- and *R*-F indicate improper ferroelectricity, which is different from the peak-like dielectric anomalies observed in proper ferroelectrics (Fig. 2c and Supplementary Fig. 9). In the basic view of the phenomenological Landau theory of phase transitions, spontaneous polarization acts as a secondary effect in improper ferroelectric transition. This step-like dielectric behavior fulfills the requirement of bistable or switchable dielectrics, and the appearance of the loop ($\epsilon'-T$), i.e., dielectric bistability, may be attributable to the improper mechanism of spontaneous polarization. (*Adv. Mater.* 2014, 26, 4515–452; *Phys. Status Solidi* 1973, 15, 579 - 590; *Ferroelectrics* 1970, 1, 11–17; *Jpn. J. Appl. Phys.* 1976, 15, 1621) Besides, we think the dielectric constant measurement results of *S*- and *R*-F do not show an obvious relationship with their optical properties.

We thus think the dielectric measurement is very important for our storyline and should be presented in the main text.

The corresponding changes in the revised version:

Such a rare bistable feature was investigated by the dielectric measurement. As shown in Fig. 2c and Supplementary Fig. 9, the step-like anomalies enclosed a rectangular loop with a wide temperature window of 132 K, indicating improper ferroelectricity³⁸. The fluorinated compounds are in the low-dielectric states below 399 K in the heating run. As temperature rose to 399 K, the dielectric constants of *S*- and *R*-F changed from low-dielectric states to high-dielectric ones. When cooling down to room temperature, the high-dielectric states maintained because of the large thermal hysteresis resulted from the martensitic transition. Therefore, both two distinct dielectric states could exist and be utilized at room temperature, which is seldom found in other ferroelectrics¹².

Fig. 2 Phase transition of the fluorinated compounds. (c) Temperature-dependent dielectric constant curves of *S*-F.

Supplementary Fig. 9: Dielectric constant curves of *R*-F.

7. In Figure 3a, the authors present a comparison of the packing between the R and S crystals of the fluorinated molecule. But the rest of the figure is focused on the structural change. It would be much clearer to have a comparison of the LTP and HTP here instead (or in addition to) for an overview of what's shown in the rest of the figure.

Response: We thank the reviewer for kind suggestion. As suggested by the reviewer, we have changed Fig. 3a from the comparison of the packing structures of crystals *S*- and *R*-F to the packing structures of *S*-F at LTP and HTP.

The corresponding changes in the revised version:

Fig. 3 Crystal structures of *S*-F. (a) Packing views of *S*-F at LTP and HTP along the *a*-axis. Green arrow represents 2_1 screw axes. Blue dotted lines represent C-H...F interactions. (b) Partial schematic diagrams of TEMPO units at LTP and HTP. (c) Comparison of the cell lattices in the LTP (blue) and HTP (red). (d) Variation of unit cell parameters in the heating-cooling mode. Abrupt changes correspond to the phase transition. (e) Schematic diagrams of the distance and interaction energy between adjacent molecules at LTP and HTP. H atoms are partly omitted for clarity.

8. Does the HTP change the ferroelectric behavior? Is this a order- disorder transition that switches from ferroelectric or paraelectric states?

Response: We thank the reviewer for constructive comment. In the LTP and HTP, crystals *S*- and *R*-F both crystallize in the monoclinic polar-chiral point group 2.

Generally, paraelectric-to-ferroelectric phase transitions accompany a notable symmetry change. According to the 88 species of full ferroelectric phase transitions derived by Aizu, the transition from point group 2 to 2 is not included. Thus, we think this order-disorder phase transition is isostructural and does not correspond to a paraelectric-to-ferroelectric transition. Neumann's principle defines the connection between the macroscopic physical properties and the macroscopic symmetries of crystals, which states that the symmetry of any physical property of a crystal must include all the symmetry elements of the point group of the crystal (*Chem. Soc. Rev.* 2016, 45, 3811–3827). Ferroelectric behavior should be investigated through some measurements of physical properties such as PFM or ferroelectric loop measurements.

As suggested by the reviewer, we have investigated the ferroelectric behaviors of compounds *S*- and *R*-F in the HTP. Firstly, we conducted P - V hysteresis loop measurements at room temperature on the thin film of *S*-F which was annealed at 413 K for 10 min to obtain the HTP. The typical P - V loop confirms the ferroelectricity of HTP (Supplementary Fig. 20). Then, we carried out PFM measurements at room temperature on the thin film of *S*-F at HTP. We observed the ferroelectric domains and these domains can be electrically switched, confirming the ferroelectricity of compound *S*-F in the HTP (Supplementary Fig. 22). Thus, *S*- and *R*-F have different ferroelectric behaviors at LTP and HTP, and both two phases can be investigated and utilized at room temperature. These two different ferroelectric phases realize the wide bistability over large temperature range, showing great potential in the applications of modern smart memory devices.

Supplementary Fig. 20: The J - V (blue) and P - V (red) curves of S -F at HTP measured at room temperature.

Fig. 5 Ferroelectricity of S -F. (a) The J - V (dotted) and P - V (solid) curves of S -F showing a typical ferroelectric hysteresis loop. Vertical PFM phase (b) and amplitude (c) images of S -F overlaid on the 3D topographic image. (d–f) The electrical switching of the ferroelectric domain at LTP by applying ± 150 V voltage on the white dot for 20 s: (d) pristine, (e) after applying a positive bias voltage, and (f) after applying a negative bias voltage. (g–i) The electrical switching of the ferroelectric domain at HTP by

applying ± 120 V voltage in the white boxes: (g) pristine, (h) after applying a positive bias voltage, and (i) after applying a negative bias voltage.

Supplementary Fig. 22: Vertical PFM topography (a), amplitude (b) and phase (c) images of *S*-F in the HTP. (b) Inset: phase–voltage hysteresis loop and amplitude–voltage butterfly loop. Topography (d–f), amplitude (g–i)) and phase (j–l) images of *S*-F before and after applying ± 120 V voltage on the boxes.

The corresponding changes in the revised version:

The enantiomeric fluorinated crystals also adopt the chiral-polar space group $P2_1$ at 418 K (Fig. 3a and Supplementary Table 3). This indicates that *S*- and *R*-F undergo isostructural phase transitions.

We have also conducted P – V hysteresis loop measurements at room temperature on the thin film of *S*-F which was annealed at 413 K for 10 min to obtain the HTP. The typical

P - V hysteresis loop confirms the ferroelectricity of HTP (Supplementary Fig. 20). Thus, S - and R -F have different ferroelectric behaviors at LTP and HTP, and both two phases can be investigated and utilized at room temperature.

To further investigate their ferroelectric behavior in the HTP, we carried out the PFM measurement at room temperature on the annealed thin film of S -F. We observed the ferroelectric domains and these domains can be electrically switched, confirming the ferroelectricity of compound S -F in the HTP (Supplementary Fig. 22).

Minor points:

- Curie temperature is listed in fig 1 as an important aspect that makes these molecules better but isn't discussed in the text.

Response: We thank the reviewer for reminding us for that. We have changed the term "Curie temperature" into "phase transition temperature", and we have abbreviated them as T_c for clarity. It is known that ferroelectrics with high T_c are indispensable for practical applications (*Adv. Mater.* 2019, 31, 1902163). As mentioned in the Introduction section, TEMPO is a typical organic radical compound having ferroelectricity and ferroelasticity, but this compound has a limited working-temperature range of a relatively low T_c of 287 K and a low melting point of 311 K. Another organic radical ferroelectric crystal [(NH₃-TEMPO)([18]crown-6)](ReO₄) is devoid of structural phase transition, and thus its ferroelectric state cannot be thermally modulated. In our work, with the combination of introducing homochirality and organic radical component, we have designed a pair of organic single-component ferroelectric compounds S - and R -F, exhibiting a high-temperature structural phase transition at 399 K with a large thermal hysteresis (ΔT) of 132 K. Thus, S - and R -F show better performance in T_c than other radical ferroelectrics.

The corresponding changes in the revised version:

Unfortunately, this compound has a limited working-temperature range of a relatively

low phase transition temperature (T_c) of 287 K and a low melting point of 311 K.

- Could the Authors comment on the disorder observed in the SCXRD? Is it fully rotating, or in two distinct states or vibrating back and forth?

Response: We greatly appreciate the reviewer's comments. The TEMPO components in crystals *S*- and *R*-F are partially disordered and the corresponding molecules exhibit a back-and-forth vibrating motion in the HTP.

The corresponding changes in the revised version:

As the temperature increased, the TEMPO radical units displayed a partially disordered state, exhibiting a back-and-forth vibrating motion, and thus the C–H···F interactions disappeared in the HTP (Fig. 3b).

- Discussion of DSC jumps back and forth between interactions and the DSC data along with I suspect the simulation results discussed later which is confusing.

Response: We greatly appreciate the reviewer's comments. According to the suggestion by the reviewer, we have substantially revised the manuscript.

The corresponding changes in the revised version:

Differential scanning calorimetry (DSC) was used to investigate the thermodynamic phase transition of NDI derivatives (Fig. 2a and Supplementary Fig. 7). As shown in Supplementary Fig. 7, we did not observe the structural phase transitions of the parent compounds *S*-/*R*-H and racemic compounds *Rac*-H/F. H/F substitution provides new possibilities to induce phase transition through the regulation of molecular rotational energy barriers and intermolecular forces¹². As expected, the fluorinated compounds showed thermal anomalies at 399 K upon heating and 267 K upon cooling in both powder and single-crystalline forms (Fig. 2a and Supplementary Fig. 7b). They exhibited reversible first-order phase transitions with large thermal hysteresis (ΔT) of 132 K. On the basis of Boltzmann's equation, N , the ratio of the number of possible

configurations in the HTP and LTP, was deduced to be 4.3 and 5.1 for *S*- and *R*-F respectively (Supplementary Fig. 7b)³⁷. This indicates that *S*- and *R*-F undergo order-disorder phase transitions.

- More details of how the *N* values in the DSC discussion were obtained would be nice.

Response: We thank the reviewer for reminding us for that. Taking *S*-F as an example, ΔS is the entropy change at phase transition temperatures, and $\Delta S_{(S)} = \Delta H / T = 12.13 \text{ J K}^{-1} \text{ mol}^{-1}$, where $T = 399 \text{ K}$, and $\Delta H = 4839.87 \text{ J mol}^{-1}$ (the enthalpy change obtained from the DSC measurement). Similarly, $\Delta S_{(R)}$ is equal to $13.56 \text{ J K}^{-1} \text{ mol}^{-1}$. (*Angew. Chem. Int. Ed.* 2011, 50, 11947–11951). On the basis of Boltzmann's equation, ΔS could be expressed as follows: $\Delta S = R \ln N$, where R is the ideal gas constant ($8.314 \text{ J K}^{-1} \text{ mol}^{-1}$) and N represents the ratio of the number of possible configurations in the HTP and LTP. Thus, the $N_{(S)}$ and $N_{(R)}$ values were deduced to be 4.3 and 5.1, respectively. Detailed information has been added in the Supplementary Information.

The corresponding changes in the revised version:

On the basis of Boltzmann's equation, the change of entropy (ΔS) could be expressed as follows: $\Delta S = R \ln N$, where R is the ideal gas constant ($8.314 \text{ J K}^{-1} \text{ mol}^{-1}$) and N represents the ratio of the number of possible configurations in HTP and LTP. Taking *S*-F as an example, ΔS is the entropy change at T_c , and $\Delta S_{(S)} = \Delta H / T = 12.13 \text{ J K}^{-1} \text{ mol}^{-1}$, where $T = 399 \text{ K}$, and $\Delta H = 4839.87 \text{ J mol}^{-1}$ (enthalpy change obtained from the DSC measurement). Similarly, $\Delta S_{(R)}$ is equal to $13.56 \text{ J K}^{-1} \text{ mol}^{-1}$. Thus, the $N_{(S)}$ and $N_{(R)}$ values were deduced to be 4.3 and 5.1, respectively.

- Figure 2C it appears that 298/243 are swapped.

Response: We thank the reviewer for reminding us for that. As suggested by the reviewers, to highlight the main discovery of our work, we have deleted the discussion of Raman spectra as well as the corresponding figures.

- When discussing hysteresis, listing all of the molecules is probably not helpful and

listing the names makes it difficult to read.

Response: We thank the reviewer for reminding us for that. We have deleted the names of these compounds in the revised manuscript and summarized them in Supplementary Table 2 for clarity.

The corresponding changes in the revised version:

To our knowledge, this large thermal hysteresis of *S*-/*R*-F is extremely rare in molecular ferroelectrics whose ΔT do not exceed 70 K³⁹ (Supplementary Tables 1 and 2).

Supplementary Table 2: Molecular ferroelectrics with large thermal hysteresis.

Compound or acronym in the original publication	ΔT	References
diisopropylammonium bromide	8 K	Science 2013, 339, 425–428
D -chiro-inositol-SiMe ₃	15 K	Angew. Chem. Int. Ed. 2022, 61, e202210809
TMCM-MnCl ₃	~18 K	Science 2017, 357, 306–309
[3-oxoquinuclidinium]ClO ₄	35 K	J. Am. Chem. Soc. 2019, 141, 1781–1787
MDABCO-NH ₄ I ₃	~56 K	Science 2018, 361, 151–155
[3.2.1-dabco]BF ₄	70 K	J. Am. Chem. Soc. 2020, 142, 1995–2000

- The way the jumping crystal effect/optical properties/magnetic/ferroelectric properties sections are presented is confusing. Having so many disparate parts in a single figure makes it difficult to follow.

Response: We thank the reviewer for reminding us for that and we have made modifications for clarity.

- The simulated structure transition should probably be discussed before the optical properties as this seems to add confirmation for the martensitic behavior.

Response: We thank the reviewer for reminding us for that and we have substantially revised the manuscript for clarity.

Response to Reviewer 4

Comments:

The molecular system developed by Zhang et al. is a highly interesting multi-functional system that exhibits unique martensitic transition and thermosensitive effects as well as ferroelectricity. Additionally, the system displays properties such as photochromism and paramagnetism. All these properties are characterized through the adequate set of experiments. By reading the manuscript, this reviewer was fascinated by the material and could imagine its potential for unprecedented applications. However, I believe that the manuscript does not fully capture the appeal of the material system, considering its great potential. This is particularly because the research fails to establish an intriguing connection between martensitic transition behavior and ferroelectricity. In addition, there are several points that should be revised: 1) the paper includes so many properties of the system that are somewhat unrelated to the key findings (martensitic transition and ferroelectricity), 2) when the authors are describing their experimental results, there are several logical leaps, and 3) there are so many typographical errors. Despite these issues, the material system presented in this paper is still novel and very interesting, therefore, it seems reasonable to offer the authors an opportunity to revise their manuscript.

Response: We sincerely thank reviewer 4 for spending his/her valuable time carefully reviewing our manuscript, the in-depth comments, as well as his/her compliments on the impact and significance of our work. According to the reviewers' comments and suggestions, we have carried out new experiments, addressed all problems and made substantial modifications to our manuscript. Specifically, we have highlighted the key finding of our work in the main text (for details, please see our responses to reviewers' comments and the revised manuscript). We have also reorganized our manuscript and moved some parts unrelated to our main storyline to the Supplementary Information in order to highlight the key findings of martensitic phase transition as well as its combination with ferroelectricity. We have carefully revised our manuscript to avoid

typographical errors. The Tables and Figures shown in the revised manuscript and Supplementary Information are marked with purple, and the response are in blue color.

1. This system exhibits an enormous hysteresis associated with martensitic transition, which can be considered as the biggest advantage. This factor enables easy analysis of characteristics of the low-temperature phase (LTP) and high-temperature phase (HTP) at room temperature. This implies the possibility of forming one more state through phase transition and can be considered as a switch concept of a memory device. Therefore, this reviewer suggests that the authors explore the possibility of associating this characteristic with ferroelectricity.

Response: We greatly appreciate the reviewer's comments. As indicated by the reviewer, the biggest advantage of our work is the large thermal hysteresis associated with martensitic transition. This feature makes both high- and low-temperature ferroelectric states for *S*- and *R*-F can be easily investigated and utilized at room temperature, which is seldom found in other high-temperature ferroelectric materials (Fig. 1). Such rare characteristics make these compounds have great application potential in memory devices acting as the switch concept. We totally agree with the inspiring idea of reviewer 4. To further complete this idea, we have carried out new experiments and confirmed the ferroelectricity of HTP by the $P-V$ hysteresis loop and PFM measurements (Supplementary Fig. 20 and Fig. 5). Thus, both the high- and low-temperature ferroelectric states for *S*- and *R*-F can be investigated and utilized at room temperature. Accordingly, the feature of bistable ferroelectric states at room temperature makes *S* and *R*-F a new type of functional material with great application potential in next-generation smart memory devices with multiple physical channels.

Fig. 1 Comparison between traditional high-temperature ferroelectrics and *S-/R-F*. (a) Traditional high-temperature ferroelectrics show only one ferroelectric phase at room temperature. (b) The combination of martensitic phase transition and ferroelectricity in *S-/R-F* brings forth bistability with two different ferroelectric phases at room temperature. LTP and HTP represent low-temperature phase and high-temperature phase respectively. RT FE, RT FE1, and RT FE2 represent different ferroelectric phases at room temperature.

The association of the important martensitic phase transition and ferroelectricity in *S* and *R-F* ferroelectric materials can be concluded as follow: 1) This martensitic phase transition brings forth a large ΔT crossing the high-temperature and low-temperature region, and thus realizes the wide bistability with two different ferroelectric phases at room temperature. 2) For *S* and *R-F* ferroelectric materials, mechanical strain can be induced by the electric field and thermosalient effect. Thus, the actuation of *S* and *R-F* ferroelectric materials can realize the modulation between the millimeter-scale (thermodynamic control) and the picometer-scale movement (electric control) because of the combination of ferroelectricity and martensitic phase transition. Consequently, we think the interaction of ferroelectricity and martensitic phase transition endows ferroelectric materials *S* and *R-F* with many modulable physical properties which are seldom found in other pure ferroelectric materials or jumping crystals.

The corresponding changes in the revised version:

The combination of the martensitic phase transition and ferroelectricity endows *S*- and *R*-F with many advantages over the other ferroelectric and thermosensitive materials. This martensitic phase transition brings forth a large ΔT crossing the high-temperature and low-temperature regions and thus realizes the wide bistability beyond all of the molecular ferroelectrics (Supplementary Tables 1 and 2). For most high-temperature ferroelectric compounds, their high-temperature ferroelectric or paraelectric states can only be obtained at high temperature, which somewhat impedes their study and application (Fig 1)¹². However, for martensitic ferroelectrics *S*- and *R*-F, their physical properties especially ferroelectricity of HTP can be retained at room temperature. Therefore, both the high- and low-temperature ferroelectric states for *S*- and *R*-F can be investigated and utilized at room temperature, which is seldom found in other high-temperature ferroelectric materials¹² (Fig. 1). The feature of bistable ferroelectric states at room temperature makes *S* and *R*-F a new type of functional material with great application potential in next-generation smart memory devices with multiple physical channels. Besides, ferroelectric compounds can induce mechanical strain under the electric field⁴². Meanwhile, this martensitic phase transition causes thermosensitive effect, which could show larger actuation by an order of magnitude in comparison with the conventional converse piezoelectric effect. Thus, the actuation of *S*- and *R*-F can realize the modulation between the millimeter-scale (thermodynamic control) and the picometer-scale movement (electric control)⁵². This rare feature would satisfy specific application scenarios for actuation, thereby making *S*- and *R*-F good candidates for lightweight actuators including sensors, artificial muscles, soft robotics, and energy harvesters¹⁴.

We have also conducted P - V hysteresis loop measurements at room temperature on the thin film of *S*-F which was annealed at 413 K for 10 min to obtain the HTP. The typical P - V hysteresis loop confirms the ferroelectricity of HTP (Supplementary Fig. 20). Thus, *S*- and *R*-F have different ferroelectric behaviors at LTP and HTP, and both two phases can be investigated and utilized at room temperature.

To further investigate their ferroelectric behavior in the HTP, we carried out the PFM measurement at room temperature on the annealed thin film of *S-F*. We observed the ferroelectric domains and these domains can be electrically switched, confirming the ferroelectricity of compound *S-F* in the HTP (Supplementary Fig. 22).

Supplementary Fig. 20: The J - V (blue) and P - V (red) curves of *S-F* at HTP measured at room temperature.

Fig. 5 Ferroelectricity of *S-F*. (a) The J - V (dotted) and P - V (solid) curves of *S-F* showing a typical ferroelectric hysteresis loop. Vertical PFM phase (b) and amplitude

(c) images of *S*-F overlaid on the 3D topographic image. (d–f) The electrical switching of the ferroelectric domain at LTP by applying ± 150 V voltage on the white dot for 20 s: (d) pristine, (e) after applying a positive bias voltage, and (f) after applying a negative bias voltage. (g–i) The electrical switching of the ferroelectric domain at HTP by applying ± 120 V voltage in the white boxes: (g) pristine, (h) after applying a positive bias voltage, and (i) after applying a negative bias voltage.

2. In addition to the comment above, this reviewer recommends that the authors will associate the switch concept with not only ferroelectricity but also other properties of the material system.

Response: We thank the reviewer for instructive suggestions. We have emphasized that the interaction between the martensitic phase transition and ferroelectricity brings forth wide bistability beyond all of the molecular ferroelectrics. Such a rare bistable feature can also be found in dielectric property. As shown in Fig. 2c and Supplementary Fig. 9, the temperature-dependent dielectric constant shows two plateaus. The heating and cooling runs enclose a rectangular loop with a wide temperature window of about 132 K. Thus, the high-dielectric state and low-dielectric state could both exist and be utilized at room temperature, confirming the bistability of *S*- and *R*-F.

The corresponding changes in the revised version:

Such a rare bistable feature was investigated by the dielectric measurement. As shown in Fig. 2c and Supplementary Fig. 9, the step-like anomalies enclosed a rectangular loop with a wide temperature window of 132 K, indicating improper ferroelectricity³⁸. The fluorinated compounds are in the low-dielectric states below 399 K in the heating run. As temperature rose to 399 K, the dielectric constants of *S*- and *R*-F changed from low-dielectric states to high-dielectric ones. When cooling down to room temperature, the high-dielectric states maintained because of the large thermal hysteresis resulted from the martensitic transition. Therefore, both two distinct dielectric states could exist and be utilized at room temperature, which is seldom found in other ferroelectrics¹².

Fig. 2 Phase transition of the fluorinated compounds. (c) Temperature-dependent dielectric constant curves of S-F.

Supplementary Fig. 9: Dielectric constant curves of R-F.

3. The peculiar hysteresis behavior and bistability of LTP and HTP in the system reminded me of the mechanosalient behavior of the terephthalic acid crystal studied by the Naumov group (*J. Am. Chem. Soc.* 2016, 138, 40, 13298).

Response: We thank the reviewer for reminding us for that. Terephthalic acid is a thermosalient crystal with shape-memory and self-healing effects, undergoing a martensitic phase transition at around 350 K (*J. Am. Chem. Soc.* 2016, 138, 13298–13306). This phase transformation can be triggered by mechanical stress, whereby crystals leap several centimeters in the air because of the sudden release of strain. In

comparison with our work, terephthalic acid crystal shows intriguing shape-memory and self-healing effects, and it can leap several centimeters in the air (mechanosalient effect) which is great than that of the *S* and *R-F* ferroelectric materials (thermosalient effect). It should also be highlighted that *S* and *R-F* crystals have ferroelectricity and other interesting physical properties. *S* and *R-F* undergo a high-temperature structural phase transition at 399 K with an ultra-large ΔT of 132 K, while terephthalic acid has a narrower ΔT of 40 K. This indicates that *S* and *R-F* have a wide bistability beyond all of the molecular ferroelectrics, thereby making their physical properties of both HTP and LTP can be utilized at room temperature.

The corresponding changes in the revised version:

During the martensitic phase transition, many intriguing phenomena would occur including the shape memory effect^{18,19}, superelasticity²⁰, negative thermal expansion²¹, and thermosalient effect^{22, 23, 24, 25}.

18. Karothu DP, Weston J, Desta IT, Naumov P. Shape-Memory and Self-Healing Effects in Mechanosalient Molecular Crystals. *J. Am. Chem. Soc.* **138**, 13298–13306 (2016).

4. Due to attempting to cover many topics, the Introduction section appears to be scattered and somewhat not well-organized. In particular, while attempting to introduce multi-ferroic and TEMPO in the first paragraph, the connection between them seems disjointed or not smooth. Such an issue is present throughout the entirety of the Introduction section. Therefore, there is a need to revise the Introduction section to make it more organized and concise.

Response: We thank the reviewer for reminding us for that and we have made substantial modifications to the Introduction part in the revised manuscript. As indicated by reviewers 2 and 4, the connection between the attempt to induce multiferroic property and the introduction of TEMPO component is disjointed. Organic radical compounds generally exhibit paramagnetic properties, although there are some

cases that show intriguing magnetoelectric coupling. However, the introduction of a radical TEMPO component only results in the paramagnetic behavior in compounds *S* and *R-F*. Therefore, we have changed the motivation from multiferroics to multi-functional ferroelectric materials. We have also cited several multi-functional materials with organic radicals to clarify the connection between multi-functional ferroelectric materials and TEMPO. Furthermore, we have made some modifications to the design strategy of ferroelectric materials with martensitic phase transition for clarity. We have added and highlighted the biggest advantages of our work which is the combination of martensitic phase transition and ferroelectricity in the revised Introduction section. We believe that the revised Introduction section is well-organized and logical.

5. There are various contents that seem to deviate from the main discoveries - the martensitic transition and ferroelectricity, which can disrupt the main storyline. This issue is particularly noticeable in the sections on the thermosalient effect and optical properties. While the thermosalient effect is an important clue in proving the martensitic nature of the transition, I feel the separate section of thermosalient effect hinders the readers to follow the overall storyline. Therefore, it might be better to briefly mention the thermosalient effect in the section where structural analysis is performed to support the martensitic nature, and then move the entire section to the supplementary information. Similarly, except for SHG which supports polar structure of the materials, most of the results in the optical properties section do not strongly support the main argument and may hinder the readers from following the paper's main storyline. Therefore, it might be worth considering moving the section to the supplementary information.

Response: We thank the reviewer for reminding us for that and the corresponding manuscript has been substantially revised. To highlight the main discovery of our work, we have moved the descriptions of “jumping crystal” to the section of “phase transition”. Meanwhile, as suggested by the reviewer, we have moved the description of SHG which supports the polar structure of the materials to the section of “crystal structure”

and moved part of the descriptions about the optical properties to the Supplementary Information.

6. (Page 9, Line 177 ~ Page 10, Line 181)

The temperature-dependent Raman study described by the authors lacks supporting evidence. There is no clear basis for the assignment of the peak at around 3000 cm⁻¹. Also, although the authors claim a connection between the peak broadening and TEMPO's order-disorder states, their results do not provide sufficient evidence to support this claim. Although the authors confirmed the disordering of TEMPO in the structural analysis section later on, it should be noted that readers may not be aware of this information. Therefore, it is needed to reconsider the structure of the manuscript.

Response: We thank the reviewer for instructive suggestions. According to the calculated Raman spectra, the bands located at around 3000 cm⁻¹ can be assigned to the C–H stretching vibration, and they broadened/narrowed as the temperature rose/dropped. As suggested by reviewers 3 and 4, the broadening of the C–H stretching peak indicates the increase in disorder, while it is unclear whether this disorder comes from the TEMPO portion of the molecule or the methyl group at the chiral center. We do agree with the reviewers' suggestion, and think it is improper to analyze this order-disorder transition related to the TEMPO component by the C–H stretching peak. We mainly intended to prove the occurrence of structural phase transitions through the measurement of temperature-dependent Raman spectra. The order-disorder transition related to the TEMPO component was investigated by the single crystal structural analyses. Thus, as suggested by the reviewers, to highlight the main discovery of our work, we have deleted the discussion of Raman spectra since phase transition has been proved by many physical measurements.

7. It appears difficult to support the absence of a phase transition in parent counterparts solely based on the presence or absence of C-H...F interactions. Can you provide additional reasons for this?

Response: We thank the reviewer for constructive comment. The question raised by the reviewer is interesting but challenging since nobody even Landau can tell which compound has a structural phase transition before some physical measurements such as DSC measurements. As proven by the DSC measurements, compounds *S* and *R*-H do not show any obvious phase transitions before their melting points. This conclusion comes from experimental results, but we can try to provide some possible explanations for this phenomenon. It is known that a structural phase transition can take place in a crystal when some distortion or reorientation in the active groups is collectively developed. The interaction between active groups and their hosting lattice is responsible for a structural change at a specific thermodynamic condition. (*The Physics of Structural Phase Transitions (2nd Edition)*, Springer Science+Business Media, Inc, 2009) H/F substitution, resembling the isotope effect, could produce new opportunities to the inducing of phase transition by modulating the potential energy barrier of rotating motions (*Trends Chem.* 2021, 3, 1088–1099; *Adv. Mater.* 2020, 32, e2003530).

Herein, as suggested by reviewer 3, we have carefully compared the crystal structures of *S*-F and *S*-H. The fluorinated compounds have similar crystal symmetry and packing arrangements with the H counterparts (Supplementary Fig. 14a and c). Differently, H/F substitution alters the crystal packing environment, namely, the intermolecular interactions. Most importantly, the formation of intermolecular interaction such as the newly formed C–H···F interactions and the enhanced π – π stacking could mainly result in the phase transition inducing in ferroelectrics *S*- and *R*-F. Specifically, fluorination brings new C–H···F interactions between TEMPO components and F atoms of the adjacent molecules (Supplementary Fig. 14a). Besides, the introduction of F atoms has also enhanced other intermolecular interactions such as π – π interactions between NDI planes (Supplementary Fig. 14b and d). The intensity of molecular interaction is mapped onto the Hirshfeld surface by using a red-blue-white color scheme (Supplementary Fig. 15). According to the red regions shown in Hirshfeld surfaces and the distances between NDI layers, *S*-F has a stronger π – π stacked interaction than *S*-H. Besides, *S*-F has C–H···F interaction which does not exist in *S*-H.

Consequently, the enhanced intermolecular interactions play an important role in the phase transition of the fluorinated compounds.

To further confirm above discussion, we have appended the DFT structural relaxation for *S*-/*R*-H. For reference, the optimized lattice constants *a*, *b*, *c* of LTP and HTP are significantly different by -5.58%, -7.43% and 13.19% for *S*-/*R*-F, respectively, as shown in supplementary Table 6. However, the optimized lattice constants of LTP and hypothetical HTP (derived from the F-case) of *S*-/*R*-H are very close: -0.05%, -1.75%, 1.24% for *a*, *b*, *c*, respectively. Considering the numerical precision and soft vdW packing between molecules, such tiny differences are in the tolerant range to conclude that no phase transition occurred for *S*-/*R*-H, which is consistent with the experimental observation and can support above analysis.

Supplementary Fig. 14: (a) Packing view of *S*-F at LTP along the *a*-axis. Red dashed lines represent C-H...F interaction, which do not exist in *S*-H. (b) Schematic diagrams of the distance and interaction energy between adjacent *S*-F molecules. H atoms are partly omitted for clarity. (c) Packing view of *S*-H along the *a*-axis. (d) Schematic diagrams of the distance and interaction energy between adjacent *S*-H molecules.

Supplementary Fig. 15: The Hirshfeld d_{norm} surfaces and the 2D fingerprint plots of *S-F* at LTP (a) and *S-H* (b). The interactions between different atoms are marked in the Figures. According to the red regions shown in Hirshfeld surfaces, *S-F* has C–H \cdots F interactions as well as the enhanced interlayer π - π interactions in comparison with *S-H*. The corresponding changes in the revised version:

As shown in Supplementary Fig. 7, we did not observe the structural phase transitions of the parent compounds *S/R-H* and racemic compounds *Rac-H/F*. H/F substitution provides new possibilities to induce phase transition through the regulation of molecular rotational energy barriers and intermolecular forces¹². As expected, the fluorinated compounds showed thermal anomalies at 399 K upon heating and 267 K upon cooling in both powder and single-crystalline forms (Fig. 2a and Supplementary Fig. 7b).

Differently, the introduction of F atoms changes the crystal packing environment, that is, the intermolecular interactions⁴³ (Supplementary Fig. 14). The molecular Hirshfeld surface and the related 2D-fingerprint plot calculations indicated that the fluorination brought forth C–H \cdots F interactions and enhanced other intermolecular interactions such as π - π stacking between the NDI planes (Supplementary Fig. 15). Accordingly, the newly formed and enhanced intermolecular interactions may play an important role in

inducing phase transition in the fluorinated compounds.

The optimized lattice constants are close to their experimental one, where the lattice constants a , b , c of LTP and HTP are significantly different by -5.58%, -7.43% and 13.19%, respectively, as compared in Supplementary Table 6. What's more, the internal energy of HTP is indeed higher (51.8 meV/f.u.) than the LTP one, as expected. The above analysis shows that there is indeed a structural phase transition in S -/ R -F. However, the optimized lattice constants of LTP and hypothetical HTP (derived from the F-case) of S -/ R -H are very close: -0.05%, -1.75%, 1.24% for a , b , c , respectively. Considering the numerical precision and soft van der Waals (vdW) packing between molecules, such tiny differences are in the tolerant range to conclude that no phase transition occurred for S -/ R -H which is consistent with the experimental observation.

Also, in the transition of S -/ R -F crystal, the authors pointed out that disappearance of this interaction causes disorder in TEMPO, but I believe that the opposite would make more sense.

Response: We thank the reviewer for constructive comment. We do agree with the statement by the reviewer that the disorder in the TEMPO component causes the disappearance of C–H···F interaction, and the corresponding manuscript has been revised.

The corresponding changes in the revised version:

As the temperature increased, the TEMPO radical units displayed a partially disordered state, exhibiting a back-and-forth vibrating motion, and thus the C–H···F interactions disappeared in the HTP (Fig. 3b).

8. (Page 12, Line 225-227)

It is not sufficient to support the enormous window of bistable states based solely on cohesive energy, especially considering only one neighboring molecule. As the authors already know, this is related to Gibbs free energy and activation energy barrier between

the LTP and HTP, which is described by the authors in the last section. Thus, readers may not accept this conclusion while reading this part. Therefore, it would be helpful to indicate that a more detailed description will be provided through DFT study later in the manuscript.

Response: We thank the reviewer reminding us for that. In the last version, we used the Helmholtz free energy to describe the martensitic transformation process. The Gibbs free energy adds a PV term (P and V are pressure and volume respectively) on the Helmholtz free energy. Since all our experiments were done at ambient condition, indeed it should be related to Gibbs free energy. Following this suggestion, we have updated our figure and discussion. However, it should also be noted that the result does not change too much, since the PV term (the difference between LTP and HTP is only ~ 0.004 meV/f.u.) is small comparing with other terms. Also, a more detailed description is provided to enhance the readability and make it easier for readers to accept. We have moved the DFT calculation section to the end of the description about the interaction energy between the neighboring molecules.

The corresponding changes in the revised version:

The optimized lattice constants are close to their experimental one, where the lattice constants a , b , c of LTP and HTP are significantly different by -5.58%, -7.43% and 13.19%, respectively, as compared in Supplementary Table 6. What's more, the internal energy of HTP is indeed higher (51.8 meV/f.u.) than the LTP one, as expected. The above analysis shows that there is indeed a structural phase transition in S -/ R -F. However, the optimized lattice constants of LTP and hypothetical HTP (derived from the F-case) of S -/ R -H are very close: -0.05%, -1.75%, 1.24% for a , b , c , respectively. Considering the numerical precision and soft van der Waals (vdW) packing between molecules, such tiny differences are in the tolerant range to conclude that no phase transition occurred for S -/ R -H which is consistent with the experimental observation.

9. (Page 11 Line 216-219)

The change in lattice parameters is remarkably surprising. Typically, for the occurrence of martensitic transitions, there should be an invariant plane. Are there any clues regarding this? Also, was there any evidence of interface motion or phase front sweeping associated with the phase transition?

Response: We thank the reviewer for constructive comment. Organic martensitic materials have common features such as rapid progression of the phase front (the habit plane), preservation of crystal symmetry, and conformational and packing similarity between the two phases (*Nat. Commun.* 2014, 5, 4811; *J. Am. Chem. Soc.* 2013, 135, 12241–12251; *J. Am. Chem. Soc.* 2013, 135, 13843–13850; *Angew. Chem. Int. Ed.* 2017, 56, 8104–8109). Among them, the study of the habit plane contributes to the deep understanding of orientation relationships as well as the phase transition behavior. In general, the sudden release of the accrued elastic strain occurs in the progressing habit plane; however, this study is challenging due to the short time scales of their transitions and frequent disintegration with explosive outcomes (*J. Am. Chem. Soc.* 2016, 138, 13298–13306). Besides, the changes in lattice parameters of these compounds are different, and some of them are relatively large, such as terephthalic acid (*Nat. Commun.* 2019, 10, 3723), *N'*-2-propyldene-4-hydroxybenzohydrazide (*Sci. Rep.* 2016, 6, 29610). For example, Naumov *et al.* used SEM to investigate the surface morphology change of the terephthalic acid crystal after bending and pointed out its habit plane (as shown in Figure R7). The line that runs across the center of the crystal (marked with red arrows) corresponds to the habit plane (inter-phase boundary) between the two phases.

Figure R7. SEM images of the crystal after bending with striations from the grain

boundaries visible on the surface.

Zhang *et al.* reported a one-dimensional hybrid perovskite ferroelastic crystal (NMEA)PbI₃ (NMEA = *N*-methylethylammonium), showing near-room-temperature martensitic actuation approximately along the *c* axis. The habit plane of it is on (001). In our work, the phase transition is accompanied by a dramatic expansion of the unit cell along *a* by 10.4/10% and *b* by 7.2/7.4% and contraction along *c* by 12.6/12.5%, while the overall volume barely changes (Figs. 3D and 3E, Supplementary Table 5). Such a dramatic change in lattice parameters as well as the jumping behavior of *S* and *R*-F crystal instead of expansion make the study of their habit plane difficult. As shown in Figure R8, we can clearly observe a morphology change of *R*-F crystal along the long side of the prism, and suggest the habit plane may be perpendicular to the long side of the crystal.

Figure R8. Optical images of the morphology change in *R*-F crystals upon heating. Scale bar: 100 μm .

10. Because the materials are new, the chemical structure should be characterized by NMR, EA, and MS.

Response: We greatly appreciate the reviewer's comments. According to the suggestion by the reviewer, EA and MS measurements have been performed on the compounds *S*-F and *S*-H. The results are shown below:

N-[(*S*)-1-phenylethyl]-*N'*-[1-oxyl-2,2,6,6-tetramethyl-4-piperidinyl]-1,4,5,8-naphthalenediimide, *S*-H (C₃₁H₃₀N₃O₅•, 524.22). Elemental analyses: calcd C, 70.98; H, 5.76; N, 8.01; O, 15.25. found C, 70.71; H, 5.15; N, 8.01; O, 15.28. HRMS (ESI, *m/z*) calcd: 525.2219 [M+H]⁺; found 525.2257. *N*-[(*S*)-1-(4-fluorophenyl)ethyl]-*N'*-[1-oxyl-2,2,6,6-tetramethyl-4-piperidinyl]-1,4,5,8-naphthalenediimide, *S*-F

(C₃₁H₂₉FN₃O₅•, 542.21). Elemental analyses: calcd C, 68.62; H, 5.39; N, 7.74; O, 14.74. found C, 68.53; H, 4.95; N, 7.71; O, 14.77. HRMS (ESI, m/z) calcd: 543.2125 [M+H]⁺; found: 543.2161. These results confirm the chemical structures of the synthesized compounds.

¹H NMR and ¹³C NMR signals of the TEMPO skeleton could not be observed owing to the paramagnetic nature of the nitroxide spin labels (*Inorg. Chem.* 2023, 62, 14, 5543–5552, *J. Org. Chem.* 1975, 40, 3145–3147). Thus, we cannot provide the NMR data. Besides, the structures of the synthesized compounds have been verified by single-crystal XRD, PXRD, IR, and EPR. Therefore, we believe that we have fully characterized their chemical structures.

The corresponding changes in the revised version:

N-[(*S*)-1-phenylethyl]-*N'*-[1-oxyl-2,2,6,6-tetramethyl-4-piperidiny]l-1,4,5,8-naphthalenediimide, *S*-H (C₃₁H₃₀N₃O₅•, 524.22). Elemental analyses: calcd C, 70.98; H, 5.76; N, 8.01; O, 15.25. found C, 70.71; H, 5.15; N, 8.01; O, 15.28. HRMS (ESI, m/z) calcd: 525.2219 [M+H]⁺; found 525.2257.

N-[(*S*)-1-(4-fluorophenyl)ethyl]-*N'*-[1-oxyl-2,2,6,6-tetramethyl-4-piperidiny]l-1,4,5,8-naphthalenediimide, *S*-F (C₃₁H₂₉FN₃O₅•, 542.21). Elemental analyses: calcd C, 68.62; H, 5.39; N, 7.74; O, 14.74. found C, 68.53; H, 4.95; N, 7.71; O, 14.77. HRMS (ESI, m/z) calcd: 543.2125 [M+H]⁺; found: 543.2161.

Elemental analyses and high-resolution mass spectroscopy (HRMS)

Elemental analyses were carried out on an Elementar Vario EL cube with CHNO mode. The mean content values of C, H, N and O were calculated from twice the measurements of each compound. HRMS was obtained using a Q-TOF instrument equipped with an ESI source.

Supplementary Fig. 28: The HRMS spectrum of S-F.

Supplementary Fig. 29: The HRMS spectrum of S-H.

11. Thin films are used for characterizing the ferroelectricity of the materials. However, the structure or the phase of the films is not characterized yet.

Response: We thank the reviewer for instructive suggestions. We have investigated the structure of the R-F thin film through PXRD measurements. The PXRD pattern of the thin film is consistent with the simulated one derived from the single-crystal structure at 298 K, indicating that the thin film and single crystal of compound R-F have the same structure.

The corresponding changes in the revised version:

Supplementary Fig. 18: The measured PXRD pattern of the *S-F* thin film matches well with the simulated one derived from the single-crystal structure at 298 K, indicating that the thin film and single crystal of compound *S-F* have the same structure.

12. (page 3, Line 29) Is 'shape memory a ferroelectric' a typo?

Response: We thank the reviewer for reminding us. We have made corrections to the corresponding description.

The corresponding changes in the revised version:

Ferroelectric is a type of important functional material whose ferroelectric state can be modulated under various external stimuli including electric, light, magnetic, and thermal fields.

13. (page 9, Line 175) I think broadened or similar word is missing in 'since the bandwidths in Raman spectra arises from...'

Response: We thank the reviewer for reminding us. We have substantially revised the Raman section and this sentence has been deleted.

14. (page9, Line 177-178) do you mean molecule not molecular in the sentence 'One is that the variation of Raman bands under 200 cm⁻¹ could be caused by the external modes of the fluorinated molecular.'?

Response: We thank the reviewer for reminding us. We have substantially revised the Raman section and this sentence has been deleted.

15. (Line 189) crystalized is a typo.

Response: We thank the reviewer for reminding us. We have corrected the corresponding description.

The corresponding changes in the revised version:

Rac-H and *Rac*-F both crystallized in the non-chiral polar space group *Pc* (Supplementary Fig. 10, Tables 3 and 4).

16. (Page 12, Line 222-224) Fig. 3C should be changed to Fig. 3b.

Response: We thank the reviewer for reminding us. We have made corrections to the corresponding description.

The corresponding changes in the revised version:

This movement is specifically realized as that the distance between π -stacked NDI units decreased by 0.113 Å in the vertical direction and increased by 0.777 Å in the lateral direction from LTP to HTP (Fig. 3e). The interaction energy of the π - π interaction between NDI units was calculated by the UNI force field in the Mercury program⁴⁴. The calculated energies are $-79.8 \text{ kJ mol}^{-1}$ in the LTP and $-60.7 \text{ kJ mol}^{-1}$ in the HTP, respectively (Fig. 3e).

Fig. 3 Crystal structures of S-F. (e) Schematic diagrams of the distance and

interaction energy between adjacent molecules at LTP and HTP. H atoms are partly omitted for clarity.

17. all the abbreviations should be defined when the terminologies first appear. For instance, DFT in page 7, line 126 is not defined before.

Response: We thank the reviewer for reminding us. We have made corresponding revisions in the main text.

The corresponding changes in the revised version:

The density functional theory (DFT) results show that this thermal hysteresis can be attributed to the large barrier and entropy change during this martensitic phase transition.

18. There are many typos throughout the manuscript. The authors are encouraged to correct them diligently.

Response: We thank the reviewer for reminding us for that. Following the reviewer's suggestion, we have substantially revised the manuscript.

😊 Finally, we sincerely thank the reviewers for their helpful comments and suggestions again. By learning those comments, performing new experiments and re-analyzing data, we have gained a lot of in-depth understanding of these materials. To express our sincere appreciation, we have added one sentence in the Acknowledgement part of the manuscript: "The manuscript was improved by the insightful reviews by the anonymous reviewers."

REVIEWER COMMENTS

Reviewer #1 (Remarks to the Author):

The authors have carefully considered all remarks. The revised version of the manuscript is recommended for publication in its present form.

Reviewer #3 (Remarks to the Author):

I really appreciate the author's thorough response to my points. The comparison between the HTP and LTP and amongst the R and S versions of these molecules is much more consistent and easier to understand. Additionally, the analysis of the data presented is much deeper and provides better insight into these materials. The paper also does a better job conveying the importance and impact of obtaining dual ferroelectric and martensitic transition materials. Overall, I think the authors have significantly improved the manuscript discussion of the results and presented a much stronger case for the interest in studying these materials. However, there were a couple of points I would like the authors to consider before the paper is finally published.

1. The authors state there are differences between the S and R-F ferroelectric behavior, but it is not clear what those differences are and how they impact potential device applications.
2. Previously, I mentioned the lack of clear motivation for why this material is interesting compared to other similar systems. While the authors have done a good job clarifying the value of these materials, I think the discussing the applications in the discussion section is still weak at parts. It is still not obvious what "specific applications scenarios" are important for control of the strain at picometer scales and millimeter scales in actuators.
3. I really appreciate the detailed discussion and further analysis of the Raman data. I understand the desire to keep the paper concise with the removal of this data from the paper. However, I am not sure that was necessary here and it adds important nuance to the discussion of the transition mechanism. The Raman data suggest the disorder in the tempo moiety was a result of the polymorph transition as opposed to the driving force which is important to the claims on the transition mechanism. This along with your discussion of the fluorine and tempo moiety I think are especially important to consider when rationally designing materials with this behavior. I think the data are compelling and wanted to see a more in-depth analysis and discussion in the paper so I hope the authors will reconsider this removal.

Minor points

Line 19 in the introduction, check the grammar with the word ferroelectric

Line 261 "both two phases" should be deleted

In figure 5, the labels for h and I indicate a different voltage and a frequency value instead of the same conditions as the d,e and f. Is there a reason different conditions were used? Perhaps some of the labels were switched as well, since figure d is labeled , but f has no condition labeled?

Also, in figure 5 It would be nice to have a similar box in e f on h and i.

Reviewer #4 (Remarks to the Author):

The reviewer carefully examined the authors' point-by-point responses. There have been substantial improvements compared to the previous version, and it is evident that the authors made considerable efforts to incorporate comments from the reviewers. Nevertheless, this reviewer couldn't refrain from providing new comments on the revised version, which are listed below. Because addressing some of the comments may require authors to invest considerable time, this reviewer suggests major revision before acceptance in Nature Communications.

1. [Regarding Comment #1 of the first round] The authors have conducted an appropriate set of experiments to address the most critical comment I raised in the first round, which are reflected in Figure 5 and S20 of the revised manuscript. However, there seems to be no significant difference in hysteresis loops between LTP and HTP. Authors are encouraged to compare the properties more precisely and provide reasons for the small differences. Additionally, can the authors claim to have implemented "a useful new state" based on such a small difference in polarization? These aspects should be thoroughly discussed in the revised manuscript.

2. [Regarding Comment #3 of the first round] It appears that there was some misunderstanding regarding my previous comment. This reviewer was wondering if the crystals also undergo a transition from HTP to LTP due to stimulation such as mechanical poking. This aspect is of interest as it could serve as an alternative pathway for state conversion besides cooling. This reviewer suggests conducting this experiment and mentioning it in the manuscript.

3. [Regarding Comment #4 of the first round] The introduction still needs to be much more concise and clear. There are numerous unnecessary details in each paragraph that are distracting and wordy. Additionally, the lack of connection with preceding paragraphs makes it difficult to grasp the main idea of this work. I strongly recommend rewriting the introduction thoroughly.

(The first paragraph)

In this paragraph, it is important to demonstrate the significance of the main content of this study, which is the controllability of ferroelectricity. It appears that the authors included the sentences, "However, this compound is devoid of structural phase transition, and thereby its ferroelectric state cannot be thermally modulated. Thus, it is challenging to construct organic single-component radical ferroelectric compounds with high T_c and other interesting properties," with this intention. However, without mentioning the importance of thermal modulation capability of the ferroelectric state and the application potential of such a property, it is not possible to justify the significance of materials with corresponding characteristics.

(The second paragraph)

This paragraph primarily introduces the concept of Martensitic transition and mentions that organic ferroelectric materials exhibiting Martensitic transitions are rare. However, there is no discussion about the anticipated effects when Martensitic transition and ferroelectricity are coupled. In other words, this paragraph lacks coherence with the previous paragraph. Additionally, thermosalient materials, which are not dealt mainly in this study, are excessively described, and distracts readers' attention quite a lot.

(The third paragraph)

In this paragraph, the design strategy for the material is mainly described. While many papers introduce such strategies in the introduction, in this manuscript, it appears to be overly verbose and lacking coherence with the second paragraph. Considering this, it might be better to move this content to the main body of the manuscript. Alternatively, it is suggested to condense the essential information and include it in the fourth paragraph.

(The fourth paragraph)

In this paragraph, all the key points are nicely summarized. However, as mentioned above, it is quite challenging to anticipate the content of the fourth paragraph while reading the first three paragraphs. It is particularly important to build up the content gradually from the first paragraph to enable better anticipation of the fourth paragraph.

Reviewer #1 (Remarks to the Author):

The authors have carefully considered all remarks. The revised version of the manuscript is recommended for publication in its present form.

Response: We sincerely thank reviewer 1 for spending his/her valuable time carefully reviewing our manuscript, the in-depth comments, as well as his/her compliments on the impact and significance of our work. We have learned a lot during the revision process and such experience would be a priceless treasure in our future research.

Reviewer #3 (Remarks to the Author):

I really appreciate the author's thorough response to my points. The comparison between the HTP and LTP and amongst the R and S versions of these molecules is much more consistent and easier to understand. Additionally, the analysis of the data presented is much deeper and provides better insight into these materials. The paper also does a better job conveying the importance and impact of obtaining dual ferroelectric and martensitic transition materials. Overall, I think the authors have significantly improved the manuscript discussion of the results and presented a much stronger case for the interest in studying these materials. However, there were a couple of points I would like the authors to consider before the paper is finally published.

Response: We would like to thank the reviewer for the valuable comments and suggestions. We also would like to thank him/her for the seriousness and carefulness in the reviewing process. We have addressed all problems and made proper modifications to our manuscript.

1. The authors state there are differences between the S and R-F ferroelectric behavior, but it is not clear what those differences are and how they impact potential device applications.

Response: We greatly appreciate the reviewer's comments. Generally, the enantiomers of chiral compounds have identical physical properties except for optical activity. As expected, enantiomers *S* and *R-F* exhibit similar UV/Vis and IR spectra as well as thermal, dielectric, magnetic, and ferroelectric behaviors, while their CD spectra show the corresponding Cotton effect. It appears that there was some misunderstanding regarding our manuscript, "Thus, *S* and *R-F* have different ferroelectric behaviors at LTP and HTP". We claimed that the LTP and HTP of the fluorinated compounds are different, while the enantiomers basically have similar physical behaviors including ferroelectricity. Considering the ambiguity in our previous statement, we have revised the corresponding description in the revised manuscript to minimize the misunderstanding.

Although we fail to find the interesting difference between compounds *S* and *R-F* in physical properties, some intriguing phenomena in relation to homochirality are expected to be explored in the future. Homochiral materials are inherently optically active. They have promising prospects because of their unique properties in optical

activity, physiological activity, and chemical selectivity. For instance, the ferroelectrics (*S*)- and (*R*)-3-(fluoropyrrolidinium)MnBr₃ undergo 222F₂-type paraelectric-to-ferroelectric phase transitions and show circularly polarized luminescence (CPL) activities (*J. Am. Chem. Soc.* 2020, 142, 10, 4756–4761). Materials with CPL activity will have a variety of potential applications in optical sensors, asymmetric photosynthesis, and 3D optical displays. Zhao *et al.* found that *L*-Pen-NP films accelerate cell proliferation, whereas the *D*-Pen-NP films have the opposite effect (*Nat. Commun.* 2017, 8, 2007). Their findings will facilitate the development of cell culture in biomedical applications and may help to understand natural homochirality. Furthermore, molecular ferroelectrics with chirality might bring birth to chiral domains with different orientations, which may have different active sites during catalysis (*J. Am. Chem. Soc.* 2020, 142, 15205–15218).

The corresponding changes in the revised version:

Thus, the LTP and HTP of fluorinated compounds have different ferroelectric behaviors and both of them can be investigated and utilized at room temperature.

2. Previously, I mentioned the lack of clear motivation for why this material is interesting compared to other similar systems. While the authors have done a good job clarifying the value of these materials, I think the discussing the applications in the discussion section is still weak at parts. It is still not obvious what “specific applications scenarios” are important for control of the strain at picometer scales and millimeter scales in actuators.

Response: We sincerely thank the reviewer for spending his/her comments and compliments on the significance of our work. In this work, the actuation of *S*- and *R*-F can realize the modulation between the millimeter-scale (thermodynamic control) and the picometer-scale movement (electric control). Realizing the “two-in-one” effect to toggle between picometer- and millimeter-scale in one smart device is useful. Actuation is essential for artificial machines to interact with their surrounding environment and to accomplish the functions for which they are designed. Over the past few decades, there has been considerable progress in developing new actuation technologies. However, controlled motion still represents a considerable bottleneck for many applications and hampers the development of advanced robots (*Sci. Robot.* 2017, 2, eaaq0495). In particular, to adapt to different application scenarios, new materials with different actuation scales are urgently required. Specifically, at the macroscale, soft robots are expected to lead to the development of squeezing,

stretching, climbing, growing, and morphing machines and can act as cooperative systems in many applications. Meanwhile, to satisfy the development of the miniaturization of electronic controllers and sensors, robots exhibiting microscale movement are also required. Among them, picometer-scale actuation can enable robots to achieve many operations with high precision requirements, such as precise positioning, high-precision manufacturing, and minimally invasive procedure. Therefore, materials with both picometer- and millimeter-scale actuation can realize switching between specific application scenarios with multiple actuation requirements from macroscale to microscale. We believe that with the advances in manufacturing technology, such materials can play a role in future electronic fields.

The corresponding changes in the revised version:

This rare feature would satisfy application scenarios requiring multiple-scale actuation, thereby making *S*- and *R-F* good candidates for lightweight actuators, sensors, and soft robots¹⁴.

3. I really appreciate the detailed discussion and further analysis of the Raman data. I understand the desire to keep the paper concise with the removal of this data from the paper. However, I am not sure that was necessary here and it adds important nuance to the discussion of the transition mechanism. The Raman data suggest the disorder in the tempo moiety was a result of the polymorph transition as opposed to the driving force which is important to the claims on the transition mechanism. This along with your discussion of the fluorine and tempo moiety I think are especially important to consider when rationally designing materials with this behavior. I think the data are compelling and wanted to see a more in-depth analysis and discussion in the paper so I hope the authors will reconsider this removal.

Response: We greatly thank the reviewer for the suggestion and his/her compliments on our work. The Raman data can reflect the change in the degree of order when the fluorinated compounds were heated and cooled, confirming the occurrence of the phase transition. However, we do not suggest that these Raman data can reveal the specific mechanism of this phase transition in depth. In the last version, we provided a comprehensive discussion of the phase transition mechanism from the perspective of crystallography; meanwhile, we used Helmholtz free energy based on the DFT calculations to describe this transformation process. Thus, we concluded that the fluorinated compounds undergo phase transitions associated with the order-disorder

movement of the TEMPO component. In the last revision, the C–C stretching vibration bands was suggested to be investigated by reviewer 3 for an in-depth analysis of this order-disorder transition related to the TEMPO component. However, we failed to assign the C–C stretching vibration from the TEMPO portion of the molecule because of the relatively weak Raman intensity as well as the unapparent variation. Thus, we cannot obtain accurate analyses of the phase transition mechanism from the Raman data. After serious consideration of the reviewer’s suggestions, we have re-added the temperature-dependent Raman spectra to Supplementary Information as proof of phase transition.

The corresponding changes in the revised version:

They exhibited reversible first-order phase transitions with large thermal hysteresis (ΔT) of 132 K, which was also confirmed by Raman spectra (Supplementary Fig. 8).

Supplementary Fig. 8: Temperature-dependent Raman spectra of *R-F* at the range of 40–4000 cm⁻¹ (A) and at the range of 1350–1650 cm⁻¹ (B).

Minor points

Line 19 in the introduction, check the grammar with the word ferroelectric

Response: We thank the reviewer for kind suggestion. We have checked and corrected the grammar with the word ferroelectric in Line 19 and adjacent paragraphs. Considering the request of reviewer 4, we have revised the Introduction section, and this sentence has been changed.

The corresponding changes in the revised version:

This kind of material basically undergoes solid-solid structural phase transition and thus has at least two stable low-energy states that can be modulated.

Line 261 “both two phases” should be deleted

Response: We thank the reviewer for instructive suggestions. It appears that there was some misunderstanding regarding our manuscript, “Thus, *S* and *R-F* have different ferroelectric behaviors at LTP and HTP”. We claimed that the LTP and HTP of the fluorinated compounds are different, while the enantiomers basically have similar physical behaviors including ferroelectricity. Considering the ambiguity in our previous statement, we have revised the corresponding description in the revised manuscript to minimize the misunderstanding.

The corresponding changes in the revised version:

Thus, the LTP and HTP of fluorinated compounds have different ferroelectric behaviors and both of them can be investigated and utilized at room temperature.

In figure 5, the labels for h and I indicate a different voltage and a frequency value instead of the same conditions as the d,e and f. Is there a reason different conditions were used? Perhaps some of the labels were switched as well, since figure d is labeled, but f has no condition labeled?

Also, in figure 5 It would be nice to have a similar box in e f on h and i.

Response: We thank the reviewer for reminding us for that. Domain switching process involves the reverse or reorientation of dipole moments of the surface sample under the electric field. Typically, the polarization switching process contains the following steps: (1) domain nucleation, (2) forward extension, (3) lateral expansion, and (4) domain merging (*Phys. Rev. Lett.* 2014, 112, 247603; *Adv. Electron. Mater.* 2016, 2, 1600038). Different processes may require different activation energy of the domain wall. Generally, the energy required for domain nucleation is greater than the one for the motion of the domain walls (*J. Am. Chem. Soc.* 2017, 139, 3954–3957). In our work, as shown in Fig. 5, the domain switching process of compound *S-F* in the LTP and HTP only involves the domain wall movement instead of the domain nucleation because of its relatively high coercive voltage. For domain switching tests based on PFM, we have several methods to apply the external electric field, including the electric poling on a region, along a line, or at a point. We performed the first electric poling method to investigate the ferroelectricity of compound *S-F* in the HTP. As shown in Fig. 5g–i, the electrical switching of their ferroelectric domain was realized

by applying a DC tip bias of ± 120 V with the scan rate of 1 Hz in the white boxes (divided into a grid of 256×256 points). The scan rate of 1 Hz represents applying voltage on 256×256 points in 256 s. In other words, the voltage applied on each point in the selected region is ~ 0.0039 s at HTP. Besides, electric writing on a selected point was also carried out to investigate the domain switching process of compound S-F at LTP (Fig. 5d–f). After the application of voltage pulses of ± 150 V on the white dot for 20 s, the movement of the domain wall is achieved. Thus, the electric switching of the ferroelectric domain at HTP can be realized with a relatively lower pulse amplitude and a shorter pulse duration. This result is reasonable since the P – V hysteresis loop measurements indicate that the coercive voltage of HTP is lower than that of the LTP (Supplementary Figs. 19 and 20). According to the reviewer’s suggestions, we have revised the labels and marks for clarity.

The corresponding changes in the revised version:

Fig. 5 Ferroelectricity of S-F. (d–f) The electrical switching of the ferroelectric domain at LTP by applying ± 150 V voltage on the white dots for 20 s: (d) pristine, (e) after applying a positive bias voltage, and (f) after applying a negative bias voltage. The regions of domain wall movement are marked with white dashed circles. (g–i) The electrical switching of the ferroelectric domain at HTP by applying ± 120 V voltage in the white boxes: (g) pristine, (h) after applying a positive bias voltage, and (i) after applying a negative bias voltage.

Reviewer #4 (Remarks to the Author):

The reviewer carefully examined the authors' point-by-point responses. There have been substantial improvements compared to the previous version, and it is evident that the authors made considerable efforts to incorporate comments from the reviewers. Nevertheless, this reviewer couldn't refrain from providing new comments on the revised version, which are listed below. Because addressing some of the comments may require authors to invest considerable time, this reviewer suggests major revision before acceptance in Nature Communications.

Response: We sincerely thank the reviewer for spending his/her valuable time on carefully reviewing our manuscript, the in-depth comments, as well as his/her compliments on the impact and significance of our work. We have addressed all problems and made substantial modifications to our manuscript.

1. [Regarding Comment #1 of the first round] The authors have conducted an appropriate set of experiments to address the most critical comment I raised in the first round, which are reflected in Figure 5 and S20 of the revised manuscript. However, there seems to be no significant difference in hysteresis loops between LTP and HTP. Authors are encouraged to compare the properties more precisely and provide reasons for the small differences. Additionally, can the authors claim to have implemented “a useful new state” based on such a small difference in polarization? These aspects should be thoroughly discussed in the revised manuscript.

Response: We greatly appreciate the reviewer's comments. The most intrinsic and valuable feature of ferroelectrics is certainly the switchable spontaneous polarization (P_s) under an external electric field (E) greater than the coercive field (E_c). Ferroelectric materials with large P_s and low coercive voltage (V_c) are more desirable for practical applications (*J. Am. Chem. Soc.* 2018, 140, 8051–8059). As suggested by the reviewer, we have precisely compared the physical properties of these two states and provided reasons for the small but important differences. Because of the similar molecular configuration and crystal structure, there is no significant difference between the P_s of LTP and HTP. However, according to the P - V hysteresis loop measurements, the V_c of HTP (~95 V) is lower than that of the LTP (~116 V). The low V_c will make the utilization of ferroelectricity easier. Meanwhile, the difference in V_c can also be proved in PFM measurements. As shown in Fig. 5g–i, the electrical domain switching of compound S-F in the HTP was realized by applying a DC tip

bias of ± 120 V with the scan rate of 1 Hz in the white boxes (divided into a grid of 256×256 points). The scan rate of 1 Hz represents applying voltage on 256×256 points in 256 s. In other words, the voltage applied on each point in the selected region is ~ 0.0039 s at HTP. Then, electric writing on a selected point was also carried out to investigate the domain switching process of compound *S-F* at LTP (Fig. 5d–f). After the application of voltage pulses of ± 150 V on the white dot for 20 s, the movement of the domain wall is achieved. Thus, the electric switching of the ferroelectric domain at HTP can be realized with a relatively lower pulse amplitude and a shorter pulse duration. The difference in V_c between HTP and LTP may be attributed to the variation of intermolecular interactions (Fig. 3). The weaker interactions in the HTP (as fully discussed in our last revised manuscript) make the polarization easier to be switched.

Apart from ferroelectricity, different dielectric constants also make HTP different from LTP. As shown in Fig. 2c and Supplementary Fig. 9, the temperature-dependent dielectric constant shows two plateaus. The heating and cooling runs enclose a rectangular loop with a wide temperature window of about 132 K. Thus, the high-dielectric state and low-dielectric state could both exist and be utilized at room temperature, confirming the bistability of *S-* and *R-F*. Thus, both the differences in V_c and dielectric behaviors make HTP a useful new state. As suggested by the inspirational idea raised by the reviewer in the last revision, such characteristics make these compounds have great application potential in memory devices acting as the switch concept.

The corresponding changes in the revised version:

According to the P – V hysteresis loops, the V_c of HTP is lower than that of LTP. This could be attributed to the weaker interactions between molecules in the HTP which enables easier switching of ferroelectric polarization. The lower V_c makes HTP a useful new state whose ferroelectricity is easier to be investigated and used.

2. [Regarding Comment #3 of the first round] It appears that there was some misunderstanding regarding my previous comment. This reviewer was wondering if the crystals also undergo a transition from HTP to LTP due to stimulation such as mechanical poking. This aspect is of interest as it could serve as an alternative pathway for state conversion besides cooling. This reviewer suggests conducting this experiment and mentioning it in the manuscript.

Response: We thank the reviewer for instructive suggestions. It has been found that terephthalic acid is thermosalient crystal whose shape can reversibly shift between two forms with different crystal habits (*J. Chem. Soc., Faraday Trans.*1994, 90, 1003–1009). In that article mentioned by the reviewer, Naumov *et al.* discovered the mechanosalient effect (mechanically stimulated motility) on that thermosalient crystal terephthalic acid (*J. Am. Chem. Soc.* 2016, 138, 13298–13306). Although the transition of terephthalic acid from form II to form I is spontaneous, the transition of form I to form II is latent and can be triggered by applying local mechanical stress, whereby crystals leap several centimeters in the air. This mechanosalient effect is due to a sudden release of strain that has accrued in the crystal of form I, which is a metastable structure at ambient conditions. They also found that the mechanically triggered transition is much faster than the thermally induced one (Figure R2). As mentioned by the reviewer, this feature is of great interest since it provides an alternative pathway for phase transition besides thermodynamics. Meanwhile, this interesting and rare combination of thermosalient and mechanosalient effects in one compound would arouse great interest and show great application prospects.

Figure R1. Snapshots extracted from optical high-speed video recordings of the transition of form II to I (A) and of form I to II (B). (C) Snapshots extracted from a movie of the transition of form II to form I recorded using variable-temperature serial hot-stage scanning electron microscopy. (D) Snapshots of the mechanically induced phase transition recorded using an optical microscope which triggers the

mechanoslient effect. (*J. Am. Chem. Soc.* 2016, 138, 13298–13306)

As suggested by the reviewer, we have conducted a similar experiment to investigate the mechanoslient effects on our martensitic ferroelectric compounds. The crystal of *S-F* was heated to obtain HTP and further investigation was performed at room temperature. The mechanical poking was applied to the crystal in the HTP. Unfortunately, no obvious phenomenon was observed (Figure R2). Thus, the transition of *S-F* crystal from HTP to LTP could not be realized by mechanical poking. Considering that our materials do not have mechanoslient effect and this effect is not the requisite property of martensitic materials, we did not add the relevant descriptions in the revised manuscript as suggested by the reviewer to prevent irrelevant parts from distracting readers' attention.

Figure R2. Snapshots of *S-F* crystal under an optical microscope: crystal in the HTP obtained after annealing (left); crystal at HTP under external force by using a needle (middle); crystal after external mechanical stimuli (right). The scale bar is 200 μm .

3. [Regarding Comment #4 of the first round] The introduction still needs to be much more concise and clear. There are numerous unnecessary details in each paragraph that are distracting and wordy. Additionally, the lack of connection with preceding paragraphs makes it difficult to grasp the main idea of this work. I strongly recommend rewriting the introduction thoroughly.

Response: We would like to thank the reviewer for the valuable comments and suggestions. We also would like to thank him/her for the seriousness and carefulness in the reviewing process. According to the reviewers' comments and suggestions, the introduction section is revised in a very serious and deliberate way.

(The first paragraph)

In this paragraph, it is important to demonstrate the significance of the main content

of this study, which is the controllability of ferroelectricity. It appears that the authors included the sentences, "However, this compound is devoid of structural phase transition, and thereby its ferroelectric state cannot be thermally modulated. Thus, it is challenging to construct organic single-component radical ferroelectric compounds with high T_c and other interesting properties," with this intention. However, without mentioning the importance of thermal modulation capability of the ferroelectric state and the application potential of such a property, it is not possible to justify the significance of materials with corresponding characteristics.

Response: We thank the reviewer for reminding us. The physical properties of ferroelectric materials with phase transition can be thermally modulated. Such switchable feature makes them promising candidates for sensors, switches, and memory devices. Thus, ferroelectrics with the capability of thermal modulation are of great importance. As suggested by the reviewer, we have deleted part of the description of organic radical ferroelectrics for clarity and highlighted the importance of ferroelectric materials with tunable physical properties.

The corresponding changes in the revised version:

This kind of material basically undergoes solid-solid structural phase transition and thus has at least two stable low-energy states that can be modulated. Such switchable feature makes them promising candidates for sensors, switches, and memory devices⁶. In pursuit of biocompatibility and light weight, organic multi-functional ferroelectric materials are highly desirable for smart medicine and wearable electronics^{7,8}. Organic radicals are attractive building blocks for achieving multifunctionality since they usually exhibit hysteretic responses to external stimuli^{9, 10}. 2,2,6,6-Tetramethylpiperidine-*N*-oxyl (TEMPO) is well-known as a stable single-component radical compound¹¹. Unfortunately, this ferroelectric-ferroelastic compound has a limited working-temperature range of a relatively low phase transition temperature (T_c) of 287 K and a low melting point of 311 K (Supplementary Fig. 1). Recently, another organic radical ferroic crystal [(NH₃-TEMPO)([18]crown-6)](ReO₄) has been found but it is devoid of structural phase transition (Supplementary Fig. 1)¹². Thus, it is challenging to construct organic single-component radical ferroelectric compounds with high- T_c phase transition behavior.

(The second paragraph)

This paragraph primarily introduces the concept of Martensitic transition and mentions that organic ferroelectric materials exhibiting Martensitic transitions are rare. However, there is no discussion about the anticipated effects when Martensitic transition and ferroelectricity are coupled. In other words, this paragraph lacks coherence with the previous paragraph. Additionally, thermosalient materials, which are not dealt mainly in this study, are excessively described, and distracts readers' attention quite a lot.

Response: We are very grateful to the reviewer for the inspiring suggestions. According to the suggestion of the reviewer, we have added the description of the anticipated effects of the combination of martensitic transition and ferroelectricity. Meanwhile, the description of the 'jumping crystal' phenomenon has been moved to the corresponding part of "phase transition behavior" in the Results section.

The corresponding changes in the revised version:

Therefore, martensitic phase transition can endow ferroelectric materials with not only modifiable physical properties but also intriguing functions. The combination of ferroelectricity and martensitic transition will realize the "all-in-one" effect (where multiple functions coexist in one material) and bring new insight into the application of ferroelectric materials.

(The third paragraph)

In this paragraph, the design strategy for the material is mainly described. While many papers introduce such strategies in the introduction, in this manuscript, it appears to be overly verbose and lacking coherence with the second paragraph. Considering this, it might be better to move this content to the main body of the manuscript. Alternatively, it is suggested to condense the essential information and include it in the fourth paragraph.

Response: We would like to thank the reviewer for the valuable comments and suggestions. Considering the importance of phase transition in ferroelectrics mentioned in the first paragraph, we provide a detailed description of the type of phase transition we want to induce in the second paragraph. The second paragraph introduces the advantages of martensitic phase transition for ferroelectric materials and their current dilemma. We believe that when readers understand the advantages and problems faced by such materials through our second paragraph, they naturally want to know the corresponding solutions. Therefore, in the third paragraph, we

provide our design ideas to let readers know that although the implementation of this type of material is difficult, it is not completely clueless. We believe that the design concept described in the third paragraph is reasonable, in line with the readers' curiosity and logical context. Considering the suggestions of the reviewers, we also hope that our article will be more concise and easier to read, so we have extracted the key points from the third paragraph and merged them into the fourth paragraph. We believe that with our efforts, our article will meet your requirements and be more suitable for readers to read.

(The fourth paragraph)

In this paragraph, all the key points are nicely summarized. However, as mentioned above, it is quite challenging to anticipate the content of the fourth paragraph while reading the first three paragraphs. It is particularly important to build up the content gradually from the first paragraph to enable better anticipation of the fourth paragraph.

Response: We thank the reviewer for his valuable comments. We are very glad that you find our work satisfactory. According to your suggestion, we have carefully revised the content of the first three paragraphs and improved the logic of this article.

REVIEWERS' COMMENTS

Reviewer #3 (Remarks to the Author):

The authors have adequately addressed the reviewer remarks and recommend the paper should be accepted in its current form.

Reviewer #4 (Remarks to the Author):

The authors sufficiently addressed all the comments. Therefore, I recommend the acceptance of the manuscript in Nature Communications.

Reviewer #3 (Remarks to the Author):

The authors have adequately addressed the reviewer remarks and recommend the paper should be accepted in its current form.

Response: We would like to express our gratitude to the Reviewer for spending time and effort to help increase the impact of our work.

Reviewer #4 (Remarks to the Author):

The authors sufficiently addressed all the comments. Therefore, I recommend the acceptance of the manuscript in Nature Communications.

Response: We thank the Reviewer for their generally positive assessment of the work and the valuable suggestions.